# GRAPH NEURAL NETWORKS AS GRADIENT FLOWS: UNDERSTANDING GRAPH CONVOLUTIONS VIA ENERGY

## ABSTRACT

Gradient flows are differential equations that minimize an energy functional and constitute the main descriptors of physical systems. We apply this formalism to Graph Neural Networks (GNNs) to develop new frameworks for learning on graphs as well as provide a better theoretical understanding of existing ones. We derive GNNs as a gradient flow equation of a parametric energy that provides a physics-inspired interpretation of GNNs as learning particle dynamics in the feature space. In particular, we show that in graph convolutional models (GCN), the positive/negative eigenvalues of the channel mixing matrix correspond to attractive/repulsive forces between adjacent features. We rigorously prove how the channel-mixing can learn to steer the dynamics towards low or high frequencies, which allows to deal with heterophilic graphs. We show that the same class of energies is decreasing along a larger family of GNNs; albeit not gradient flows, they retain their inductive bias. We experimentally evaluate an instance of the gradient flow framework that is principled, more efficient than GCN, and achieves competitive performance on graph datasets of varying homophily often outperforming recent baselines specifically designed to target heterophily.

## 1 INTRODUCTION

Graph neural networks (GNNs) (Sperduti, 1993; Goller & Kuchler, 1996; Gori et al., 2005; Scarselli et al., 2008; Bruna et al., 2014; Defferrard et al., 2016; Kipf & Welling, 2017; Battaglia et al., 2016; Gilmer et al., 2017) have become the standard ML tool for dealing with different types of relations and interactions. Limitations of GNNs that have recently attracted attention in the literature are *over-smoothing* (node features becoming increasingly similar with the depth of the model, see Nt & Maehara (2019); Oono & Suzuki (2020); Cai & Wang (2020); Zhou et al. (2021)), *over-squashing* (the difficulty of message passing to propagate information on the graph, see Alon & Yahav (2021); Topping et al. (2022)), and poor performance on *heterophilic* data (i.e. where adjacent nodes tend to have different labeles, see Pei et al. (2020); Zhu et al. (2020); Bo et al. (2021); Yan et al. (2021)).

**General motivations and contributions.** In the spirit of neural ODEs (Haber & Ruthotto, 2018; Chen et al., 2018), we regard (residual) GNNs as discrete dynamical systems. A fundamental idea in physics is that particles evolve by minimizing an energy: one can then study the dynamics through the functional expression of the energy. The class of differential equations that minimize an energy are called *gradient flows* and their extension and analysis in the context of GNNs represent the main focus of this work. We study two ways of understanding the dynamics induced by GNNs: starting from the energy functional or from the evolution equations.

**From energy to evolution equations: a new conceptual approach to GNNs.** We propose a general framework where one parameterises an energy functional and then takes the GNN equations to follow the direction of steepest descent of such energy. We introduce a class of energy functionals that extend those adopted for label

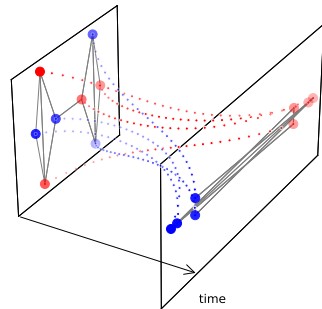

Figure 1: Gradient flow dynamics: attractive and repulsive forces lead to a process able to separate heterophilic labels.

propagation (Zhou & Schölkopf, 2005) and whose gradient flow equations consist of generalized graph convolutions (GCN-type architectures, (Kipf & Welling, 2017)) with *symmetric weights*. We provide a physical interpretation for GNNs as multi-particle dynamics: this new framework sheds light on the role of the 'channel-mixing' matrix used in graph convolutional models as an edge-wise potential inducing attraction (repulsion) via its positive (negative) eigenvalues. We conduct theoretical analysis of the dynamics including explicit expansions of the GNN learned features, showing that differently from other continuous models, the gradient flow can learn to magnify either the low or high frequencies. This also establishes new links to techniques like residual connections and negative edge weights that have been previously used in heterophilic settings. We experimentally evaluate our framework using gradient flow equations yielding a principled variant of GCNs that is also more efficient due to weight symmetry and sharing across layers. Our experiments demonstrate competitive performance on homophilic and heterophilic graphs of varying size.

**From evolution equations to energy: understanding graph convolutions via multi-particle dynamics.** Recent works of Cai & Wang (2020); Bodnar et al. (2022) studied the behaviour of the Dirichlet energy in graph convolutional models in order to determine if (over)smoothing of the features is occurring. The key idea is that the monotonicity of an energy functional along a system of equations conveys significant information about the dynamics, both in terms of its dominating effects and limit points. However, these results are restricted to the classical Dirichlet energy and assume (non-residual) graph convolutions activated by the ReLU nonlinearity. We extend this approach by proving that a much more general multi-particle energy is in fact decreasing along residual graph-convolutions with symmetric weights and with respect to a more general class of nonlinear activation functions. Our result sheds light onto the dynamics of non-linear graph convolutions showing that the 'channel-mixing' matrix used in GCN-type models can be interpreted as a potential in feature space that promotes alignment (repulsion) of adjacent node features depending on its spectrum.

**Outline.** In Section 2 we review non-parametric instances of gradient flows on graphs: the heat equation and label propagation. In Section 3 we extend this approach to the parametric case by introducing a class of energies that generalize the one used for label propagation and whose associated gradient flows are continuous graph convolutions. We provide a physical interpretation for the energy showing that it can induce attraction and repulsion along edges. In Section 4 we discretize the gradient flow into GNN update equations and derive explicit expansions of the learned node representations highlighting how the spectrum of the channel-mixing $\mathbf{W}$ controls whether the dynamics is dominated by the low or high frequencies of the graph Laplacian. To our knowledge, ours is the first analysis that studies the interplay of the spectral properties of the graph Laplacian and the channel mixing matrix. In Section 5 we extend the theory by showing that the same multi-particle energy introduced in Section 3 still decreases along more general graph convolutions with symmetric weights, meaning that the physics interpretation is preserved. In Section 6 we evaluate the framework for node classification on a broad range of datasets.

**Related work.** Our analysis is related to studying GNNs as filters (Defferrard et al., 2016; Hammond et al., 2019; Balcilar et al., 2020; He et al., 2021) and adopts techniques similar to Oono & Suzuki (2020); Cai & Wang (2020). Gradient flows were adapted from geometry (Eells & Sampson, 1964), to image processing (Kimmel et al., 1997), label propagation (Zhou & Schölkopf, 2005) and recently in ML (Sander et al., 2022) for the analysis of Transformers (Vaswani et al., 2017). Our work follows the spirit of GNNs as continuous dynamical systems (Xhonneux et al., 2020; Zang & Wang, 2020; Chamberlain et al., 2021a; Eliasof et al., 2021; Chamberlain et al., 2021b; Bodnar et al., 2022; Rusch et al., 2022).

**Notations.** Let $\mathsf{G} = (\mathsf{V}, \mathsf{E})$ be an *undirected* graph with $n$ nodes. Its adjacency matrix $\mathbf{A}$ is defined as $a_{ij} = 1$ if $(i, j) \in \mathsf{E}$ and zero otherwise. We let $\mathbf{D} = \mathrm{diag}(d_i)$ be the degree matrix and define the *normalized adjacency* $\bar{\mathbf{A}} := \mathbf{D}^{-1/2}\mathbf{A}\mathbf{D}^{-1/2}$. We denote by $\mathbf{F} \in \mathbb{R}^{n \times d}$ the matrix of $d$-dimensional node features, by $\mathbf{f}_i \in \mathbb{R}^d$ its $i$-th row (transposed), by $\mathbf{f}^r \in \mathbb{R}^n$ its $r$-th column, and by $\mathrm{vec}(\mathbf{F}) \in \mathbb{R}^{nd}$ the vectorization of $\mathbf{F}$ obtained by stacking its columns. Given a symmetric matrix $\mathbf{B}$, we let $\lambda_+^{\mathbf{B}}, \lambda_-^{\mathbf{B}}$ denote its most positive and negative eigenvalues, respectively, and $\rho_{\mathbf{B}}$ be its *spectral radius*. $\dot{f}(t)$ denotes the temporal derivative, $\otimes$ is the Kronecker product and 'a.e.' means *almost every* w.r.t. Lebesgue measure. Proofs and additional results appear in the Appendix.

## 2 GRADIENT FLOWS ON GRAPHS: THE NON-PARAMETRIC CASE

In this Section we review important concepts on graphs and two examples of non-parametric gradient flows that partly motivate our approach to the GNN framework.

**What is a gradient flow?** Consider an $N$-dimensional dynamical system governed by the *evolution equation* $\dot{\mathbf{F}}(t) = \text{ODE}(\mathbf{F}(t))$ that evolves some initial state $\mathbf{F}(0)$ for time $t \geq 0$. In deep learning, the discretisation of such differential equations using the Euler method allows to draw a parallel between iterations of a numerical solver and the layers of a neural network (Haber & Ruthotto, 2018; Chen et al., 2018). We say that the evolution equation is a *gradient flow* if there exists an energy functional $\mathcal{E} : \mathbb{R}^N \to \mathbb{R}$ such that $\text{ODE}(\mathbf{F}(t)) = -\nabla\mathcal{E}(\mathbf{F}(t))$. Since $\dot{\mathcal{E}}(\mathbf{F}(t)) = -||\nabla\mathcal{E}(\mathbf{F}(t))||^2$, the energy $\mathcal{E}$ is *decreasing* along the solution $\mathbf{F}(t)$ of such equations. The existence of $\mathcal{E}$ and the knowledge of its functional expression allow for a better understanding of the dynamical system.

**A prototypical gradient flow: heat equation.** Let $\mathbf{F} \in \mathbb{R}^{n \times d}$ be the matrix representation of vector features assigned to each node in $\mathsf{G}$. Its *graph gradient* is defined edge-wise as $(\nabla\mathbf{F})_{ij} := \mathbf{f}_j/\sqrt{d_j} - \mathbf{f}_i/\sqrt{d_i}$. We can then set the *Laplacian* as $\boldsymbol{\Delta} := -\text{div}\,\nabla/2$ (the *divergence* div is the adjoint of $\nabla$), represented by $\boldsymbol{\Delta} = \mathbf{I} - \bar{\mathbf{A}} \succeq 0$. We refer to the eigenvalues of $\boldsymbol{\Delta}$ as *frequencies*: the lowest frequency is always $0$ while the highest frequency is $\rho_{\boldsymbol{\Delta}} \leq 2$ (Chung & Graham, 1997). The *heat equation* on each channel is the system $\dot{\mathbf{f}}^r(t) = -\boldsymbol{\Delta}\mathbf{f}^r(t)$, for $1 \leq r \leq d$. This is an example of gradient flow: if we stack the columns of $\mathbf{F}$ into $\text{vec}(\mathbf{F}) \in \mathbb{R}^{nd}$, we can rewrite the heat equation as

$$\text{vec}(\dot{\mathbf{F}}(t)) = -\nabla\mathcal{E}^{\text{Dir}}(\text{vec}(\mathbf{F}(t))), \tag{1}$$

where $\mathcal{E}^{\text{Dir}} : \mathbb{R}^{nd} \to \mathbb{R}$ is the *(graph) Dirichlet energy* defined by (Zhou & Schölkopf, 2005)

$$\mathcal{E}^{\text{Dir}}(\mathbf{F}) := \frac{1}{4} \sum_{(i,j) \in \mathsf{E}} ||(\nabla\mathbf{F})_{ij}||^2 = \frac{1}{2}\text{trace}(\mathbf{F}^\top \boldsymbol{\Delta}\mathbf{F}) = \frac{1}{2}\langle\text{vec}(\mathbf{F}), (\mathbf{I}_d \otimes \boldsymbol{\Delta})\text{vec}(\mathbf{F})\rangle. \tag{2}$$

$\mathcal{E}^{\text{Dir}}$ measures the *smoothness* of the signal since it accounts for the variations of $\mathbf{F}$ (i.e., its gradient) on the edges. The evolution of $\mathbf{F}(t)$ by the heat equation (1) decreases the Dirichlet energy $\mathcal{E}^{\text{Dir}}(\mathbf{F}(t))$; in the limit $\mathcal{E}^{\text{Dir}}(\mathbf{F}(t \to \infty)) = 0$, attained by the projection of the initial state $\mathbf{F}(0)$ onto $\ker(\boldsymbol{\Delta})$.

**A more general gradient flow: label propagation.** Assume we have a graph $\mathsf{G}$, node-features $\mathbf{F}_0$ and labels $\{\mathbf{y}_i\}$ on $\mathsf{V}_{\text{train}} \subset \mathsf{V}$, and that we want to predict the labels on $\mathsf{V}_{\text{test}} \subset \mathsf{V}$. Zhou & Schölkopf (2005) proposed *label propagation* (LP) where they first extend the input labels $\mathbf{Y}(0)$ outside the training set as $\mathbf{y}_i(0) = \mathbf{0}$ for each $i \in \mathsf{V} \setminus \mathsf{V}_{\text{train}}$ and then solve the equation:

$$\dot{\mathbf{Y}}(t) = -\boldsymbol{\Delta}\mathbf{Y}(t) - 2\mu(\mathbf{Y}(t) - \mathbf{Y}(0)).$$

This is another example of gradient flow; in fact, Zhou & Schölkopf (2005) originally introduced the following energy and then derived the aforementioned update formula in order to minimize it:

$$\dot{\mathbf{Y}}(t) = -\nabla\mathcal{E}^{\text{LP}}(\mathbf{Y}(t)), \quad \mathcal{E}^{\text{LP}}(\mathbf{Y}) := \mathcal{E}^{\text{Dir}}(\mathbf{Y}) + \mu||\mathbf{Y} - \mathbf{Y}(0)||^2. \tag{3}$$

The prediction is then attained by the signal that minimizes both $\mathcal{E}^{\text{Dir}}$ – which enforces smoothness – and the fitting term arising from the available labels (a form of soft boundary conditions).

**Motivations and goals.** In graph ML problems, we often also have node features that can be leveraged for the label prediction. Our goal is to extend the gradient flow formalism from the non-parametric case (heat equation and label propagation) to a deep learning setting, where we (i) parameterise an energy functional and let the GNN equations be the associated gradient flow, and (ii) investigate when existing GNNs admit an energy that is decreasing along their evolution equations.

## 3 GRADIENT FLOWS ON GRAPHS: THE PARAMETRIC CASE

We can think of a (residual) graph neural network as a parametric evolution equation $\dot{\mathbf{F}}(t) = \text{GNN}_{\theta(t)}(\mathsf{G}, \mathbf{F}(t))$ discretized using the Euler method with fixed time step $0 < \tau \leq 1$:

$$\mathbf{F}(t + \tau) = \mathbf{F}(t) + \tau\text{GNN}_{\theta(t)}(\mathsf{G}, \mathbf{F}(t)). \tag{4}$$

Each iteration corresponds to a GNN layer, which in general can have a different set of parameters $\theta(t)$. We choose $\text{GNN}_\theta$ to be the gradient flow of some parametric class of energies $\mathcal{E}_\theta : \mathbb{R}^{n \times d} \to \mathbb{R}$ generalizing $\mathcal{E}^{\text{Dir}}$, resulting in feature evolution by $\dot{\mathbf{F}}(t) = -\nabla \mathcal{E}_\theta(\mathbf{F}(t))$ starting from input features $\mathbf{F}(0)$, with $\{\theta\}$ learned via backpropagation on the task loss function. This approach extends the LP technique to a framework where the parameters we learn can be interpreted as *'finding the right notion of smoothness'* for our task. In fact, minimizing $\mathcal{E}^{\text{LP}}$ as in Equation (3) works only if the labels are smooth—an assumption known as *homophily*. We investigate how learning a more general energy yields gradient flow GNNs that can also perform well on *heterophilic* data.

## 3.1 Energies giving rise to graph-convolutional models

Similarly to the LP approach in Equation (3), our first step consists in choosing a parametric class of energy functionals $\{\mathcal{E}_\theta\}$ giving rise to the GNN equations via gradient flow. GNNs of the convolutional flavor (Bronstein et al., 2021) evolve the features via (4) using some parametric rule $\text{GNN}_\theta(\mathsf{G}, \mathbf{F}_0)$ typically consisting of two operations: applying a shared linear transformation to the features (*'channel mixing'*) and propagating them along the edges (*'diffusion'*). Accordingly, we introduce the class of *(generalized) graph convolutions*:

$$\mathbf{F}(t + \tau) = \mathbf{F}(t) + \tau \, \sigma \left( -\mathbf{F}(t)\boldsymbol{\Omega}_t + \bar{\mathbf{A}}\mathbf{F}(t)\mathbf{W}_t - \mathbf{F}(0)\tilde{\mathbf{W}}_t \right), \tag{5}$$

where the learnable parameters $\{\theta(t)\}$ are the $d \times d$ weight matrices $\boldsymbol{\Omega}_t, \mathbf{W}_t$, and $\tilde{\mathbf{W}}_t$ acting on each node feature vector independently and performing channel mixing; the normalized adjacency $\bar{\mathbf{A}}$ performs the diffusion of features from adjacent nodes. The setting $\tau = 1$, no residual connection, and $\boldsymbol{\Omega}_t = \tilde{\mathbf{W}}_t = \mathbf{0}$ corresponds to GCN (Kipf & Welling, 2017). The case of $\boldsymbol{\Omega}_t \neq \mathbf{0}$ results in an anisotropic instance of GraphSAGE Hamilton et al. (2017), while by choosing $\boldsymbol{\Omega}_t = \mathbf{0}$ and $\mathbf{W}_t$ and $\tilde{\mathbf{W}}_t$ as convex combinations with the identity we recover GCNII (Chen et al., 2020).

We consider a class of energies $\{\mathcal{E}_\theta\}$ consisting of quadratic terms

$$\mathcal{E}_\theta(\mathbf{F}) = \underbrace{\frac{1}{2} \sum_i \langle \mathbf{f}_i, \boldsymbol{\Omega}\mathbf{f}_i \rangle}_{\mathcal{E}_{\boldsymbol{\Omega}}^{\text{ext}}} - \underbrace{\frac{1}{2} \sum_{i,j} \bar{\mathbf{A}}_{ij} \langle \mathbf{f}_i, \mathbf{W}\mathbf{f}_j \rangle}_{\mathcal{E}_{\mathbf{W}}^{\text{pair}}} + \underbrace{\varphi^0(\mathbf{F}, \mathbf{F}(0))}_{\mathcal{E}_{\varphi^0}^{\text{source}}}, \tag{6}$$

parameterised by $d \times d$ weight matrices $\boldsymbol{\Omega}, \mathbf{W}$. We motivate our choice by first recovering the non-parametric cases of Section 2. If $\boldsymbol{\Omega} = \mathbf{W} = \mathbf{I}_d$ and $\varphi^0 = 0$, then $\mathcal{E}_\theta = \mathcal{E}^{\text{Dir}}$ as per Equation (2); choosing $\varphi^0$ as an $L_2$-penalty gives $\mathcal{E}_\theta = \mathcal{E}^{\text{LP}}$ as per Equation (3). We can also recover manifold harmonic energies applied to graphs (see Appendix B.5). More importantly, if $\varphi^0(\mathbf{F}, \mathbf{F}(0)) = \sum_i \langle \mathbf{f}_i, \tilde{\mathbf{W}}\mathbf{f}_i(0) \rangle$, for $\tilde{\mathbf{W}} \in \mathbb{R}^{d \times d}$, we can rewrite

$$\mathcal{E}_\theta(\mathbf{F}) = \langle \text{vec}(\mathbf{F}), \tfrac{1}{2}(\boldsymbol{\Omega} \otimes \mathbf{I}_n - \mathbf{W} \otimes \bar{\mathbf{A}})\text{vec}(\mathbf{F}) + (\tilde{\mathbf{W}} \otimes \mathbf{I}_n)\text{vec}(\mathbf{F}(0)) \rangle \tag{7}$$

and then derive its *gradient flow* as:

$$\dot{\mathbf{F}}(t) = -\nabla_{\mathbf{F}}\mathcal{E}_\theta(\mathbf{F}(t)) = -\mathbf{F}(t)\left(\frac{\boldsymbol{\Omega} + \boldsymbol{\Omega}^\top}{2}\right) + \bar{\mathbf{A}}\mathbf{F}(t)\left(\frac{\mathbf{W} + \mathbf{W}^\top}{2}\right) - \mathbf{F}(0)\tilde{\mathbf{W}}. \tag{8}$$

Since $\boldsymbol{\Omega}, \mathbf{W}$ appear in Equation (8) in a symmetrized way, without loss of generality we can assume $\boldsymbol{\Omega}$ and $\mathbf{W}$ to be *symmetric* $d \times d$ channel mixing matrices. Therefore, Equation (8) simplifies as

$$\dot{\mathbf{F}}(t) = -\mathbf{F}(t)\boldsymbol{\Omega} + \bar{\mathbf{A}}\mathbf{F}(t)\mathbf{W} - \mathbf{F}(0)\tilde{\mathbf{W}}. \tag{9}$$

Thus, a quadratic energy as in Equation (35) leads to continuous linear graph convolutions with *symmetric weights shared over time*. Equivalently, for generalized graph convolutions to fit the gradient flow formalism, the channel-mixing matrices must be symmetric. Importantly, while reducing the number of parameters and offering a gradient flow interpretation of the GNN, this symmetric constraint does not diminish its expressive power (Hu et al., 2019). Next, we show that $\mathcal{E}_\theta$ has a simple interpretation in terms of pairwise forces among adjacent features.

## 3.2 Attraction and repulsion: a physics-inspired framework

**Why gradient flows? A multi-particle point of view.** Consider the node features as particles in $\mathbb{R}^d$ with energy $\mathcal{E}_\theta$. The first term $\mathcal{E}_{\boldsymbol{\Omega}}^{\text{ext}}$ is *independent of the pairwise interactions* and hence represents an 'external' energy in the feature space. The second term $\mathcal{E}_{\mathbf{W}}^{\text{pair}}$ instead accounts for *pairwise interactions* along edges via the symmetric matrix $\mathbf{W}$ and hence represents an 'internal'

energy. We set the source term $\varphi^0$ to zero and write $\mathbf{W} = \mathbf{\Theta}_+^\top \mathbf{\Theta}_+ - \mathbf{\Theta}_-^\top \mathbf{\Theta}_-$, by decomposing it into components with positive and negative eigenvalues. We can then rewrite $\mathcal{E}_\theta$ in Equation (6) as

$$\mathcal{E}_\theta(\mathbf{F}) = \underbrace{\frac{1}{2}\sum_i \langle \mathbf{f}_i, (\mathbf{\Omega} - \mathbf{W})\mathbf{f}_i \rangle}_{dampening} + \underbrace{\frac{1}{4}\sum_{i,j} ||\mathbf{\Theta}_+(\nabla\mathbf{F})_{ij}||^2}_{attraction} - \underbrace{\frac{1}{4}\sum_{i,j} ||\mathbf{\Theta}_-(\nabla\mathbf{F})_{ij}||^2}_{repulsion}, \quad (10)$$

which we have derived in Appendix B. To understand the dynamics induced by the minimization of $\mathcal{E}_\theta$ by the gradient flow (36), recall that the edge gradient $(\nabla\mathbf{F})_{ij}$ measures the difference between features $\mathbf{f}_i$ and $\mathbf{f}_j$. We note that: (i) if $\mathbf{\Omega}$ commutes with $\mathbf{W}$, then the projections of $(\nabla\mathbf{F})_{ij}$ onto $\ker(\mathbf{W})$ *remain invariant* and are preserved along the gradient flow; (ii) The channel-mixing $\mathbf{W}$ encodes *attractive edge-wise interactions* via its positive-definite component $\mathbf{\Theta}_+$ since the gradient terms $||\mathbf{\Theta}_+(\nabla\mathbf{F})_{ij}||$ decrease along the solution of Equation (36), hence resulting in a smoothing effect where adjacent node features $\mathbf{f}_i$ and $\mathbf{f}_j$ are 'aligned'; (iii) The channel-mixing $\mathbf{W}$ encodes *repulsive edge-wise interactions* via its negative-definite component $\mathbf{\Theta}_-$ since the gradient terms $||\mathbf{\Theta}_-(\nabla\mathbf{F})_{ij}||$ increase along the solution of Equation (36), hence resulting in a sharpening effect which could be desirable on heterophilic graphs where we need to disentangle adjacent node representations. Next, we formalize the smoothing vs sharpening effects by introducing a new quantity to monitor along a GNN to assess whether the latter is magnifying the low or high frequencies.

**Low vs high frequency enhancement.** Attractive forces minimize the edge gradients and are associated with smoothing effects which magnify low frequencies, while repulsive forces increase the edge gradients and hence afford a sharpening action enhancing the high frequencies. Since we are interested in finding which frequency is dominating the dynamics, we monitor the Dirichlet energy along the normalized solution: $\mathcal{E}^{\mathrm{Dir}}(\mathbf{F}(t))/||\mathbf{F}(t)||^2$. This is the *Rayleigh quotient* of $\mathbf{I}_d \otimes \mathbf{\Delta}$ and so it satisfies $0 \leq \mathcal{E}^{\mathrm{Dir}}(\mathbf{F})/||\mathbf{F}||^2 \leq \rho_\mathbf{\Delta}/2$ (see Appendix A.2). If the normalized Dirichlet energy is approaching its minimum, then the lowest frequency component is dominating, whereas if the normalized Dirichlet energy is converging to its maximum, then the dynamics is dominated by the highest frequencies. This allows us to introduce the following

**Definition 3.1.** $\dot{\mathbf{F}}(t) = \mathrm{GNN}_{\theta(t)}(\mathsf{G}, \mathbf{F}(t))$ initialized at $\mathbf{F}(0)$ is *Low/High-Frequency-Dominant* (L/HFD) if $\mathcal{E}^{\mathrm{Dir}}(\mathbf{F}(t))/||\mathbf{F}(t)||^2 \to 0$ (respectively, $\mathcal{E}^{\mathrm{Dir}}(\mathbf{F}(t))/||\mathbf{F}(t)||^2 \to \rho_\mathbf{\Delta}/2$ for $t \to \infty$.

In Appendix B.2 we provide justifications and explicit examples. If a graph is *homophilic*, we expect a smoothing or LFD dynamics enhancing the low-frequency components to be successful for node classification (Wu et al., 2019; Klicpera et al., 2019). In the opposite case of *heterophily*, the high-frequency components might contain more relevant information for separating classes (Bo et al., 2021) – the classical example being the eigenvector of $\mathbf{\Delta}$ with largest frequency $\rho_\mathbf{\Delta}$ separating a bipartite graph. Accordingly, an ideal framework for learning on graphs must at least accommodate both of these opposite scenarios by being able to induce either an LFD or a HFD dynamics.

We can now investigate the gradient flow equations of the energy in Equation (10).

**Theorem 3.2** (Informal). *The continuous gradient flow in Equation* (36) *can learn to be either* LFD *(mostly edge-wise attractive) or* HFD *(mostly edge-wise repulsive) depending on the spectrum of* $\mathbf{W}$.

(A precise result, along with convergence rates, is stated as Theorem B.3 in the Appendix). Informally, Theorem 3.2 shows that the gradient flow in Equation (36) is expressive enough to induce repulsion along edges if needed — as expected based on the decomposition of $\mathcal{E}_\theta$ in Equation (10). As argued above, a dynamical system that can never be HFD might instead struggle on heterophilic graphs where the feature signal needs to be sharpened rather than smoothed out. The property that the gradient flow *can* be HFD is non-trivial and in fact some continuous-time GNNs — such as those introduced in Xhonneux et al. (2020); Chamberlain et al. (2021a); Eliasof et al. (2021) — are never HFD and as a result suffer on heterophilic datasets (as confirmed in our experiments in Table 1):

**Theorem 3.3** (Informal). *Models like* CGNN, GRAND *and* $\mathrm{PDE-GCN}_D$ *are never* HFD.

We refer to Theorem B.4 for a statement including convergence rates and over-smoothing results.

**Message of Section 3:** We introduced an energy $\mathcal{E}_\theta$ allowing to learn attractive/repulsive forces along edges via the spectrum of the channel-mixing inducing an LFD/HFD dynamics as per Theorem 3.2. We argue that *energies* rather than *evolution equations* should be the object to parameterise for deriving more principled GNNs that are easier to interpret and analyse. This is studied next.

# 4 FROM ENERGY TO EVOLUTION EQUATIONS: GNNS AS GRADIENT FLOWS

In order to connect our theory to practice, we discretize Equation (36) as in Equation (4), replacing continuous time by fixed steps corresponding to GNN layers.

## 4.1 DISCRETE GRADIENT FLOWS AND SPECTRAL ANALYSIS

As in Equation (4), we use the Euler scheme with step size $\tau$ to solve Equation (36). In our framework we parameterise the energy rather than the equations, which leads to *symmetric* channel-mixing matrices $\mathbf{\Omega}, \mathbf{W} \in \mathbb{R}^{d \times d}$. The use of the explicit Euler discretization yields a residual architecture:

$$\mathbf{F}(t + \tau) = \mathbf{F}(t) + \tau \left( -\mathbf{F}(t)\mathbf{\Omega} + \bar{\mathbf{A}}\mathbf{F}(t)\mathbf{W} - \mathbf{F}(0)\tilde{\mathbf{W}} \right), \quad \mathbf{F}(0) = \psi_{\text{EN}}(\mathbf{F}_0), \qquad (11)$$

where an *encoder* $\psi_{\text{EN}} : \mathbb{R}^{n \times p} \to \mathbb{R}^{n \times d}$ processes input features $\mathbf{F}_0$ and the prediction $\psi_{\text{DE}}(\mathbf{F}(T))$ is produced by a *decoder* $\psi_{\text{DE}} : \mathbb{R}^{n \times d} \to \mathbb{R}^{n \times k}$. Here, $k$ is the number of label classes, $T = m\tau$ is the *integration time*, and $m$ is the number of *layers*. We note that (i) non-linear activations can be included in $\psi_{\text{EN}}, \psi_{\text{DE}}$ making the entire model non-linear; (ii) since the framework is residual, even if the message-passing is linear, this is *not* equivalent to collapsing the dynamics into a single layer with diffusion matrix $\bar{\mathbf{A}}^m$ as done in Wu et al. (2019) — see Equation (37) in the Appendix.

**Interaction between the graph and channel-mixing spectra.** We restrict our theoretical analysis to the gradient flows in Equation (11) where we remove dampening and source term effects (i.e., $\mathbf{\Omega} = \tilde{\mathbf{W}} = \mathbf{0}$, which corresponds to a residual GCN). Our technique consists in vectorizing the solution $\mathbf{F}(t) \mapsto \text{vec}(\mathbf{F}(t))$ and rewriting the update as $\text{vec}(\mathbf{F}(t + \tau)) = \text{vec}(\mathbf{F}(t)) + \tau \left( \mathbf{W} \otimes \bar{\mathbf{A}} \right) \text{vec}(\mathbf{F}(t))$ (see Appendix A.2 for details). In particular, once we choose bases $\{\phi_r^{\mathbf{W}}\}$ and $\{\phi_\ell^{\mathbf{\Delta}}\}$ of orthonormal eigenvectors for $\mathbf{W}$ and $\mathbf{\Delta}$ respectively, *we can write the solution after $m$ layers explicitly*:

$$\text{vec}(\mathbf{F}(m\tau)) = \sum_{r=1}^{d} \sum_{\ell=0}^{n-1} \left( 1 + \tau\lambda_r^{\mathbf{W}}(1 - \lambda_\ell^{\mathbf{\Delta}}) \right)^m c_{r,\ell}(0)\phi_r^{\mathbf{W}} \otimes \phi_\ell^{\mathbf{\Delta}}, \qquad (12)$$

where $c_{r,\ell}(0) := \langle \text{vec}(\mathbf{F}(0)), \phi_r^{\mathbf{W}} \otimes \phi_\ell^{\mathbf{\Delta}} \rangle$. We see that the interaction of the spectra $\{\lambda_r^{\mathbf{W}}\}$ and $\{\lambda_\ell^{\mathbf{\Delta}}\}$ is the 'driving' factor for the dynamics, with positive (negative) eigenvalues of $\mathbf{W}$ magnifying the frequencies $\lambda_\ell^{\mathbf{\Delta}} < 1$ ($> 1$ respectively). In the following we let $\lambda_\pm^{\mathbf{W}}$ denote the most positive/negative eigenvalue of $\mathbf{W}$ with associated eigenvectors $\phi_\pm^{\mathbf{W}}$.[1] Note that $\phi_{n-1}^{\mathbf{\Delta}}$ is the Laplacian eigenvector associated with largest frequency $\rho_{\mathbf{\Delta}}$. We now consider the following:

$$\lambda_+^{\mathbf{W}}(\rho_{\mathbf{\Delta}} - 1))^{-1} < |\lambda_-^{\mathbf{W}}| < 2(\tau(2 - \rho_{\mathbf{\Delta}}))^{-1}. \qquad (13)$$

The first inequality means that the negative eigenvalues of $\mathbf{W}$ dominate the positive ones (once we factor in the graph spectrum contribution), while the second is a constraint on the step-size since if $\tau$ is too large, then we no longer approximate the gradient flow in Equation (36).

**Theorem 4.1.** *Let $m$ be the number of layers. Consider $\mathbf{F}(t + \tau) = \mathbf{F}(t) + \tau\bar{\mathbf{A}}\mathbf{F}(t)\mathbf{W}$, with symmetric $\mathbf{W}$. If Equation (13) holds, then there exists $\delta < 1$ s.t. for all $i \in \mathsf{V}$ we have:*

$$\mathbf{f}_i(m\tau) = \left( 1 + \tau|\lambda_-^{\mathbf{W}}|(\rho_{\mathbf{\Delta}} - 1) \right)^m \left( c_{-,n-1}(0)\, \phi_{n-1}^{\mathbf{\Delta}}(i) \cdot \phi_-^{\mathbf{W}} + \mathcal{O}\left( \delta^m \right) \right). \qquad (14)$$

*Conversely, if $\lambda_+^{\mathbf{W}}(\rho_{\mathbf{\Delta}} - 1))^{-1} > |\lambda_-^{\mathbf{W}}|$, then*

$$\mathbf{f}_i(m\tau) = \left( 1 + \lambda_+^{\mathbf{W}} \right)^m \left( c_{+,0}(0)\, \sqrt{d_i} \cdot \phi_+^{\mathbf{W}} + \mathcal{O}\left( \delta^m \right) \right). \qquad (15)$$

We report the explicit value of $\delta$ in Equation (38) in Appendix C.1. We now comment on the consequences of Theorem 4.1. Equation (14) implies that if the negative eigenvalues of $\mathbf{W}$ are *sufficiently larger* than the positive ones (in absolute value, as per Equation (13)), then repulsive forces and hence high frequencies dominate. Indeed for $i \in \mathsf{V}$ we have $\mathbf{f}_i(m\tau) \sim \phi_{n-1}^{\mathbf{\Delta}}(i) \cdot \phi_-^{\mathbf{W}}$ at the fastest scale, up to *lower order terms in the number of layers*. Thus as we increase the depth, any feature representation $\mathbf{f}_i(m\tau)$ becomes dominated by a multiple of $\phi_-^{\mathbf{W}} \in \mathbb{R}^d$ only depending on the value taken by the Laplacian eigenvector $\phi_{n-1}^{\mathbf{\Delta}}$ at node $i$. On the other hand, if Equation (15) holds, then at the largest scale we have $\mathbf{f}_i(m\tau) \sim \sqrt{d_i} \cdot \phi_+^{\mathbf{W}} \in \mathbb{R}^d$, meaning that the node representation becomes dominated by a multiple of $\phi_+^{\mathbf{W}}$ only depending on the degree of $i$ — which recovers the over-smoothing phenomenon (Nt & Maehara, 2019; Oono & Suzuki, 2020).

---

[1]Our arguments extend trivially to the degenerate case.

**Corollary 4.2.** *If Equation* (14) *holds, then the system is* HFD *for a.e.* $\mathbf{F}(0)$ *and* $\mathbf{F}(m\tau)/||\mathbf{F}(m\tau)|| \to \mathbf{F}_\infty$ *s.t.* $\Delta\mathbf{f}^r_\infty = \rho_\Delta \mathbf{f}^r_\infty$ *for each* $r$. *Conversely, if Equation* (15) *holds, then the system is* LFD *for a.e.* $\mathbf{F}(0)$ *and* $\mathbf{F}(m\tau)/||\mathbf{F}(m\tau)|| \to \mathbf{F}_\infty$ *s.t.* $\Delta\mathbf{f}^r_\infty = \mathbf{0}$ *for each* $r$.

**Remark.** In general neither the highest nor the lowest frequency Laplacian eigenvectors constitute ideal classifiers and in fact we always have a finite depth so that $\mathbf{F}(m\tau)$ also depends on the lower-order terms of the asymptotic expansion in Theorem 4.1. Whether the dynamics is LFD or HFD will affect if the lower or higher frequencies have a larger contribution to the prediction; indeed, *we can compute the 'impact' of each graph frequency explicitly* thanks to Equation (12).

### 4.2 CONNECTIONS TO EXISTING RESULTS

**Residual connection.** The following result shows that the residual connection is crucial:

**Theorem 4.3.** *If* G *is not bipartite, and we remove the residual connection, i.e.* $\mathbf{F}(t+\tau) = \tau\bar{\mathbf{A}}\mathbf{F}(t)\mathbf{W}$, *with* $\mathbf{W}$ *symmetric, then the dynamics is* LFD *for a.e.* $\mathbf{F}(0)$ *independent of the spectrum of* $\mathbf{W}$.

Differently from previous over-smoothing results of Oono & Suzuki (2020); Cai & Wang (2020), here we have no constraints on the spectral radius of $\mathbf{W}$ coming from the graph topology. In other words, the residual connection *fully enables the channel-mixing to steer the evolution* towards low or high frequencies depending on the task. If we drop the residual connection, $\mathbf{W}$ is less powerful; this is also confirmed by our ablation studies (see Figure 3 in the Appendix).

**Negative eigenvalues flip the edge signs.** Let $\mathbf{W} = \mathbf{\Phi}^{\mathbf{W}}\mathbf{\Lambda}^{\mathbf{W}}(\mathbf{\Phi}^{\mathbf{W}})^\top$ be the eigendecomposition of $\mathbf{W}$ yielding the Fourier coefficients $\mathbf{Z}(t) = \mathbf{F}(t)\mathbf{\Phi}^{\mathbf{W}}$. We rewrite the discretized gradient flow $\mathbf{F}(t+\tau) = \mathbf{F}(t) + \tau\bar{\mathbf{A}}\mathbf{F}(t)\mathbf{W}$ in the Fourier domain of $\mathbf{W}$ as $\mathbf{Z}(t+\tau) = \mathbf{Z}(t) + \tau\bar{\mathbf{A}}\mathbf{Z}(t)\mathbf{\Lambda}^{\mathbf{W}}$ and note that along the eigenvectors of $\mathbf{W}$, if $\lambda_r^{\mathbf{W}} < 0$ then the dynamics is *equivalent* to flipping the sign of the edges. This shows that negative edge weight mechanisms proposed in Li et al. (2020); Bo et al. (2021); Yan et al. (2021) for heterophilic graphs can be achieved with a simple GCN model where the channel-mixing matrix $\mathbf{W}$ has negative eigenvalues. We refer to Equation (39) in the appendix for a thorough discussion and derivation.

**The message of Section 4:** *Discrete gradient flows of* $\mathcal{E}_\theta$ *are equivalent to linear graph convolutions with symmetric weights shared across layers.* This provides a 'multi-particle' interpretation for graph convolutions and sheds light onto the dynamics they generate. We can derive simple expansions of the learned features and show that the interaction between the eigenvectors and spectra of $\mathbf{W}$ and $\mathbf{\Delta}$ is what drives the dynamics and determines its dominating effects. Convolutional GNN models can deal with heterophily if the channel mixing matrix has negative eigenvalues.

## 5 FROM EVOLUTION EQUATIONS TO ENERGY: INTERPRETING GNNS VIA $\mathcal{E}_\theta$

Analysing energies along GNNs is one approach to investigate their dynamics. Cai & Wang (2020); Bodnar et al. (2022) showed that $\mathcal{E}^{\mathrm{Dir}}$ is decreasing (exponentially) along some classes of graph convolutions, implying *over-smoothing* – see also Rusch et al. (2022). In this Section, we start from time-continuous graph convolutions as in Equation (5), with $\sigma$ acting elementwise:

$$\dot{\mathbf{F}}(t) = \sigma\left(-\mathbf{F}(t)\mathbf{\Omega} + \bar{\mathbf{A}}\mathbf{F}(t)\mathbf{W} - \mathbf{F}(0)\tilde{\mathbf{W}}\right) \quad (16)$$

Although this is no longer necessarily a gradient flow due to $\sigma$, we prove that if the weights are symmetric, then $\mathcal{E}_\theta$ in Equation (35) still decreases along Equation (16).

**Theorem 5.1.** *Consider* $\sigma : \mathbb{R} \to \mathbb{R}$ *satisfying* $x \mapsto x\sigma(x) \geq 0$. *If* $\mathbf{F}$ *solves Equation* (16) *with* $\mathbf{\Omega}, \mathbf{W}$ *symmetric, then* $t \mapsto \mathcal{E}_\theta(\mathbf{F}(t))$ *is decreasing. If we discretize the system using the Euler method with step size* $\tau$ *and* $C_+$ *denotes the most positive eigenvalue of* $\mathbf{\Omega} \otimes \mathbf{I}_n - \mathbf{W} \otimes \bar{\mathbf{A}}$, *then*

$$\mathcal{E}_\theta(\mathbf{F}(t+\tau)) - \mathcal{E}_\theta(\mathbf{F}(t)) \leq C_+ \cdot ||\mathbf{F}(t+\tau) - \mathbf{F}(t)||^2.$$

An important consequence of Theorem 5.1 is that for non-linear graph convolutions with symmetric weights, the physical interpretation is preserved since the same multi-particle energy $\mathcal{E}_\theta$ in Equation (10) is decreasing along the solution. Note that in the discrete case the energy monotonicity can be interpreted as a Lipschitz regularity result. Namely, the channel-mixing $\mathbf{W}$ still induces attraction/repulsion along edges via its positive/negative eigenvalues (more explicitly, see Lemma

D.1). We again emphasize that the requirement of symmetric weights is not restrictive thanks to the universal approximation results of Hu et al. (2019). Theorem 5.1 differs from Cai & Wang (2020); Bodnar et al. (2022) in two ways: (i) It asserts monotonicity of an energy $\mathcal{E}_\theta$ more general than $\mathcal{E}^{\mathrm{Dir}}$, since it is parametric and in fact also able to enhance the high frequencies; and (ii) it holds for an infinite class of non-linear activations (beyond $\mathrm{ReLU}$). So far we have considered *time-independent* energies. We can generalize our discussion to energies of the form $\mathcal{E}_\theta(\cdot, t)$ whose potentials now vary in time. The equations then take the form of Equation (5) with $\mathbf{\Omega}_t$ and $\mathbf{W}_t$ symmetric.

**The message of Section 5**: We show that graph convolutions with symmetric weights identify curves along which the multi-particle energy $\mathcal{E}_\theta$ decreases hence acting as 'approximate' gradient flows. Despite the non-linear activation of $\sigma$ we can still interpret the *learning dynamics of convolution on graphs as finding the 'right' edge-wise attractive/repulsive potentials through channel mixing*.

## 6 EXPERIMENTS

In Theorem 4.1 we have shown that the subset of linear graph convolutions in Equation (5) with shared, symmetric weights is characterized by the strong inductive bias that the multi-particle energy $\mathcal{E}_\theta$ in Equation (10) is being minimized along the equations. Although not a gradient flow, even when we activate the equations with $\sigma$ as in Theorem 5.1, we can preserve such inductive bias since the same energy is decreasing. This means that both frameworks can provably induce attraction or repulsion along edges thanks to the spectrum of the channel-mixing. We validate our theoretical analysis by testing that these principled (and more efficient) classes of convolutional models along which $\mathcal{E}_\theta$ decreases can compete with baselines designed to target heterophilic graphs.

**The model and the parameterisation.** In the following we evaluate a subclass of gradient flows in Equation (11) giving rise to a framework termed $\mathrm{GRAFF}$ (Gradient Flow Framework):

$$\mathrm{GRAFF}: \quad \mathbf{F}(t+\tau) = \mathbf{F}(t) + \tau\left(-\mathbf{F}(t)\mathrm{diag}(\boldsymbol{\omega}) + \bar{\mathbf{A}}\mathbf{F}(t)\mathbf{W} - \beta\mathbf{F}(0)\right), \qquad (17)$$

where $\boldsymbol{\omega} \in \mathbb{R}^d$ and $\beta \in \mathbb{R}$, $\mathbf{W}$ is a *symmetric* $d \times d$-matrix *shared* across layers and (node-wise) encoder and decoder are MLPs. We consider two possible implementations for $\mathbf{W}$: (i) diagonally-dominant (see Appendix E), where we learn an off-diagonal symmetric matrix and the diagonal terms separately, and (ii) the case with $\mathbf{W}$ diagonal and we report best numbers over these two configurations. We note that both these parameterisations allow the model to control the spectrum of $\mathbf{W}$ more easily which we know to be essential from Theorem 4.1; we refer to the methodology description in Appendix E for further details. By Theorem 5.1, if we 'activate' Equation (17) as

$$\mathrm{GRAFF}_{\mathrm{NL}}: \quad \mathbf{F}(t+\tau) = \mathbf{F}(t) + \tau\sigma\left(-\mathbf{F}(t)\mathrm{diag}(\boldsymbol{\omega}) + \bar{\mathbf{A}}\mathbf{F}(t)\mathbf{W} - \beta\mathbf{F}(0)\right), \qquad (18)$$

with $\sigma$ s.t. $x\sigma(x) \geq 0$, then $\mathcal{E}_\theta$ in Equation (35) is decreasing, so that we can think of such equations as more general 'approximate' gradient flows termed $\mathrm{GRAFF}_{\mathrm{NL}}$ (where NL stands for non-linear).

**Complexity.** $\mathrm{GRAFF}$ scales as $\mathcal{O}(|\mathsf{V}|pd + m|\mathsf{E}|d)$, where $p$ and $d$ are input feature and hidden dimension respectively, with $p \geq d$ usually, and $m$ is the number of layers. Note that GCN has complexity $\mathcal{O}(m|\mathsf{E}|(p+d))$ and in fact *our model is slightly faster than GCN* mainly due to the preliminary encoding performed node-wise rather than edge-wise; main baselines on heterophilic graphs like GGCN and Sheaf learn edge-wise weights based on features which is slower (as confirmed in Figure 5 in Appendix E). Moreover, for $\mathrm{GRAFF}$ the number of parameters scales as $\mathcal{O}(pd + d^2)$ while for other baselines they scale with the number of layers at least as $\mathcal{O}(pd + md^2)$.

**Synthetic experiments.** To investigate our claims we first use the synthetic Cora dataset of (Zhu et al., 2020, Appendix G) where graphs are generated for target levels of homophily see Appendix E.3. Figure 2 reports the test accuracy vs true label homophily. For *Neg-prod* we set $\mathbf{W} = -\mathbf{W}_0\mathbf{W}_0^\top$ so to only have non-positive eigenvalues: we see that this is better than the opposite case of *prod* ($\mathbf{W} = \mathbf{W}_0\mathbf{W}_0^\top$) on low-homophily (and viceversa on high-homophily). This confirms Theorem 4.1 where we have shown that the gradient flow can be HFD – that is generally desirable with low-homophily – through the negative eigenvalues of $\mathbf{W}$. In practice 'non-signed' variants

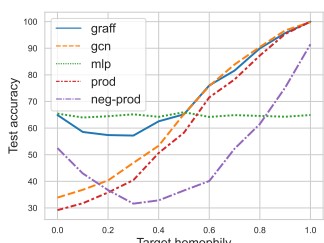

Figure 2: Synthetic experiments with controlled homophily.

| | Texas | Wisconsin | Cornell | Film | Squirrel | Chameleon | Citeseer | Pubmed | Cora |
|---|---|---|---|---|---|---|---|---|---|
| Hom level | **0.11** | **0.21** | **0.30** | **0.22** | **0.22** | **0.23** | **0.74** | **0.80** | **0.81** |
| #Nodes | 183 | 251 | 183 | 7,600 | 5,201 | 2,277 | 3,327 | 18,717 | 2,708 |
| #Edges | 295 | 466 | 280 | 26,752 | 198,493 | 31,421 | 4,676 | 44,327 | 5,278 |
| #Classes | 5 | 5 | 5 | 5 | 5 | 5 | 7 | 3 | 6 |
| GGCN | $84.86 \pm 4.55$ | $86.86 \pm 3.29$ | $85.68 \pm 6.63$ | $37.54 \pm 1.56$ | $55.17 \pm 1.58$ | $71.14 \pm 1.84$ | $77.14 \pm 1.45$ | $89.15 \pm 0.37$ | $87.95 \pm 1.05$ |
| GPRGNN | $78.38 \pm 4.36$ | $82.94 \pm 4.21$ | $80.27 \pm 8.11$ | $34.63 \pm 1.22$ | $31.61 \pm 1.24$ | $46.58 \pm 1.71$ | $77.13 \pm 1.67$ | $87.54 \pm 0.38$ | $87.95 \pm 1.18$ |
| H2GCN | $84.86 \pm 7.23$ | $87.65 \pm 4.98$ | $82.70 \pm 5.28$ | $35.70 \pm 1.00$ | $36.48 \pm 1.86$ | $60.11 \pm 2.15$ | $77.11 \pm 1.57$ | $89.49 \pm 0.38$ | $87.87 \pm 1.20$ |
| GCNII | $77.57 \pm 3.83$ | $80.39 \pm 3.40$ | $77.86 \pm 3.79$ | $37.44 \pm 1.30$ | $38.47 \pm 1.58$ | $63.86 \pm 3.04$ | $77.33 \pm 1.48$ | $90.15 \pm 0.43$ | $88.37 \pm 1.25$ |
| Geom-GCN | $66.76 \pm 2.72$ | $64.51 \pm 3.66$ | $60.54 \pm 3.67$ | $31.59 \pm 1.15$ | $38.15 \pm 0.92$ | $60.00 \pm 2.81$ | $78.02 \pm 1.15$ | $89.95 \pm 0.47$ | $85.35 \pm 1.57$ |
| PairNorm | $60.27 \pm 4.34$ | $48.43 \pm 6.14$ | $58.92 \pm 3.15$ | $27.40 \pm 1.24$ | $50.44 \pm 2.04$ | $62.74 \pm 2.82$ | $73.59 \pm 1.47$ | $87.53 \pm 0.44$ | $85.79 \pm 1.01$ |
| GraphSAGE | $82.43 \pm 6.14$ | $81.18 \pm 5.56$ | $75.95 \pm 5.01$ | $34.23 \pm 0.99$ | $41.61 \pm 0.74$ | $58.73 \pm 1.68$ | $76.04 \pm 1.30$ | $88.45 \pm 0.50$ | $86.90 \pm 1.04$ |
| GCN | $55.14 \pm 5.16$ | $51.76 \pm 3.06$ | $60.54 \pm 5.30$ | $27.32 \pm 1.10$ | $53.43 \pm 2.01$ | $64.82 \pm 2.24$ | $76.50 \pm 1.36$ | $88.42 \pm 0.50$ | $86.98 \pm 1.27$ |
| GAT | $52.16 \pm 6.63$ | $49.41 \pm 4.09$ | $61.89 \pm 5.05$ | $27.44 \pm 0.89$ | $40.72 \pm 1.55$ | $60.26 \pm 2.50$ | $76.55 \pm 1.23$ | $87.30 \pm 1.10$ | $86.33 \pm 0.48$ |
| MLP | $80.81 \pm 4.75$ | $85.29 \pm 3.31$ | $81.89 \pm 6.40$ | $36.53 \pm 0.70$ | $28.77 \pm 1.56$ | $46.21 \pm 2.99$ | $74.02 \pm 1.90$ | $75.69 \pm 2.00$ | $87.16 \pm 0.37$ |
| CGNN | $71.35 \pm 4.05$ | $74.31 \pm 7.26$ | $66.22 \pm 7.69$ | $35.95 \pm 0.86$ | $29.24 \pm 1.09$ | $46.89 \pm 1.66$ | $76.91 \pm 1.81$ | $87.70 \pm 0.49$ | $87.10 \pm 1.35$ |
| GRAND | $75.68 \pm 7.25$ | $79.41 \pm 3.64$ | $82.16 \pm 7.09$ | $35.62 \pm 1.01$ | $40.05 \pm 1.50$ | $54.67 \pm 2.54$ | $76.46 \pm 1.77$ | $89.02 \pm 0.51$ | $87.36 \pm 0.96$ |
| Sheaf (max) | $85.95 \pm 5.51$ | $89.41 \pm 4.74$ | $84.86 \pm 4.71$ | $37.81 \pm 1.15$ | $56.34 \pm 1.32$ | $68.04 \pm 1.58$ | $76.70 \pm 1.57$ | $89.49 \pm 0.40$ | $86.90 \pm 1.13$ |
| GRAFF | $88.38 \pm 4.53$ | $88.83 \pm 3.29$ | $84.05 \pm 6.10$ | $37.11 \pm 1.08$ | $58.72 \pm 0.84$ | $71.08 \pm 1.75$ | $77.30 \pm 1.85$ | $90.04 \pm 0.41$ | $88.01 \pm 1.03$ |
| $\text{GRAFF}_{\text{NL}}$ | $86.49 \pm 4.84$ | $87.26 \pm 2.52$ | $77.30 \pm 3.24$ | $35.96 \pm 0.95$ | $59.01 \pm 1.31$ | $71.38 \pm 1.47$ | $76.81 \pm 1.12$ | $89.81 \pm 0.50$ | $87.81 \pm 1.13$ |

Table 1: Node-classification results. Top three models are coloured by First, Second, Third

like GRAFF are more flexible and outperform GCN with low homophily, confirming Theorem 4.3 where we have shown that without a residual connection convolutional models are LFD irrespectively of the spectrum of $\mathbf{W}$ – further results in Figure 4 in Appendix E.

**Real world experiments.** In Table 1 we test GRAFF and $\text{GRAFF}_{\text{NL}}$ on datasets with varying homophily (Sen et al., 2008; Rozemberczki et al., 2021; Pei et al., 2020) (details in Appendix E.4). We use results provided in (Yan et al., 2021, Table 1), which include GCNs models, GAT (Veličković et al., 2018), PairNorm (Zhao & Akoglu, 2019) and models designed for heterophily (GGCN (Yan et al., 2021), Geom-GCN (Pei et al., 2020), H2GCN (Zhu et al., 2020) and GPRGNN (Chien et al., 2021)). For Sheaf (Bodnar et al., 2022), a recent strong baseline with heterophily, we took the best performing variant (out of six) for each dataset. We include continuous baselines CGNN (Xhonneux et al., 2020) and GRAND (Chamberlain et al., 2021a) to corroborate Theorem 3.3. Training, validation and test splits are taken from Pei et al. (2020) for all datasets for comparison. We also evaluate GRAFF on larger heterophilic datasets discussed in Lim et al. (2021) (see Appendix E.6) and we compare with further recent baselines in Appendix E.7.

**Results.** GRAFF and $\text{GRAFF}_{\text{NL}}$ are both *versions of graph convolutions with stronger 'inductive bias' given by the energy $\mathcal{E}_\theta$ decreasing along the solution*; in fact, we can recover them from graph convolutions by simply requiring that the channel-mixing is *symmetric and shared across layers*. Nonetheless they achieve competitive results on all datasets often outperforming slower and more complex models. They are extremely competitive on more homophilic datasets as well, in contrast with the performance of models like Sheaf mainly designed to handle heterophily.

# 7 CONCLUSIONS

We argued that when studying and developing GNNs we should focus on energy functionals rather than the evolution equations. We introduced a new framework for GNNs where the evolution is a gradient flow of a multi-particle learnable energy. This gives rise to principled graph convolutions where the channel-mixing is a symmetric matrix and induces attraction (repulsion) along edges via its positive (negative) eigenvalues. We explored the theoretical implications by investigating the dominating terms in the learned feature expansion and corroborated that this graph convolutional framework can perform well in heterophilic settings. We proved that existing (generalized) graph convolutions maintain the dynamics induced by the same class of multi-particle energies if the channel-mixing is symmetric even when they are not strictly gradient flows due to non-linear activations; this provides a deeper connection between energy functionals and GNNs and extends several recent results that have monitored the classical Dirichlet energy along GCNs to shed light on their dynamics.

**Limitations and future works.** We limited our attention to a class of energy functionals whose gradient flows give rise to evolution equations of the generalized graph convolution type. In future work, we plan to study other families of energies that generalize different GNN architectures and provide new models that are more 'physics'-inspired. We will also investigate time-dependent energy functionals and how to generalize our results to this setting. To the best of our knowledge, our analysis is a first step into studying the interaction of the graph and 'channel-mixing' spectra. In future work, we will explore other more general dynamics (i.e., that are neither LFD nor HFD).

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

OVERVIEW OF THE APPENDIX

To facilitate navigating the appendix, where we report several additional theoretical results, analysis of different cases along with further experiments and ablation studies, we provide the following detailed outline.

- In Appendix A.1 we review properties of the classical Dirichlet energy on manifolds that inspired traditional PDE variational methods for image processing whose extension to graphs and GNNs more specifically partly constitutes one of the main motivations of our work. We also review important elementary properties of the Kronecker product of matrices that are used throughout our proofs in Appendix A.2. We also comment on the choice of the normalization (and symmetrization) of the graph Laplacian, briefly mentioning the impact of different choices.

- In Appendix B.1 we derive the energy decomposition reported in Equation (10). In Appendix B.2 we derive additional rigorous results to justify our characterization of LFD and HFD dynamics in Definition 3.1 along with explicit examples. We also formalize more explicitly and quantitatively Theorem 3.2 in Theorem B.3. In Appendix B.3 we report a more explicit statement with convergence rates and over-smoothing results which covers the informal version in Theorem 3.3. In Appendix B.4 we explore the special case of $\Omega = \mathbf{W}$ which is equivalent to choosing $\Delta$ rather than $\bar{\mathbf{A}}$ as message-passing matrix *providing new arguments as to why propagating messages using $\bar{\mathbf{A}}$ rather than the graph Laplacian is actually 'more robust'*. Finally in Appendix B.5 we formally derive an analogy between the continuous energy used for manifolds (images) and a subset of the parametric energies in Equation (6).

- In Appendix C we prove the main results of Section 4, namely Theorem 4.1, Corollary 4.2, and Theorem 4.3.

- In Appendix D we prove Theorem 5.1 and an extra result confirming that even in the non-linear case the channel-mixing $\mathbf{W}$ still induces attraction and repulsion via its spectrum hence magnifying the low or high frequencies respectively.

- In Appendix E we report additional details on hyperparameter tuning, datasets adopted, further synthetic and ablation studies, along with extra experiments on larger heterophilic datasets in Appendix E.6.

**Additional notations and conventions used throughout the appendix.** Any graph G is taken to be *connected*. We order the eigenvalues of the graph Laplacian as $0 = \lambda_0^{\Delta} \leq \lambda_1^{\Delta} \leq \ldots \leq \lambda_{n-1}^{\Delta} = \rho_{\Delta} \leq 2$ with associated orthonormal basis of eigenvectors $\{\phi_\ell^{\Delta}\}_{\ell=0}^{n-1}$ so that in particular we have $\Delta \phi_0^{\Delta} = \mathbf{0}$. Moreover, given a symmetric matrix $\mathbf{B}$, we generally denote the spectrum of $\mathbf{B}$ by $\mathrm{spec}(\mathbf{B})$ and if $\mathbf{B} \succeq 0$, then $\mathrm{gap}(\mathbf{B})$ denotes the *positive smallest eigenvalue* of $\mathbf{B}$. Finally, if we write $\mathbf{F}(t)/\|\mathbf{F}(t)\|$ we always take the norm to be the Frobenius one and tacitly assume that the dynamics is s.t. the solution is not zero.

## A  PROOFS AND ADDITIONAL DETAILS OF SECTION 2

### A.1  DISCUSSION ON CONTINUOUS DIRICHLET ENERGY AND HARMONIC MAPS

**Starting point: a geometric parallelism.** To motivate a gradient-flow approach for GNNs, we start from the continuous case (see Appendix A.1 for details). Consider a smooth map $f : \mathbb{R}^n \to (\mathbb{R}^d, h)$ with $h$ a constant metric represented by $\mathbf{H} \succeq 0$. The *Dirichlet energy* of $f$ is defined by

$$\mathcal{E}(f, h) = \frac{1}{2} \int_{\mathbb{R}^n} \|\nabla f\|_h^2 \, dx = \frac{1}{2} \sum_{q,r=1}^{d} \sum_{j=1}^{n} \int_{\mathbb{R}^n} h_{qr} \partial_j f^q \partial_j f^r(x) dx \qquad (19)$$

and measures the 'smoothness' of $f$. A natural approach to find minimizers of $\mathcal{E}$ - called *harmonic maps* - was introduced in Eells & Sampson (1964) and consists in studying the **gradient flow** of $\mathcal{E}$, wherein a given map $f(0) = f_0$ is evolved according to $\dot{f}(t) = -\nabla_f \mathcal{E}(f(t))$. These type of evolution equations have historically been the core of *variational* and *PDE-based image processing*;

in particular, gradient flows of the Dirichlet energy were shown Kimmel et al. (1997) to recover the Perona-Malik nonlinear diffusion Perona & Malik (1990).

In this subsection we briefly expand on the formulation of continuous Dirichlet energy in Section 2 to provide more context. Consider a smooth map $f : (M, g) \to (N, h)$, where $N$ is usually a larger manifold we embed $M$ into, and $g, h$ are Riemannian metrics on domain and codomain respectively. The *Dirichlet energy* of $f$ is defined by

$$\mathcal{E}(f, g, h) := \frac{1}{2} \int_M |df|_g^2 d\mu(g),$$

with $|df|_g$ the norm of the Jacobian of $f$ measured with respect to $g$ and $h$. If $(M, g)$ is standard Euclidean space $\mathbb{R}^n$, $N = \mathbb{R}^d$ and $h$ is a constant positive semi-definite matrix, then we can rewrite the Dirichlet energy in a more familiar form as

$$\mathcal{E}(f, h) = \frac{1}{2} \int_{\mathbb{R}^n} \text{trace} \left( Df^\top h Df \right) d\mu = \frac{1}{2} \sum_{q,r=1}^d \sum_{j=1}^n \int_{\mathbb{R}^n} h_{qr} \partial_j f^q \partial_j f^r(x) dx.$$

The Dirichlet energy measures the smoothness of the map $f$, and indeed if $h$ is the identity in $\mathbb{R}^d$, then we recover the classical definition

$$\mathcal{E}(f) = \frac{1}{2} \sum_{r=1}^d \int_{\mathbb{R}^n} ||\nabla f^r||^2(x) dx.$$

**Gradient flow of Dirichlet energy.**   Minimizers of $\mathcal{E}$ - referred to as *harmonic maps* - are important objects in geometry: to mention a few, geodesics, minimal isometric immersions and maps $f : M \to \mathbb{R}^d$ solving $\Delta_g f = 0$ are all instances of harmonic maps. To identify such critical points, one computes the first variation of the energy $\mathcal{E}$ along an arbitrary direction $\partial_t f$, which can be written as

$$d\mathcal{E}_f(\partial_t f) = - \int_M \langle \tau_g(f), \partial_t f \rangle_h d\mu(g).$$

for some tensor field $\tau$ with explicit form

$$(\tau_{g_M}(f))^\alpha := \Delta_{g_M} f^\alpha + {}^{h_N}\Gamma^\alpha_{\beta\gamma} \partial_i f^\beta \partial_j f^\gamma g_M^{ij},$$

for $1 \leq \alpha \leq \dim(N)$, with $\{y^\alpha\}$ local coordinates on $N$ and $\Gamma^\alpha_{\beta\gamma}$ Christoffel symbols. It follows that harmonic maps are identified by the condition $\tau_g(f)) = 0$. In Eells & Sampson (1964), the pivotal idea of harmonic map flow – which has shaped much of modern research in geometric analysis – was introduced for the first time: in order to identify minimizers of $\mathcal{E}$, an input map $f_0$ is evolved along the direction of (minus) the gradient of the energy $\mathcal{E}$ leading to the dynamics

$$\partial_t f = \tau_g(f). \tag{20}$$

As a special case, when the target space is the classical Euclidean space one recovers the *heat equation* induced by the input Riemannian structure. We also note that when $(M, g)$ is a surface representing an image and $f : (u_1, u_2) \mapsto (u_1, u_2, \phi(u_1, u_2))$ with $\phi$ a color map, then Equation (20) becomes

$$\partial_t \phi = \text{div}(C_g \nabla \phi), \tag{21}$$

with $C_g$ a constant depending on the metric on $M$. If we now let $g$ to depend on $\phi$, one can recover the celebrated Perona-Malik flow Kimmel et al. (1997).

## A.2   Review of Kronecker product and properties of Laplacian kernel

**Kronecker product.**   In this subsection we summarize a few relevant notions pertaining the Kronecker product of matrices that are going to be applied throughout our spectral analysis of gradient flow equations for GNNs in both the continuous and discrete time setting.

Given a matricial equation of the form

$$\mathbf{Y} = \mathbf{AXB},$$

we can vectorize $\mathbf{X}$ and $\mathbf{Y}$ by stacking columns into $\mathrm{vec}(\mathbf{X})$ and $\mathrm{vec}(\mathbf{Y})$ respectively, and rewrite the previous system as

$$\mathrm{vec}(\mathbf{Y}) = \left(\mathbf{B}^\top \otimes \mathbf{A}\right)\mathrm{vec}(\mathbf{X}). \tag{22}$$

If $\mathbf{A}$ and $\mathbf{B}$ are symmetric with spectra $\mathrm{spec}(\mathbf{A})$ and $\mathrm{spec}(\mathbf{B})$ respectively, then the spectrum of $\mathbf{B} \otimes \mathbf{A}$ is given by $\mathrm{spec}(\mathbf{A}) \cdot \mathrm{spec}(\mathbf{B})$. Namely, if $\mathbf{Ax} = \lambda^{\mathbf{A}}\mathbf{x}$ and $\mathbf{By} = \lambda^{\mathbf{B}}\mathbf{y}$, for $\mathbf{x}$ and $\mathbf{y}$ non-zero vectors, then $\lambda^{\mathbf{B}}\lambda^{\mathbf{A}}$ is an eigenvalue of $\mathbf{B} \otimes \mathbf{A}$ with eigenvector $\mathbf{y} \otimes \mathbf{x}$:

$$\left(\mathbf{B} \otimes \mathbf{A}\right)\mathbf{y} \otimes \mathbf{x} = (\lambda^{\mathbf{B}}\lambda^{\mathbf{A}})\mathbf{y} \otimes \mathbf{x}. \tag{23}$$

One can also define the *Kronecker sum* of matrices $\mathbf{A} \in \mathbb{R}^{n \times n}$ and $\mathbf{B} \in \mathbb{R}^{d \times d}$ as

$$\mathbf{A} \oplus \mathbf{B} := \mathbf{A} \otimes \mathbf{I}_d + \mathbf{I}_n \otimes \mathbf{B}, \tag{24}$$

with spectrum $\mathrm{spec}(\mathbf{A} \oplus \mathbf{B}) = \{\lambda^{\mathbf{A}} + \lambda^{\mathbf{B}} : \ \lambda^{\mathbf{A}} \in \mathrm{spec}(\mathbf{A}), \ \lambda^{\mathbf{B}} \in \mathrm{spec}(\mathbf{B})\}$.

**Additional details on $\mathcal{E}^{\mathrm{Dir}}$ and the choice of Laplacian.** We recall that the classical graph Dirichlet energy $\mathcal{E}^{\mathrm{Dir}}$ is defined by

$$\mathcal{E}^{\mathrm{Dir}}(\mathbf{F}) = \frac{1}{2}\mathrm{trace}\left(\mathbf{F}^\top \boldsymbol{\Delta}\mathbf{F}\right),$$

where the (unusual) extra factor of $\frac{1}{2}$ is to avoid rescaling the gradient flow by 2 – which is the more common convention. We can use the Kronecker product to rewrite the Dirichlet energy as

$$\mathcal{E}^{\mathrm{Dir}}(\mathbf{F}) = \frac{1}{2}\mathrm{vec}(\mathbf{F})^\top(\mathbf{I}_d \otimes \boldsymbol{\Delta})\mathrm{vec}(\mathbf{F}), \tag{25}$$

from which we immediately derive that $\nabla_{\mathrm{vec}(\mathbf{F})}\mathcal{E}^{\mathrm{Dir}}(\mathbf{F}) = (\mathbf{I}_d \otimes \boldsymbol{\Delta})\mathrm{vec}(\mathbf{F})$ – since $\boldsymbol{\Delta}$ is *symmetric* – and hence recover the gradient flow in Equation (1) leading to the graph heat equation across each channel.

Before we further comment on the characterizations of LFD and HFD dynamics, we review the main choices of graph Laplacian and the associated harmonic signals (i.e. how we can characterize the kernel spaces of the given Laplacian operator). Recall that throughout the appendix we always assume that the underlying graph G is *connected*. The symmetrically normalized Laplacian $\boldsymbol{\Delta} = \mathbf{I} - \bar{\mathbf{A}}$ is symmetric, positive semi-definite with harmonic space of the form Chung & Graham (1997)

$$\ker(\boldsymbol{\Delta}) := \mathrm{span}(\mathbf{D}^{\frac{1}{2}}\mathbf{1}_n : \ \mathbf{1}_n = (1, \ldots, 1)^\top). \tag{26}$$

This confirms that if a given GNN evolution $\dot{\mathbf{F}}(t) = \mathrm{GNN}_\theta(\mathbf{F}(t), t)$ with initial condition $\mathbf{F}(0)$ over-smooths meaning that $\boldsymbol{\Delta}\mathbf{f}^r(t) \to \mathbf{0}$ for $t \to \infty$ for each column $1 \leq r \leq d$, then the only information persisting in the asymptotic regime is the degree and any dependence on the input features is lost, as studied in Oono & Suzuki (2020); Cai & Wang (2020). A slightly different behaviour occurs if instead of $\boldsymbol{\Delta}$, we consider the unnormalized Laplacian $\mathbf{L} = \mathbf{D} - \mathbf{A}$ with kernel $\mathrm{span}(\mathbf{1}_n)$, meaning that if $\mathbf{L}\mathbf{f}^r(t) \to \mathbf{0}$ as $t \to \infty$ for each $1 \leq r \leq d$, then any node would be embedded to a single point, hence making any separation task impossible. The same consequence applies to the random walk Laplacian $\boldsymbol{\Delta}_{\mathrm{RW}} = \mathbf{I} - \mathbf{D}^{-1}\mathbf{A}$. In particular, we note that generally a row-stochastic matrix is not symmetric – if it was, then this would in fact be doubly-stochastic – and the same applies to the random-walk Laplacian (a special exception is given by the class of *regular* graphs). In fact, in general any dynamical system governed by $\boldsymbol{\Delta}_{\mathrm{RW}}$ (or simply $\mathbf{D}^{-1}\mathbf{A}$) is not the gradient flow of an energy due to the lack of symmetry, as further confirmed below in Equation (27).

# B  PROOFS AND ADDITIONAL DETAILS OF SECTION 3

## B.1  ATTRACTION VS REPULSION: A PHYSICS-INSPIRED FRAMEWORK

We first note that the system in Equation (36) can be written using the Kronecker product as

$$\mathrm{vec}(\dot{\mathbf{F}}(t)) = -(\boldsymbol{\Omega} \otimes \mathbf{I}_n)\mathrm{vec}(\mathbf{F}(t)) + (\mathbf{W} \otimes \bar{\mathbf{A}})\mathrm{vec}(\mathbf{F}(t)) - (\tilde{\mathbf{W}} \otimes \mathbf{I}_n)\mathrm{vec}(\mathbf{F}(0)).$$

If this is the gradient flow of $\mathbf{F} \mapsto \mathcal{E}_\theta(\mathbf{F})$, then we would have

$$\nabla^2_{\mathrm{vec}(\mathbf{F})}\mathcal{E}_\theta(\mathbf{F}) = \boldsymbol{\Omega} \otimes \mathbf{I}_n - \mathbf{W} \otimes \bar{\mathbf{A}}, \tag{27}$$

which must be symmetric due to the Hessian of a function being symmetric. The latter means

$$(\mathbf{\Omega}^\top - \mathbf{\Omega}) \otimes \mathbf{I}_n = (\mathbf{W}^\top - \mathbf{W}) \otimes \bar{\mathbf{A}},$$

which is satisfied if and only if both $\mathbf{\Omega}$ and $\mathbf{W}$ are *symmetric*. This shows that Equation (36) *is the gradient flow of $\mathcal{E}_\theta$ if and only if $\mathbf{\Omega}$ and $\mathbf{W}$ are symmetric.*

We now rely on the spectral decomposition of $\mathbf{W}$ to rewrite $\mathcal{E}_\theta$ explicitly in terms of attractive and repulsive interactions. If we have a spectral decomposition $\mathbf{W} = \mathbf{\Phi^W} \mathbf{\Lambda^W} (\mathbf{\Phi^W})^\top$, we can separate the positive eigenvalues from the negative ones and write

$$\mathbf{W} = \mathbf{\Phi^W} \mathbf{\Lambda}_+ (\mathbf{\Phi^W})^\top + \mathbf{\Phi^W} \mathbf{\Lambda}_- (\mathbf{\Phi^W})^\top := \mathbf{W}_+ - \mathbf{W}_-.$$

Since $\mathbf{W}_+ \succeq 0, \mathbf{W}_- \succeq 0$, we can use the Choleski decomposition to write $\mathbf{W}_+ = \mathbf{\Theta}_+^\top \mathbf{\Theta}_+$ and $\mathbf{W}_- = \mathbf{\Theta}_-^\top \mathbf{\Theta}_-$ with $\mathbf{\Theta}_+, \mathbf{\Theta}_- \in \mathbb{R}^{d \times d}$. Equation (10) follows then by direct computation: namely

$$
\begin{aligned}
\mathcal{E}_\theta(\mathbf{F}) &= \frac{1}{2} \sum_i \langle \mathbf{f}_i, \mathbf{\Omega} \mathbf{f}_i \rangle - \frac{1}{2} \sum_{i,j} \bar{a}_{ij} \langle \mathbf{f}_i, \mathbf{W} \mathbf{f}_j \rangle \\
&= \frac{1}{2} \sum_i \langle \mathbf{f}_i, (\mathbf{\Omega} - \mathbf{W}) \mathbf{f}_i \rangle + \frac{1}{2} \sum_i \langle \mathbf{f}_i, \mathbf{W} \mathbf{f}_i \rangle - \frac{1}{2} \sum_{i,j} \bar{a}_{ij} \langle \mathbf{\Theta}_+ \mathbf{f}_i, \mathbf{\Theta}_+ \mathbf{f}_j \rangle + \frac{1}{2} \sum_{i,j} \bar{a}_{ij} \langle \mathbf{\Theta}_- \mathbf{f}_i, \mathbf{\Theta}_- \mathbf{f}_j \rangle \\
&= \frac{1}{2} \sum_i \langle \mathbf{f}_i, (\mathbf{\Omega} - \mathbf{W}) \mathbf{f}_i \rangle + \frac{1}{4} \sum_{i,j} \|\mathbf{\Theta}_+ (\nabla \mathbf{F})_{ij}\|^2 - \frac{1}{4} \sum_{i,j} \|\mathbf{\Theta}_- (\nabla \mathbf{F})_{ij}\|^2,
\end{aligned}
$$

where we have used that $\sum_{i,j} \frac{1}{d_i} \|\mathbf{\Theta}_+ \mathbf{f}_i\|^2 = \sum_i \|\mathbf{\Theta}_+ \mathbf{f}_i\|^2$.

## B.2 ADDITIONAL DETAILS ON LFD AND HFD CHARACTERIZATIONS

In this subsection we provide further details and justifications for Definition 3.1. We first prove the following simple properties.

**Lemma B.1.** *Assume we have a (continuous) process $t \mapsto \mathbf{F}(t) \in \mathbb{R}^{n \times d}$, for $t \geq 0$. The following equivalent characterizations hold:*

*(i) $\mathcal{E}^{\mathrm{Dir}}(\mathbf{F}(t)) \to 0$ for $t \to \infty$ if and only if $\mathbf{\Delta f}^r(t) \to \mathbf{0}$, for $1 \leq r \leq d$.*

*(ii) $\mathcal{E}^{\mathrm{Dir}}(\mathbf{F}(t)/\|\mathbf{F}(t)\|) \to \rho_\mathbf{\Delta}/2$ for $t \to \infty$ if and only if for any sequence $t_j \to \infty$ there exist a subsequence $t_{j_k} \to \infty$ and a unit limit $\mathbf{F}_\infty$ – depending on the subsequence – such that $\mathbf{\Delta f}_\infty^r = \rho_\mathbf{\Delta} \mathbf{f}_\infty^r$, for $1 \leq r \leq d$.*

*Proof.* (i) Given $\mathbf{F}(t) \in \mathbb{R}^{n \times d}$, we can vectorize it and decompose it in the orthonormal basis $\{\mathbf{e}_r \otimes \phi_\ell^\mathbf{\Delta} : 1 \leq r \leq d, \ 0 \leq \ell \leq n-1\}$, with $\{\mathbf{e}_r\}_{r=1}^d$ canonical basis in $\mathbb{R}^d$, and write

$$\mathrm{vec}(\mathbf{F}(t)) = \sum_{r,\ell} c_{r,\ell}(t) \mathbf{e}_r \otimes \phi_\ell^\mathbf{\Delta}, \quad c_{r,\ell}(t) := \langle \mathrm{vec}(\mathbf{F}(t)), \mathbf{e}_r \otimes \phi_\ell^\mathbf{\Delta} \rangle.$$

We can then use Equation (25) to compute the Dirichlet energy as

$$\mathcal{E}^{\mathrm{Dir}}(\mathbf{F}(t)) = \frac{1}{2} \sum_{r=1}^d \sum_{\ell=0}^{n-1} c_{r,\ell}^2(t) \lambda_\ell^\mathbf{\Delta} \equiv \frac{1}{2} \sum_{r=1}^d \sum_{\ell=1}^{n-1} c_{r,\ell}^2(t) \lambda_\ell^\mathbf{\Delta} \geq \frac{1}{2} \mathrm{gap}(\mathbf{\Delta}) \sum_{r=1}^d \sum_{\ell=1}^{n-1} c_{r,\ell}^2(t),$$

where we have used the convention above that the eigenvector $\phi_0^\mathbf{\Delta}$ is in the kernel of $\mathbf{\Delta}$. Therefore

$$\mathcal{E}^{\mathrm{Dir}}(\mathbf{F}(t)) \to 0 \quad \Longleftrightarrow \quad \sum_{r=1}^d \sum_{\ell=1}^{n-1} c_{r,\ell}^2(t) \to 0, \quad t \to \infty,$$

which occurs if and only if

$$(\mathbf{I}_d \otimes \mathbf{\Delta}) \mathrm{vec}(\mathbf{F}(t)) = \sum_{r=1}^d \sum_{\ell=1}^{n-1} c_{r,\ell}(t) \lambda_\ell^\mathbf{\Delta} \mathbf{e}_r \otimes \phi_\ell^\mathbf{\Delta} \to 0.$$

(ii) The argument here is similar. Indeed we can write $\mathbf{Q}(t) = \mathbf{F}(t)/\|\mathbf{F}(t)\|$ with $\mathbf{Q}(t)$ a unit-norm signal. Namely, we can vectorize and write

$$\mathrm{vec}(\mathbf{Q}(t)) = \sum_{r,\ell} q_{r,\ell}(t) \mathbf{e}_r \otimes \phi_\ell^{\boldsymbol{\Delta}}, \quad \sum_{r,\ell} q_{r,\ell}^2(t) = 1.$$

Then $\mathcal{E}^{\mathrm{Dir}}(\mathbf{Q}(t)) \to \rho_{\boldsymbol{\Delta}}/2$ if and only if

$$\sum_{r,\ell} q_{r,\ell}^2(t) \lambda_\ell^{\boldsymbol{\Delta}} \to \rho_{\boldsymbol{\Delta}}, \quad t \to \infty,$$

which holds if and only if

$$\sum_r q_{r,\rho_{\boldsymbol{\Delta}}}^2(t) \to 1$$
$$q_{r,\ell}^2(t) \to 0, \quad \ell : \lambda_\ell^{\boldsymbol{\Delta}} < \rho_{\boldsymbol{\Delta}}, \tag{28}$$

given the unit norm constraint. This is equivalent to the Rayleigh quotient of $\mathbf{I}_d \otimes \boldsymbol{\Delta}$ converging to its maximal value $\rho_{\boldsymbol{\Delta}}$. When this occurs, for any sequence $t_j \to \infty$ we have that $q_{r,\ell}^2(t_j) \leq 1$, meaning that we can extract a converging subsequence that due to Equation (28) will converge to a unit eigenvector $\mathbf{Q}_\infty$ of $\mathbf{I}_d \otimes \boldsymbol{\Delta}$ satisfying $(\mathbf{I}_d \otimes \boldsymbol{\Delta})\mathbf{Q}_\infty = \rho_{\boldsymbol{\Delta}}\mathbf{Q}_\infty$. Conversely assume for a contradiction that there exists a sequence $t_j \to \infty$ such that $\mathcal{E}^{\mathrm{Dir}}(\mathbf{F}(t_j)/\|\mathbf{F}(t_j)\|) < \rho_{\boldsymbol{\Delta}}/2 - \epsilon$, for some $\epsilon > 0$. Then Equation (28) fails to be satisfied along the sequence, meaning that no subsequence converges to a unit norm eigenvector $\mathbf{F}_\infty$ of $\mathbf{I}_d \otimes \boldsymbol{\Delta}$ with associated eigenvalue $\rho_{\boldsymbol{\Delta}}$ which is a contradiction to our assumption.

$\square$

Before we address the formulation of low(high)-frequency-dominated dynamics, we solve explicitly the system $\dot{\mathbf{F}}(t) = \bar{\mathbf{A}}\mathbf{F}(t)$ in $\mathbb{R}^{n \times d}$, with some initial condition $\mathbf{F}(0)$. We can vectorize the equation and solve $\dot{\mathrm{vec}}(\mathbf{F}(t)) = (\mathbf{I}_d \otimes \bar{\mathbf{A}})\mathrm{vec}(\mathbf{F}(t))$, meaning that

$$\mathrm{vec}(\mathbf{F}(t)) = \sum_{r=1}^d \sum_{\ell=0}^{n-1} e^{(1-\lambda_\ell^{\boldsymbol{\Delta}})t} c_{r,\ell}(0) \mathbf{e}_r \otimes \phi_\ell^{\boldsymbol{\Delta}}, \qquad c_{r,\ell}(0) := \langle \mathrm{vec}(\mathbf{F}(0)), \mathbf{e}_r \otimes \phi_\ell^{\boldsymbol{\Delta}} \rangle.$$

Consider any initial condition $\mathbf{F}(0)$ such that

$$\sum_{r=1}^d |c_{r,0}| = \sum_{r=1}^d \left| \langle \mathrm{vec}(\mathbf{F}(0)), \mathbf{e}_r \otimes \phi_0^{\boldsymbol{\Delta}} \rangle \right| > 0,$$

which is satisfied for each $\mathrm{vec}(\mathbf{F}(0)) \in \mathbb{R}^{nd} \setminus \mathcal{U}^\perp$, where $\mathcal{U}^\perp$ is the orthogonal complement of $\mathbb{R}^d \otimes \mathrm{span}(\phi_0^{\boldsymbol{\Delta}})$. Since $\mathcal{U}^\perp$ is a lower-dimensional subspace, its complement is dense. Accordingly for a.e. $\mathbf{F}(0)$, we find that the solution satisfies

$$\|\mathrm{vec}(\mathbf{F}(t))\|^2 = e^{2t} \left( \sum_{r=1}^d c_{r,0}^2 + \mathcal{O}(e^{-2\mathrm{gap}(\boldsymbol{\Delta})t}) \right) = e^{2t} \left( \|P_{\ker(\boldsymbol{\Delta})}^\perp \mathrm{vec}(\mathbf{F}(0))\|^2 + \mathcal{O}(e^{-2\mathrm{gap}(\boldsymbol{\Delta})t}) \right),$$

with $P_{\ker(\boldsymbol{\Delta})}^\perp$ the projection onto $\mathbb{R}^d \otimes \ker(\boldsymbol{\Delta})$. We see that the norm of the solution increases exponentially, however *the dominant term is given by the projection onto the lowest frequency signal* and in fact

$$\frac{\mathrm{vec}(\mathbf{F}(t))}{\|\mathrm{vec}(\mathbf{F}(t))\|} = \frac{P_{\ker(\boldsymbol{\Delta})}^\perp \mathrm{vec}(\mathbf{F}(0)) + \mathcal{O}(e^{-\mathrm{gap}(\boldsymbol{\Delta})t})(\mathbf{I} - P_{\ker(\boldsymbol{\Delta})}^\perp)\mathrm{vec}(\mathbf{F}(0))}{\left( \|P_{\ker(\boldsymbol{\Delta})}^\perp \mathrm{vec}(\mathbf{F}(0))\|^2 + \mathcal{O}(e^{-2\mathrm{gap}(\boldsymbol{\Delta})t}) \right)^{\frac{1}{2}}} \to \mathrm{vec}(\mathbf{F}_\infty),$$

such that $(\mathbf{I}_d \otimes \boldsymbol{\Delta})\mathrm{vec}(\mathbf{F}_\infty) = \mathbf{0}$ which means $\boldsymbol{\Delta}\mathbf{f}_\infty^r = \mathbf{0}$, for each column $1 \leq r \leq d$. Equivalently, one can compute $\mathcal{E}^{\mathrm{Dir}}(\mathbf{F}(t)/\|\mathbf{F}(t)\|)$ and conclude that the latter quantity converges to zero as $t \to \infty$ by the very same argument.

In fact, this motivates further the nomenclature LFD and HFD. Without loss of generality we focus now on the high-frequency case. Assume that we have a HFD dynamics $t \mapsto \mathbf{F}(t)$,

i.e. $\mathcal{E}^{\text{Dir}}(\mathbf{F}(t)/||\mathbf{F}(t)||) \to \rho_{\boldsymbol{\Delta}}/2$, then we can vectorize the solution and write $\text{vec}(\mathbf{F}(t)) = ||\mathbf{F}(t)||\text{vec}(\mathbf{Q}(t))$, for some time-dependent unit vector $\text{vec}(\mathbf{Q}(t)) \in \mathbb{R}^{nd}$:

$$\text{vec}(\mathbf{Q}(t)) = \sum_{r,\ell} q_{r,\ell}(t)\mathbf{e}_r \otimes \phi_\ell^{\boldsymbol{\Delta}}, \quad \sum_{r,\ell} q_{r,\ell}^2(t) = 1.$$

By Lemma B.1 and more explicitly Equation (28), we derive that the coefficients $\{q_{r,\rho_{\boldsymbol{\Delta}}}\}$ associated with the eigenvectors $\mathbf{e}_r \otimes \phi_{\rho_{\boldsymbol{\Delta}}}^{\boldsymbol{\Delta}}$ are dominant in the evolution hence justifying the name *high-frequency dominated* dynamics.

The next result provides a theoretical justification for the characterization of low (high) frequency dominated dynamics in Definition 3.1.

**Lemma B.2.** *Consider a dynamical system $\dot{\mathbf{F}}(t) = \text{GNN}_\theta(\mathbf{F}(t), t)$, with initial condition $\mathbf{F}(0)$.*

- *(i) $\text{GNN}_\theta$ is LFD if and only if $(\mathbf{I}_d \otimes \boldsymbol{\Delta})\frac{\text{vec}(\mathbf{F}(t))}{||\mathbf{F}(t)||} \to \mathbf{0}$ if and only if for each sequence $t_j \to \infty$ there exist a subsequence $t_{j_k} \to \infty$ and $\mathbf{F}_\infty$ (depending on the subsequence) s.t. $\frac{\mathbf{F}(t_{j_k})}{||\mathbf{F}(t_{j_k})||} \to \mathbf{F}_\infty$ satisfying $\boldsymbol{\Delta}\mathbf{f}_\infty^r = \mathbf{0}$, for each $1 \leq r \leq d$.*

- *(ii) $\text{GNN}_\theta$ is HFD if and only if for each sequence $t_j \to \infty$ there exist a subsequence $t_{j_k} \to \infty$ and $\mathbf{F}_\infty$ (depending on the subsequence) s.t. $\frac{\mathbf{F}(t_{j_k})}{||\mathbf{F}(t_{j_k})||} \to \mathbf{F}_\infty$ satisfying $\boldsymbol{\Delta}\mathbf{f}_\infty^r = \rho_{\boldsymbol{\Delta}}\mathbf{f}_\infty^r$, for each $1 \leq r \leq d$.*

*Proof.* (i) Since $\boldsymbol{\Delta}\mathbf{f}^r(t) \to \mathbf{0}$ for each $1 \leq r \leq d$ if and only if $(\mathbf{I}_d \otimes \boldsymbol{\Delta})\text{vec}(\mathbf{F}(t)) \to \mathbf{0}$, we conclude that the dynamics is LFD if and only if $(\mathbf{I}_d \otimes \boldsymbol{\Delta})\frac{\text{vec}(\mathbf{F}(t))}{||\mathbf{F}(t)||} \to \mathbf{0}$ due to (i) in Lemma B.1. Consider a sequence $t_j \to \infty$. Since $\text{vec}(\mathbf{F}(t_j))/||\mathbf{F}(t_j)||$ is a bounded sequence we can extract a converging subsequence $t_{j_k}$: $\text{vec}(\mathbf{F}(t_{j_k}))/||\mathbf{F}(t_{j_k})|| \to \text{vec}(\mathbf{F}_\infty)$. If the dynamics is LFD, then $(\mathbf{I}_d \otimes \boldsymbol{\Delta})\frac{\text{vec}(\mathbf{F}(t_{j_k}))}{||\mathbf{F}(t_{j_k})||} \to \mathbf{0}$ and hence we conclude that $\text{vec}(\mathbf{F}_\infty) \in \ker(\mathbf{I}_d \otimes \boldsymbol{\Delta})$. Conversely, assume that for any sequence $t_j \to \infty$ there exists a subsequence $t_{j_k}$ and $\mathbf{F}_\infty$ such that $\frac{\mathbf{F}(t_{j_k})}{||\mathbf{F}(t_{j_k})||} \to \mathbf{F}_\infty$ satisfying $\boldsymbol{\Delta}\mathbf{f}_\infty^r = \mathbf{0}$, for each $1 \leq r \leq d$. If for a contradiction we had $\varepsilon > 0$ and $t_j \to \infty$ such that $\mathcal{E}^{\text{Dir}}(\mathbf{F}(t_j)/||\mathbf{F}(t_j)|| \geq \varepsilon$ – for $j$ large enough – then by (i) in Lemma B.1 there exist $1 \leq r \leq d$, $\ell > 0$ and a subsequence $t_{j_k}$ satisfying

$$|\langle\left(\frac{\text{vec}(\mathbf{F}(t_{j_k}))}{||\mathbf{F}(t_{j_k})||}\right), \mathbf{e}_r \otimes \phi_\ell^{\boldsymbol{\Delta}}\rangle| > \delta(\varepsilon) > 0,$$

meaning that there is no subsequence of $\{t_{j_k}\}$ s.t. $(\mathbf{I}_d \otimes \boldsymbol{\Delta})\text{vec}(\mathbf{F}(t_{j_k}))/||\mathbf{F}(t_{j_k})|| \to \mathbf{0}$, providing a contradiction.

(ii) This is equivalent to (ii) in Lemma B.1.

$\square$

**Remark.** *We note that in Lemma B.2 an LFD dynamics does not necessarily mean that the normalized solution converges to the kernel of $\mathbf{I}_d \otimes \boldsymbol{\Delta}$ – i.e. one in general has always to pass to subsequences. Indeed, we can consider the simple example $t \mapsto \text{vec}(\mathbf{F}(t)) := \cos(t)\mathbf{e}_{\bar{r}} \otimes \phi_0^{\boldsymbol{\Delta}}$, for some $1 \leq \bar{r} \leq d$, which satisfies $\boldsymbol{\Delta}\mathbf{f}^r(t) = \mathbf{0}$ for each $r$, but it is not a convergent function due to its oscillatory nature. Same argument applies to HFD.*

We will now show that Equation (36) can lead to a HFD dynamics. To this end, we assume that $\boldsymbol{\Omega} = \tilde{\mathbf{W}} = \mathbf{0}$ so that Equation (36) becomes $\dot{\mathbf{F}}(t) = \bar{\mathbf{A}}\mathbf{F}(t)\mathbf{W}$. According to Equation (10) the negative eigenvalues of $\mathbf{W}$ lead to repulsion. We show that the latter can induce HFD dynamics as per Definition 3.1. We let $P_{\mathbf{W}}^{\rho_-}$ be the orthogonal projection into the eigenspace of $\mathbf{W} \otimes \bar{\mathbf{A}}$ associated with the eigenvalue $\rho_- := |\lambda_-^{\mathbf{W}}|(\rho_{\boldsymbol{\Delta}} - 1)$. We recall that $\lambda_\pm^{\mathbf{W}}$ are the most positive and most negative eigenvalues of $\mathbf{W}$ respectively. We define $\epsilon_{\text{HFD}}$ by:

$$\epsilon_{\text{HFD}} := \min\{\rho_- - \lambda_+^{\mathbf{W}}, |\lambda_-^{\mathbf{W}}|\text{gap}(\rho_{\boldsymbol{\Delta}}\mathbf{I} - \boldsymbol{\Delta}), \text{gap}(|\lambda_-^{\mathbf{W}}|\mathbf{I} + \mathbf{W})(\rho_{\boldsymbol{\Delta}} - 1)\}.$$

**Theorem B.3.** *If $\rho_- > \lambda_+^{\mathbf{W}}$, then $\dot{\mathbf{F}}(t) = \bar{\mathbf{A}}\mathbf{F}(t)\mathbf{W}$ is HFD for a.e. $\mathbf{F}(0)$:*

$$\mathcal{E}^{\mathrm{Dir}}(\mathbf{F}(t)) = e^{2t\rho_-}\left(\frac{\rho_{\boldsymbol{\Delta}}}{2}\|P_{\mathbf{W}}^{\rho_-}\mathbf{F}(0)\|^2 + \mathcal{O}(e^{-2t\epsilon_{\mathrm{HFD}}})\right), \quad t \geq 0,$$

*and $\mathbf{F}(t)/\|\mathbf{F}(t)\|$ converges to $\mathbf{F}_\infty \in \mathbb{R}^{n\times d}$ such that $\boldsymbol{\Delta}\mathbf{f}_\infty^r = \rho_{\boldsymbol{\Delta}}\mathbf{f}_\infty^r$, for $1 \leq r \leq d$. If instead $\rho_- < \lambda_+^{\mathbf{W}}$ then the dynamics is LFD and $\mathbf{F}(t)/\|\mathbf{F}(t)\|$ converges to $\mathbf{F}_\infty \in \mathbb{R}^{n\times d}$ such that $\boldsymbol{\Delta}\mathbf{f}_\infty^r = \mathbf{0}$, for $1 \leq r \leq d$, exponentially fast.*

*Proof of Theorem B.3.* Once we compute the spectrum of $\mathbf{W} \otimes \bar{\mathbf{A}}$ via Equation (23), we can write the solution as – recall that $\bar{\mathbf{A}} = \mathbf{I}_n - \boldsymbol{\Delta}$ so we can rephrase the eigenvalues of $\bar{\mathbf{A}}$ in terms of the eigenvalues of $\boldsymbol{\Delta}$:

$$\mathrm{vec}(\mathbf{F}(t)) = \sum_{r,\ell} e^{\lambda_r^{\mathbf{W}}(1-\lambda_\ell^{\boldsymbol{\Delta}})t}c_{r,\ell}(0)\boldsymbol{\phi}_r^{\mathbf{W}} \otimes \boldsymbol{\phi}_\ell^{\boldsymbol{\Delta}},$$

with $\mathbf{W}\boldsymbol{\phi}_r^{\mathbf{W}} = \lambda_r^{\mathbf{W}}\boldsymbol{\phi}_r^{\mathbf{W}}$, for $1 \leq r \leq d$, where $\{\boldsymbol{\phi}_r^{\mathbf{W}}\}_r$ is an orthonormal basis of eigenvectors in $\mathbb{R}^d$. We can then calculate the Dirichlet energy along the solution as

$$\mathcal{E}^{\mathrm{Dir}}(\mathbf{F}(t)) = \frac{1}{2}\langle \mathrm{vec}(\mathbf{F}(t)), (\mathbf{I}_d \otimes \boldsymbol{\Delta})\mathrm{vec}(\mathbf{F}(t))\rangle = \frac{1}{2}\sum_{r,\ell} e^{2\lambda_r^{\mathbf{W}}(1-\lambda_\ell^{\boldsymbol{\Delta}})t}c_{r,\ell}^2(0)\lambda_\ell^{\boldsymbol{\Delta}}.$$

We now consider two cases:

- If $\lambda_r^{\mathbf{W}} > 0$, then $\lambda_r^{\mathbf{W}}(1 - \lambda_\ell^{\boldsymbol{\Delta}}) \leq \lambda_+^{\mathbf{W}}$.

- If $\lambda_r^{\mathbf{W}} < 0$, then $\lambda_r^{\mathbf{W}}(1 - \lambda_\ell^{\boldsymbol{\Delta}}) \leq |\lambda_-^{\mathbf{W}}|(\rho_{\boldsymbol{\Delta}} - 1) := \rho_-$, with eigenvectors $\boldsymbol{\phi}_r^{\mathbf{W}} \otimes \boldsymbol{\phi}_{\rho_{\boldsymbol{\Delta}}}^{\boldsymbol{\Delta}}$ for each $r$ s.t. $\mathbf{W}\boldsymbol{\phi}_r^{\mathbf{W}} = \lambda_-^{\mathbf{W}}\boldsymbol{\phi}_r^{\mathbf{W}}$ – without loss of generality we can assume that $\rho_{\boldsymbol{\Delta}}$ is a simple eigenvalue for $\boldsymbol{\Delta}$. In particular, if $\lambda_r^{\mathbf{W}} < 0$ and $\lambda_r^{\mathbf{W}}(1 - \lambda_\ell^{\boldsymbol{\Delta}}) < \rho_-$, then

$$\lambda_r^{\mathbf{W}}(1 - \lambda_\ell^{\boldsymbol{\Delta}}) < \max\{|\lambda_-^{\mathbf{W}}|(\lambda_{n-2}^{\boldsymbol{\Delta}} - 1), |\lambda_{-,2}^{\mathbf{W}}|(\rho_{\boldsymbol{\Delta}} - 1)\},$$

where $\lambda_{-,2}^{\mathbf{W}}$ is the second most negative eigenvalue of $\mathbf{W}$ and $\lambda_{n-2}^{\boldsymbol{\Delta}}$ is the second largest eigenvalue of $\boldsymbol{\Delta}$. In particular, we can write

$$\lambda_{n-2}^{\boldsymbol{\Delta}} = \rho_{\boldsymbol{\Delta}} - \mathrm{gap}(\rho_{\boldsymbol{\Delta}}\mathbf{I}_n - \boldsymbol{\Delta}), \quad |\lambda_{-,2}^{\mathbf{W}}| = |\lambda_-^{\mathbf{W}}| - \mathrm{gap}(|\lambda_-^{\mathbf{W}}|\mathbf{I}_d + \mathbf{W}). \tag{29}$$

From (i) and (ii) we derive that if $\lambda_r^{\mathbf{W}}(1 - \lambda_\ell^{\boldsymbol{\Delta}}) \neq \rho_-$, then

$$\lambda_r^{\mathbf{W}}(1 - \lambda_\ell^{\boldsymbol{\Delta}}) - \rho_- < -\min\{\rho_- - \lambda_+^{\mathbf{W}}, \rho_- - |\lambda_-^{\mathbf{W}}|(\lambda_{n-2}^{\boldsymbol{\Delta}} - 1), \rho_- - |\lambda_{-,2}^{\mathbf{W}}|(\rho_{\boldsymbol{\Delta}} - 1)\}$$
$$= -\min\{\rho_- - \lambda_+^{\mathbf{W}}, |\lambda_-^{\mathbf{W}}|\mathrm{gap}(\rho_{\boldsymbol{\Delta}}\mathbf{I} - \boldsymbol{\Delta}), \mathrm{gap}(|\lambda_-^{\mathbf{W}}|\mathbf{I} + \mathbf{W})(\rho_{\boldsymbol{\Delta}} - 1)\} = -\epsilon_{\mathrm{HFD}}, \tag{30}$$

where we have used Equation (29). Accordingly, if $\rho_- > \lambda_+^{\mathbf{W}}$, then

$$\mathcal{E}^{\mathrm{Dir}}(\mathbf{F}(t)) = e^{2t\rho_-}\left(\frac{\rho_{\boldsymbol{\Delta}}}{2}\sum_{r:\lambda_r^{\mathbf{W}}=\lambda_-^{\mathbf{W}}} c_{r,\rho_{\boldsymbol{\Delta}}}^2(0) + \frac{1}{2}\sum_{r,\ell:\lambda_r^{\mathbf{W}}(1-\lambda_\ell^{\boldsymbol{\Delta}})\neq\rho_-} e^{2(\lambda_r^{\mathbf{W}}(1-\lambda_\ell^{\boldsymbol{\Delta}})-\rho_-)t}c_{r,\ell}^2(0)\right)$$
$$= e^{2t\rho_-}\left(\frac{\rho_{\boldsymbol{\Delta}}}{2}\|P_{\mathbf{W}}^{\rho_-}\mathbf{F}(0)\|^2 + \mathcal{O}(e^{-2t\epsilon_{\mathrm{HFD}}})\right).$$

By the same argument we can factor out the dominant term and derive the following limit for $t \to \infty$ and for a.e. $\mathbf{F}(0)$ since $P_{\mathbf{W}}^{\rho_-}\mathrm{vec}(\mathbf{F}(0)) = \mathbf{0}$ only if $\mathrm{vec}(\mathbf{F}(0))$ belongs to a lower dimensional subspace of $\mathbb{R}^{nd}$:

$$\frac{\mathrm{vec}(\mathbf{F}(t))}{\mathrm{vec}(\mathbf{F}(t))} = \frac{P_{\mathbf{W}}^{\rho_-}\mathrm{vec}(\mathbf{F}(0)) + \mathcal{O}(e^{-\epsilon_{\mathrm{HFD}}t})((\mathbf{I} - P_{\mathbf{W}}^{\rho_-})\mathrm{vec}(\mathbf{F}(0)))}{\left(\|P_{\mathbf{W}}^{\rho_-}\mathrm{vec}(\mathbf{F}(0))\|^2 + \mathcal{O}(e^{-2\epsilon_{\mathrm{HFD}}t})\right)^{\frac{1}{2}}} \to \frac{P_{\mathbf{W}}^{\rho_-}\mathrm{vec}(\mathbf{F}(0))}{\|P_{\mathbf{W}}^{\rho_-}\mathrm{vec}(\mathbf{F}(0))\|},$$

where the latter is a unit vector $\mathrm{vec}(\mathbf{F}_\infty)$ satisfying $(\mathbf{I}_d \otimes \boldsymbol{\Delta})\mathrm{vec}(\mathbf{F}_\infty) = \rho_{\boldsymbol{\Delta}}\mathrm{vec}(\mathbf{F}_\infty)$, which completes the proof.

For the opposite case the proof can be adapted without efforts as explicitly derived in the proof of Theorem 4.1. $\square$

### B.3 COMPARISON WITH CONTINUOUS GNNS: DETAILS AND PROOFS

**Comparison with some continuous GNN models**    In contrast with Theorem 3.2, we show that three main *linearized* continuous GNN models are either *smoothing* or more generally LFD. The linearized PDE-GCN$_D$ model Eliasof et al. (2021) corresponds to choosing $\tilde{\mathbf{W}} = \mathbf{0}$ and $\mathbf{\Omega} = \mathbf{W} = \mathbf{K}(t)^\top \mathbf{K}(t)$ in Equation (36), for some time-dependent family $t \mapsto \mathbf{K}(t) \in \mathbb{R}^{d \times d}$:

$$\dot{\mathbf{F}}_{\text{PDE}-\text{GCN}_{\text{D}}}(t) = -\mathbf{\Delta}\mathbf{F}(t)\mathbf{K}(t)^\top \mathbf{K}(t).$$

The CGNN model Xhonneux et al. (2020) can be derived from Equation (36) by setting $\mathbf{\Omega} = \mathbf{I} - \tilde{\mathbf{\Omega}}, \mathbf{W} = -\tilde{\mathbf{W}} = \mathbf{I}$:

$$\dot{\mathbf{F}}_{\text{CGNN}}(t) = -\mathbf{\Delta}\mathbf{F}(t) + \mathbf{F}(t)\tilde{\mathbf{\Omega}} + \mathbf{F}(0).$$

Finally, in linearized GRAND Chamberlain et al. (2021a) a row-stochastic matrix $\mathcal{A}(\mathbf{F}(0))$ is *learned* from the encoding via an attention mechanism and we have

$$\dot{\mathbf{F}}_{\text{GRAND}}(t) = -\mathbf{\Delta}_{\text{RW}}\mathbf{F}(t) = -(\mathbf{I} - \mathcal{A}(\mathbf{F}(0)))\mathbf{F}(t).$$

We note that if $\mathcal{A}$ is not symmetric, then GRAND is *not* a gradient flow.

**Theorem B.4.** $\text{PDE} - \text{GCN}_D$, CGNN *and* GRAND *satisfy the following:*

  *(i)* $\text{PDE} - \text{GCN}_D$ *is a smoothing model:* $\dot{\mathcal{E}}^{\text{Dir}}(\mathbf{F}_{\text{PDE}-\text{GCN}_D}(t)) \leq 0.$

  *(ii) For a.e.* $\mathbf{F}(0)$ *it holds:* CGNN *is never* HFD *and if we remove the source term, then* $\mathcal{E}^{\text{Dir}}(\mathbf{F}_{\text{CGNN}}(t)/||\mathbf{F}_{\text{CGNN}}(t)||) \leq e^{-\text{gap}(\mathbf{\Delta})t}.$

  *(iii) If* $\mathsf{G}$ *is connected,* $\mathbf{F}_{\text{GRAND}}(t) \to \boldsymbol{\mu}$ *as* $t \to \infty$, *with* $\boldsymbol{\mu}^r = \text{mean}(\mathbf{f}^r(0))$, $1 \leq r \leq d$.

By (ii) the source-free CGNN-evolution is LFD *independent of* $\tilde{\mathbf{\Omega}}$. Moreover, by (iii), over-smoothing occurs for GRAND. On the other hand, Theorem 3.2 shows that the negative eigenvalues of $\mathbf{W}$ can make the source-free gradient flow in Equation (36) HFD. Experiments in Section 6 confirm that the gradient flow model outperforms CGNN and GRAND on heterophilic graphs.

We prove the following result which covers Theorem 3.3.

*Proof of Theorem B.4.* We structure the proof by following the numeration in the statement.

(i) From direct computation we find

$$\begin{aligned}\frac{d\mathcal{E}^{\text{Dir}}(\mathbf{F}(t))}{dt} &= \frac{1}{2}\frac{d}{dt}\left(\langle \text{vec}(\mathbf{F}(t)), (\mathbf{I}_d \otimes \mathbf{\Delta})\text{vec}(\mathbf{F}(t))\rangle\right) \\ &= -\langle \text{vec}(\mathbf{F}(t)), (\mathbf{K}^\top(t)\mathbf{K}(t) \otimes \mathbf{\Delta}^2)\text{vec}(\mathbf{F}(t))\rangle \leq 0,\end{aligned}$$

since $\mathbf{K}^\top(t)\mathbf{K}(t) \otimes \mathbf{\Delta}^2 \succeq 0$. Note that we have used that $(\mathbf{A} \otimes \mathbf{B})(\mathbf{C} \otimes \mathbf{D}) = \mathbf{AC} \otimes \mathbf{BD}$.

(ii) We consider the dynamical system

$$\dot{\mathbf{F}}_{\text{CGNN}}(t) = -\mathbf{\Delta}\mathbf{F}(t) + \mathbf{F}(t)\tilde{\mathbf{\Omega}} + \mathbf{F}(0).$$

We can write $\text{vec}(\mathbf{F}(t)) = \sum_{r,\ell} c_{r,\ell}(t)\phi_r^{\tilde{\mathbf{\Omega}}} \otimes \phi_\ell^{\mathbf{\Delta}}$, leading to the system

$$\dot{c}_{r,\ell}(t) = (\lambda_r^{\tilde{\mathbf{\Omega}}} - \lambda_\ell^{\mathbf{\Delta}})c_{r,\ell}(t) + c_{r,\ell}(0), \quad 0 \leq \ell \leq n - 1, \ 1 \leq r \leq d.$$

We can solve explicitly the system as

$$c_{r,\ell}(t) = c_{r,\ell}(0)\left(e^{(\lambda_r^{\tilde{\mathbf{\Omega}}} - \lambda_\ell^{\mathbf{\Delta}})t}\left(1 + \frac{1}{\lambda_r^{\tilde{\mathbf{\Omega}}} - \lambda_\ell^{\mathbf{\Delta}}}\right) - \frac{1}{\lambda_r^{\tilde{\mathbf{\Omega}}} - \lambda_\ell^{\mathbf{\Delta}}}\right), \quad \text{if } \lambda_r^{\tilde{\mathbf{\Omega}}} \neq \lambda_\ell^{\mathbf{\Delta}}$$

$$c_{r,\ell}(t) = c_{r,\ell}(0)(1 + t), \quad \text{otherwise.}$$

We see now that for a.e. $\mathbf{F}(0)$ the projection $(\mathbf{I}_d \otimes \phi_{\rho_{\mathbf{\Delta}}}^{\mathbf{\Delta}}(\phi_{\rho_{\mathbf{\Delta}}}^{\mathbf{\Delta}})^\top)\text{vec}(\mathbf{F}(t))$ is never the dominant term. In fact, if there exists $r$ s.t. $\lambda_r^{\tilde{\mathbf{\Omega}}} \geq \rho_{\mathbf{\Delta}}$, then $\lambda_r^{\tilde{\mathbf{\Omega}}} - \lambda_\ell^{\mathbf{\Delta}} > \lambda_r^{\tilde{\mathbf{\Omega}}} - \rho_{\mathbf{\Delta}}$, for any other non-maximal graph Laplacian eigenvalue. It follows that there is *no* $\tilde{\mathbf{\Omega}}$ s.t. the normalized solution maximizes the Rayleigh quotient of $\mathbf{I}_d \otimes \mathbf{\Delta}$, proving that CGNN is never HFD.

If we have no source, then the CGNN equation becomes

$$\dot{\mathbf{F}}(t) = -\boldsymbol{\Delta}\mathbf{F}(t) + \mathbf{F}(t)\tilde{\boldsymbol{\Omega}} \quad \Longleftrightarrow \quad \mathrm{vec}(\dot{\mathbf{F}}(t)) = (\tilde{\boldsymbol{\Omega}} \oplus (-\boldsymbol{\Delta}))\mathrm{vec}(\mathbf{F}(t)),$$

using the Kronecker sum notation in Equation (24). It follows that we can write the vectorized solution in the basis $\{\boldsymbol{\phi}_r^{\tilde{\boldsymbol{\Omega}}} \otimes \boldsymbol{\phi}_\ell^{\boldsymbol{\Delta}}\}_{r,\ell}$ as

$$\mathrm{vec}(\mathbf{F}(t)) = e^{\lambda_+^{\tilde{\boldsymbol{\Omega}}}t}\left(\sum_{r:\lambda_r^{\tilde{\boldsymbol{\Omega}}}=\lambda_+^{\tilde{\boldsymbol{\Omega}}}} c_{r,0}(0)\boldsymbol{\phi}_r^{\tilde{\boldsymbol{\Omega}}} \otimes \boldsymbol{\phi}_0^{\boldsymbol{\Delta}} + \mathcal{O}(e^{-\mathrm{gap}(\lambda_+^{\tilde{\boldsymbol{\Omega}}}\mathbf{I}_d - \tilde{\boldsymbol{\Omega}})t})\sum_{r:\lambda_r^{\tilde{\boldsymbol{\Omega}}}<\lambda_+^{\tilde{\boldsymbol{\Omega}}}} c_{r,0}(0)\boldsymbol{\phi}_r^{\tilde{\boldsymbol{\Omega}}} \otimes \boldsymbol{\phi}_0^{\boldsymbol{\Delta}}\right)$$

$$+ e^{\lambda_+^{\tilde{\boldsymbol{\Omega}}}t}\left(\mathcal{O}(e^{-\mathrm{gap}(\boldsymbol{\Delta})t})\left(\sum_{r,\ell>0} c_{r,\ell}(0)\boldsymbol{\phi}_r^{\tilde{\boldsymbol{\Omega}}} \otimes \boldsymbol{\phi}_\ell^{\boldsymbol{\Delta}}\right)\right),$$

meaning that the dominant term is given by the lowest frequency component and in fact, if we normalize we find $\mathcal{E}^{\mathrm{Dir}}(\mathbf{F}(t)/||\mathbf{F}(t)||) \leq e^{-\mathrm{gap}(\boldsymbol{\Delta})t}$.

(iii) Finally we consider the dynamical system induced by linear GRAND

$$\dot{\mathbf{F}}_{\mathrm{GRAND}}(t) = -\boldsymbol{\Delta}_{\mathrm{RW}}\mathbf{F}(t) = -(\mathbf{I} - \boldsymbol{\mathcal{A}}(\mathbf{F}(0)))\mathbf{F}(t).$$

Since we have no channel-mixing, without loss of generality we can assume that $d = 1$ – one can then extend the argument to any entry. We can use the Jordan form of $\boldsymbol{\mathcal{A}}$ to write the solution of the GRAND dynamical system as

$$\mathbf{f}(t) = P\mathrm{diag}(e^{J_1 t}, \ldots, e^{J_n t})P^{-1}\mathbf{f}(0),$$

for some invertible matrix $P$ of eigenvectors, with

$$e^{J_k t} = e^{-(1-\lambda_k^{\boldsymbol{\mathcal{A}}})t}\begin{pmatrix} 1 & t & \cdots & \frac{t^{m_k-1}}{(m_k-1)!} \\ & & & \vdots \\ & & & 1 \end{pmatrix},$$

where $m_k$ are the eigenvalue multiplicities. Since by assumption G is connected and augmented with self-loops, the row-stochastic attention matrix $\boldsymbol{\mathcal{A}}$ computed in Chamberlain et al. (2021a) with softmax activation is *regular*, meaning that there exists $m \in \mathbb{N}$ such that $(\boldsymbol{\mathcal{A}}^m)_{ij} > 0$ for each entry $(i, j)$. Accordingly, we can apply Perron Theorem to derive that any eigenvalue of $\boldsymbol{\mathcal{A}}$ has real part smaller than one except the eigenvalue $\lambda_0^{\boldsymbol{\mathcal{A}}}$ with multiplicity one, associated with the Perron eigenvector $\mathbf{1}_n$. Accordingly, we find that each block $e^{J_k t}$ decays to zero as $t \to \infty$ with the exception of the one $e^{J_0 t}$ associated with the Perron eigenvector. In particular, the projection of $\mathbf{f}_0$ over the Perron eigenvector is just $\mu\mathbf{1}_n$, with $\mu$ the average of the feature initial condition. This completes the proof. $\qquad\square$

## B.4 PROPAGATING WITH THE LAPLACIAN

In this subsection we briefly review the special case of Equation (36) where $\boldsymbol{\Omega} = \mathbf{W}$, and comment on why we generally expect a framework where the propagation is governed by the graph vector field $\bar{\mathbf{A}}$ to be more flexible than one with $-\boldsymbol{\Delta}$. If $\boldsymbol{\Omega} = \mathbf{W}$ and we suppress the source term i.e. $\tilde{\mathbf{W}} = \mathbf{0}$, the gradient flow in Equation (36) becomes

$$\dot{\mathbf{F}}(t) = -\boldsymbol{\Delta}\mathbf{F}(t)\mathbf{W}. \tag{31}$$

We note that once vectorized, the solution to the dynamical system can be written as

$$\mathrm{vec}(\mathbf{F}(t)) = \sum_{r=1}^{d}\sum_{\ell=0}^{n-1} e^{-\lambda_r^{\mathbf{W}}\lambda_\ell^{\boldsymbol{\Delta}}t}c_{r,\ell}(0)\boldsymbol{\phi}_r^{\mathbf{W}} \otimes \boldsymbol{\phi}_\ell^{\boldsymbol{\Delta}}.$$

In particular, we immediately deduce the following counterpart to Theorem 3.2:

**Corollary B.5.** *If* $\mathrm{spec}(\mathbf{W}) \cap \mathbb{R}_- \neq \emptyset$, *then Equation* (31) *is* HFD *for a.e.* $\mathbf{F}(0)$.

Differently from Equation (36) the lowest frequency component is *always preserved independent of the spectrum of* $\mathbf{W}$. This means that the system cannot learn eigenvalues of $\mathbf{W}$ to either magnify or suppress the low-frequency projection. In contrast, this can be done if $\boldsymbol{\Omega} = \mathbf{0}$, or equivalently one replaces $-\boldsymbol{\Delta}$ with $\bar{\mathbf{A}}$ providing a further *justification in terms of the interaction between graph spectrum and channel-mixing spectrum for why graph-convolutional models use the normalized adjacency rather than the Laplacian for propagating messages* Kipf & Welling (2017).

## B.5 Revisiting the connection with the manifold case

In Equation (19) a constant nontrivial metric $h$ in $\mathbb{R}^d$ leads to the mixing of the feature channels. We adapt this idea by considering a symmetric positive semi-definite $\mathbf{H} = \mathbf{W}^\top \mathbf{W}$ with $\mathbf{W} \in \mathbb{R}^{d \times d}$ and using it to generalize $\mathcal{E}^{\mathrm{Dir}}$ by suitably weighting the norm of the edge gradients as

$$\mathcal{E}^{\mathrm{Dir}}_{\mathbf{W}}(\mathbf{F}) := \frac{1}{4} \sum_{q,r=1}^{d} \sum_{i} \sum_{j:(i,j) \in \mathbb{E}} h_{qr}(\nabla \mathbf{f}^q)_{ij}(\nabla \mathbf{f}^r)_{ij} = \frac{1}{4} \sum_{(i,j) \in \mathbb{E}} ||\mathbf{W}(\nabla \mathbf{F})_{ij}||^2. \qquad (32)$$

We note the analogy with Equation (19), where the sum over the nodes replaces the integration over the domain and the $j$-th derivative at some point $i$ is replaced by the gradient along the edge $(i,j) \in \mathbb{E}$. We generally treat $\mathbf{W}$ as *learnable weights* and study the gradient flow of $\mathcal{E}^{\mathrm{Dir}}_{\mathbf{W}}$:

$$\dot{\mathbf{F}}(t) = -\nabla_{\mathbf{F}} \mathcal{E}^{\mathrm{Dir}}_{\mathbf{W}}(\mathbf{F}(t)) = -\boldsymbol{\Delta}\mathbf{F}(t)\mathbf{W}^\top \mathbf{W}. \qquad (33)$$

We see that Equation (33) generalizes Equation (1).

**Proposition B.6.** *Let $P^{\mathrm{ker}}_{\mathbf{W}}$ be the projection onto $\ker(\mathbf{W}^\top \mathbf{W})$. Equation (33) is smoothing since*

$$\mathcal{E}^{\mathrm{Dir}}(\mathbf{F}(t)) \leq e^{-2t\mathrm{gap}(\mathbf{W}^\top \mathbf{W})\mathrm{gap}(\boldsymbol{\Delta})}||\mathbf{F}(0)||^2 + \mathcal{E}^{\mathrm{Dir}}((P^{\mathrm{ker}}_{\mathbf{W}} \otimes \mathbf{I}_n)\mathrm{vec}(\mathbf{F}(0))), \quad t \geq 0.$$

*In fact $\mathbf{F}(t) \to \mathbf{F}_\infty$ s.t. $\exists \boldsymbol{\phi}_\infty \in \mathbb{R}^d$: for each $i \in \mathsf{V}$ we have $(\mathbf{f}_\infty)_i = \sqrt{d_i}\boldsymbol{\phi}_\infty + P^{\mathrm{ker}}_{\mathbf{W}}\mathbf{f}_i(0)$.*

*Proof of Proposition B.6.* We can vectorize the gradient flow system in Equation (33) and use the spectral characterization of $\mathbf{W}^\top \mathbf{W} \otimes \boldsymbol{\Delta}$ in Equation (23) to write the solution explicitly as

$$\mathrm{vec}(\mathbf{F}(t)) = \sum_{r,\ell} e^{-(\lambda^{\mathbf{W}}_r \lambda^{\boldsymbol{\Delta}}_\ell)t} c_{r,\ell}(0) \boldsymbol{\phi}^{\mathbf{W}}_r \otimes \boldsymbol{\phi}^{\boldsymbol{\Delta}}_\ell,$$

where $\{\lambda^{\mathbf{W}}_r\}_r = \mathrm{spec}(\mathbf{W}^\top \mathbf{W}) \subset \mathbb{R}_{\geq 0}$ with associated basis of orthonormal eigenvectors given by $\{\boldsymbol{\phi}^{\mathbf{W}}_r\}_r$. Then

$$\mathcal{E}^{\mathrm{Dir}}(\mathbf{F}(t)) = \frac{1}{2}\langle \mathrm{vec}(\mathbf{F}(t)), (\mathbf{I}_d \otimes \boldsymbol{\Delta})\mathrm{vec}(\mathbf{F}(t))\rangle = \frac{1}{2}\sum_{r,\ell} e^{-2t(\lambda^{\mathbf{W}}_r \lambda^{\boldsymbol{\Delta}}_\ell)} c^2_{r,\ell}(0)\lambda^{\boldsymbol{\Delta}}_\ell$$

$$= \frac{1}{2}\sum_{r:\lambda^{\mathbf{W}}_r=0,\ell} c^2_{r,\ell}(0)\lambda^{\boldsymbol{\Delta}}_\ell + \frac{1}{2}\sum_{r:\lambda^{\mathbf{W}}_r>0,\ell>0} c^2_{r,\ell}(0)e^{-2t(\lambda^{\mathbf{W}}_r \lambda^{\boldsymbol{\Delta}}_\ell)}\lambda^{\boldsymbol{\Delta}}_\ell$$

$$= \mathcal{E}^{\mathrm{Dir}}((P^{\mathrm{ker}}_{\mathbf{W}} \otimes \mathbf{I}_n)\mathrm{vec}(\mathbf{F}(0))) + \frac{1}{2}\sum_{r:\lambda^{\mathbf{W}}_r>0,\ell>0} c^2_{r,\ell}(0)e^{-2t(\lambda^{\mathbf{W}}_r \lambda^{\boldsymbol{\Delta}}_\ell)}\lambda^{\boldsymbol{\Delta}}_\ell$$

$$\leq \mathcal{E}^{\mathrm{Dir}}((P^{\mathrm{ker}}_{\mathbf{W}} \otimes \mathbf{I}_n)\mathrm{vec}(\mathbf{F}(0))) + \frac{\rho_{\boldsymbol{\Delta}}}{2}e^{-2t\mathrm{gap}(\mathbf{W}^\top \mathbf{W})\mathrm{gap}(\boldsymbol{\Delta})}||\mathbf{F}(0)||^2,$$

where we recall that $P^{\mathrm{ker}}_{\mathbf{W}}$ is the projection onto $\ker(\mathbf{W}^\top \mathbf{W})$ and that by convention the index $\ell = 0$ is associated with the lowest graph frequency $\lambda^{\boldsymbol{\Delta}}_0 = 0$ – by assumption $\mathsf{G}$ is connected. This proves that the dynamics is in fact smoothing. By the very same argument we find that

$$\mathrm{vec}(\mathbf{F}(t)) \to (\mathbf{I}_d \otimes P^{\mathrm{ker}}_{\boldsymbol{\Delta}})\mathrm{vec}(\mathbf{F}(0)) + (P^{\mathrm{ker}}_{\mathbf{W}} \otimes \mathbf{I}_n)\mathrm{vec}(\mathbf{F}(0)), \quad t \to \infty,$$

with $P^{\mathrm{ker}}_{\boldsymbol{\Delta}}$ the orthogonal projection onto $\ker\boldsymbol{\Delta}$ – the other terms decay exponentially to zero. We first focus on the first quantity, which we can write as

$$(\mathbf{I}_d \otimes P^{\mathrm{ker}}_{\boldsymbol{\Delta}})\mathrm{vec}(\mathbf{F}(0)) = \sum_r c_{r,0}(0)\boldsymbol{\phi}^{\mathbf{W}}_r \otimes \boldsymbol{\phi}^{\boldsymbol{\Delta}}_0,$$

which has matrix representation $\boldsymbol{\phi}^{\boldsymbol{\Delta}}_0 \boldsymbol{\phi}^\top_\infty \in \mathbb{R}^{n \times d}$ with

$$\boldsymbol{\phi}_\infty := \sum_r c_{r,0}(0)\boldsymbol{\phi}^{\mathbf{W}}_r.$$

By Equation (26) we deduce that the $i$-th row of $\boldsymbol{\phi}^{\boldsymbol{\Delta}}_0 \boldsymbol{\phi}^\top_\infty \in \mathbb{R}^{n \times d}$ is the $d$-dimensional vector $\sqrt{d_i}\boldsymbol{\phi}_\infty$. We now focus on the term

$$(P^{\mathrm{ker}}_{\mathbf{W}} \otimes \mathbf{I}_n)\mathrm{vec}(\mathbf{F}(0)) = \sum_{r:\lambda^{\mathbf{W}}_r=0,j} c_{r,j}(0)\boldsymbol{\phi}^{\mathbf{W}}_r \otimes \boldsymbol{\phi}^{\boldsymbol{\Delta}}_j$$

which has matrix representation $\sum_{r:\lambda_r^{\mathbf{W}}=0,j} c_{r,j}(0)\phi_j^{\mathbf{\Delta}}(\phi_r^{\mathbf{W}})^\top$. In particular, the $i$-th row is given by

$$\sum_{r:\lambda_r^{\mathbf{W}}=0,j} c_{r,j}(0)(\phi_j^{\mathbf{\Delta}})_i \phi_r^{\mathbf{W}} = P_{\mathbf{W}}^{\text{ker}}\mathbf{f}_i(0).$$

This completes the proof of Proposition B.6. □

Proposition B.6 implies that no weight matrix $\mathbf{W}$ in Equation (33) **can separate the limit embeddings $\mathbf{F}_\infty$ of nodes with same degree and same input features**. In particular, we have the following characterization:

- Projections of the edge gradients $(\nabla \mathbf{F})_{ij}(0) \in \mathbb{R}^d$ into the eigenvectors of $\mathbf{W}^\top \mathbf{W}$ with positive eigenvalues *shrink* along the GNN and converge to zero exponentially fast as integration time (depth) increases.
- Projections of the edge gradients $(\nabla \mathbf{F})_{ij}(0) \in \mathbb{R}^d$ into the kernel of $\mathbf{W}^\top \mathbf{W}$ stay *invariant*.

If $\mathbf{W}$ has a trivial kernel, then nodes with same degrees converge to the same representation and *over-smoothing*. Differently from Nt & Maehara (2019); Oono & Suzuki (2020); Cai & Wang (2020), over-smoothing occurs independently of the spectral radius of the 'channel-mixing' if its eigenvalues are *positive* – **even for equations which lead to residual** GNNs when discretized Chen et al. (2018). According to Proposition B.6, we do not expect Equation (33) to succeed on heterophilic graphs where *smoothing* processes are generally harmful – this is confirmed in Figure 3 (see *prod*-curve). To deal with heterophily, one needs negative eigenvalues to generate repulsive forces among adjacent features.

**A more general energy.** Since in general one needs to generate repulsive forces too to deal with heterophilic graphs, we extend the Dirichlet energy associated with $\mathbf{H} = \mathbf{W}^\top \mathbf{W} \succeq 0$ to an energy accounting for mutual – possibly repulsive – interactions in feature space $\mathbb{R}^d$. We first rewrite the energy $\mathcal{E}_{\mathbf{W}}^{\text{Dir}}$ in Equation (32) as

$$\mathcal{E}_{\mathbf{W}}^{\text{Dir}}(\mathbf{F}) = \frac{1}{2}\sum_i \langle \mathbf{f}_i, \mathbf{W}^\top \mathbf{W}\mathbf{f}_i \rangle - \frac{1}{2}\sum_{i,j} \bar{a}_{ij}\langle \mathbf{f}_i, \mathbf{W}^\top \mathbf{W}\mathbf{f}_j \rangle. \tag{34}$$

If we replace the occurrences of $\mathbf{W}^\top \mathbf{W}$ with arbitrary symmetric matrices $\mathbf{\Omega}, \mathbf{W} \in \mathbb{R}^{d\times d}$ we obtain

$$\mathcal{E}_\theta(\mathbf{F}) := \frac{1}{2}\sum_i \langle \mathbf{f}_i, \mathbf{\Omega}\mathbf{f}_i \rangle - \frac{1}{2}\sum_{i,j} \bar{a}_{ij}\langle \mathbf{f}_i, \mathbf{W}\mathbf{f}_j \rangle \equiv \mathcal{E}_{\mathbf{\Omega}}^{\text{ext}}(\mathbf{F}) + \mathcal{E}_{\mathbf{W}}^{\text{pair}}(\mathbf{F}), \tag{35}$$

with associated gradient flow of the form (see Appendix B)

$$\dot{\mathbf{F}}(t) = -\nabla_{\mathbf{F}}\mathcal{E}_\theta(\mathbf{F}(t)) = -\mathbf{F}(t)\mathbf{\Omega} + \bar{\mathbf{A}}\mathbf{F}(t)\mathbf{W}. \tag{36}$$

If we include the source term, then we have fully recovered the general energy in Equation (6) and its associated gradient flow.

## C   PROOFS AND ADDITIONAL DETAILS OF SECTION 4

We first explicitly report here the expansion of the discrete gradient flow in Equation (11) after $m$ layers to further highlight how this is not equivalent to a single linear layer with a message passing matrix $\bar{\mathbf{A}}^m$ as for SGCN Wu et al. (2019). For simplicity we suppress the source term.

$$\mathbf{F}(t+\tau) = \mathbf{F}(t) + \tau\left(-\mathbf{F}(t)\mathbf{\Omega} + \bar{\mathbf{A}}\mathbf{F}(t)\mathbf{W}\right)$$

$$\text{vec}(\mathbf{F}(t+\tau)) = \left(\mathbf{I}_{nd} + \tau\left(-\mathbf{\Omega}\otimes\mathbf{I}_n + \mathbf{W}\otimes\bar{\mathbf{A}}\right)\right)\text{vec}(\mathbf{F}(t))$$

$$\text{vec}(\mathbf{F}(m\tau)) = \sum_{k=0}^m \binom{m}{k}\tau^k\left(-\mathbf{\Omega}\otimes\mathbf{I}_n + \mathbf{W}\otimes\bar{\mathbf{A}}\right)^k\text{vec}(\mathbf{F}(0)) \tag{37}$$

and we see how the message passing matrix $\bar{\mathbf{A}}$ actually enters the expansion after $m$ layers with each power $0 \le k \le m$. This is not surprising, given that *we are discretizing a linear dynamical system, meaning that we are approximating an exponential matrix.*

## C.1 FROM ENERGY TO EVOLUTION EQUATIONS: EXACT EXPANSION OF THE GNN SOLUTIONS

We first address the proof of the main result.

*Proof of Theorem 4.1.* We consider a linear dynamical system

$$\mathbf{F}(t + \tau) = \mathbf{F}(t) + \tau \bar{\mathbf{A}} \mathbf{F}(t) \mathbf{W},$$

with $\mathbf{W}$ symmetric. We vectorize the system and rewrite it as

$$\mathrm{vec}(\mathbf{F}(t + \tau)) = (\mathbf{I}_{nd} + \tau \mathbf{W} \otimes \bar{\mathbf{A}}) \mathrm{vec}(\mathbf{F}(t))$$

which in particular leads to

$$\mathrm{vec}(\mathbf{F}(m\tau)) = (\mathbf{I}_{nd} + \tau \mathbf{W} \otimes \bar{\mathbf{A}})^m \mathrm{vec}(\mathbf{F}(0)).$$

We can then write explicitly the solution as

$$\mathrm{vec}(\mathbf{F}(m\tau)) = \sum_{r,\ell} \left(1 + \tau \lambda_r^{\mathbf{W}} (1 - \lambda_\ell^{\mathbf{\Delta}})\right)^m c_{r,\ell}(0) \phi_r^{\mathbf{W}} \otimes \phi_\ell^{\mathbf{\Delta}}.$$

We now verify that by assumption in Equation (13) the dominant term of the solution is the projection into the eigenspace associated with the eigenvalue $\rho_- = |\lambda_-^{\mathbf{W}}|(\rho_{\mathbf{\Delta}} - 1)$. The following argument follows the same structure in the proof of Theorem B.3 with the extra condition given by the step-size. First, we note that for any $r$ such that $\lambda_r^{\mathbf{W}} > 0$, we have

$$|1 + \tau \rho_-| > |1 + \tau \lambda_+^{\mathbf{W}}| \geq |1 + \tau \lambda_r^{\mathbf{W}} (1 - \lambda_\ell^{\mathbf{\Delta}})|$$

since we required $\rho_- > \lambda_+^{\mathbf{W}}$ in Equation (13). Conversely, if $\lambda_r^{\mathbf{W}} < 0$, then

$$|1 + \tau \lambda_r^{\mathbf{W}} (1 - \lambda_\ell^{\mathbf{\Delta}})| \leq \max\{|1 + \tau \rho_-|, |1 + \tau \lambda_-^{\mathbf{W}}|\}$$

Assume that $\tau |\lambda_-^{\mathbf{W}}| > 1$, otherwise there is nothing to prove. Then $|1 + \tau \rho_-| > \tau |\lambda_-^{\mathbf{W}}| - 1$ if and only if

$$\tau |\lambda_-^{\mathbf{W}}| (2 - \rho_{\mathbf{\Delta}}) < 2,$$

which is precisely the right inequality in Equation (13). We can then argue exactly as in the proof of Theorem B.3 to derive that for each index $r$ such that $\lambda_r^{\mathbf{W}} < 0$ and $\lambda_r^{\mathbf{W}} \neq \lambda_-^{\mathbf{W}}$, then

$$|1 + \tau \lambda_r^{\mathbf{W}} (1 - \lambda_\ell^{\mathbf{\Delta}})| \leq \max\{|1 + \tau |\lambda_{-,2}^{\mathbf{W}}|(\rho_{\mathbf{\Delta}} - 1)|, |1 + \tau |\lambda_-^{\mathbf{W}}|(\lambda_{n-2}^{\mathbf{\Delta}} - 1)|\}$$

with $\lambda_{-,2}^{\mathbf{W}}$ and $\lambda_{n-2}^{\mathbf{\Delta}}$ defined in Equation (29). We can then introduce

$$\delta_{\mathrm{HFD}} := \max\{\lambda_+^{\mathbf{W}}, \rho_- - |\lambda_-^{\mathbf{W}}|\mathrm{gap}(\rho_{\mathbf{\Delta}}\mathbf{I} - \mathbf{\Delta}), \rho_- - (\rho_{\mathbf{\Delta}} - 1)\mathrm{gap}(|\lambda_-^{\mathbf{W}}|\mathbf{I} + \mathbf{W}), |\lambda_-^{\mathbf{W}}| - \frac{2}{\tau}\} \quad (38)$$

and conclude that

$$\mathbf{f}_i(m\tau) = \sum_{r,\ell} \left(1 + \tau \lambda_r^{\mathbf{W}} (1 - \lambda_\ell^{\mathbf{\Delta}})\right)^m c_{r,\ell}(0) \phi_\ell^{\mathbf{\Delta}}(i) \phi_r^{\mathbf{W}}$$

$$= (1 + \tau \rho_-)^m \left( c_{-,n-1}(0) \phi_{n-1}^{\mathbf{\Delta}}(i) \cdot \phi_-^{\mathbf{W}} + \mathcal{O}\left(\left(\frac{1 + \tau \delta_{\mathrm{HFD}}}{1 + \tau \rho_-}\right)^m\right) \sum_{\ell,r:\lambda_r^{\mathbf{W}}(1-\lambda_\ell^{\mathbf{\Delta}}) \neq \rho_-} c_{r,\ell}(0) \phi_\ell^{\mathbf{\Delta}}(i) \phi_r^{\mathbf{W}} \right)$$

$$= (1 + \tau \rho_-)^m \left( c_{-,n-1}(0) \phi_{n-1}^{\mathbf{\Delta}}(i) \cdot \phi_-^{\mathbf{W}} + \mathcal{O}(\delta^m)\right),$$

which completes the proof of Equation (14).

Conversely, if $\rho_- < \lambda_+^{\mathbf{W}}$, then the projection onto the eigenspace spanned by $\phi_+^{\mathbf{W}} \otimes \phi_0^{\mathbf{\Delta}}$ is dominating the dynamics with exponential growth $(1 + \tau \lambda_+^{\mathbf{W}}(1 + 0))^m$. We can then adapt the very same argument above by factoring out the dominating term once we note that due to the choice of symmetric normalized Laplacian $\mathbf{\Delta}$, we have $\phi_0^{\mathbf{\Delta}}(i) = \sqrt{d_i}$, which then yields Equation (15).

$\square$

We can now also address the proof of Corollary 4.2.

*Proof of Corollary 4.2.* Once we have the node-wise expansion we can simply compute the Rayleigh quotient of $\mathbf{I}_d \otimes \mathbf{\Delta}$. We report the explicit details for the HFD case since the argument for LFD extends without relevant modifications. Using Equation (12), we can compute the Dirichlet energy along a solution of $\mathbf{F}(t + \tau) = \mathbf{F}(t) + \tau \bar{\mathbf{A}} \mathbf{F}(t) \mathbf{W}$ satisfying Equation (13) by

$$\mathcal{E}^{\mathrm{Dir}}(\mathbf{F}(m\tau)) = \frac{1}{2} \sum_{r,\ell} \left(1 + \tau \lambda_r^{\mathbf{W}} (1 - \lambda_\ell^{\mathbf{\Delta}})\right)^{2m} c_{r,\ell}^2(0) \lambda_\ell^{\mathbf{\Delta}}$$

$$= (1 + \tau \rho_-)^{2m} \left( \frac{\rho_{\mathbf{\Delta}}}{2} \sum_{r:\lambda_r^{\mathbf{W}} = \lambda_-^{\mathbf{W}}} c_{r,\rho_{\mathbf{\Delta}}}^2(0) + \mathcal{O}\left(\left(\frac{1 + \tau \delta_{\mathrm{HFD}}}{1 + \tau \rho_-}\right)\right)^{2m} \sum_{\ell,r:\lambda_r^{\mathbf{W}}(1-\lambda_\ell^{\mathbf{\Delta}}) \neq \rho_-} c_{r,\ell}^2(0) \lambda_\ell^{\mathbf{\Delta}} \right)$$

$$= (1 + \tau \rho_-)^{2m} \left( \frac{\rho_{\mathbf{\Delta}}}{2} \|P_{\mathbf{W}}^{\rho_-} \mathbf{F}(0)\|^2 + \mathcal{O}\left(\left(\frac{1 + \tau \delta_{\mathrm{HFD}}}{1 + \tau \rho_-}\right)^{2m}\right) \right),$$

where $P_{\mathbf{W}}^{\rho_-}$ is the orthogonal projector onto the eigenspace associated with the eigenvalue $\rho_- = |\lambda_-^{\mathbf{W}}|(\rho_{\mathbf{\Delta}} - 1)$. In particular, since

$$\mathrm{vec}(\mathbf{F}(m\tau)) = (1 + \tau \rho_-)^m \left( P_{\mathbf{W}}^{\rho_-} \mathrm{vec}(\mathbf{F}(0)) + \mathcal{O}(\delta^m) \right),$$

we find that the dynamics is HFD with $\mathrm{vec}(\mathbf{F}(t))/\|\mathrm{vec}(\mathbf{F}(t))\|$ converging to the unit projection of the initial projection by $P_{\mathbf{W}}^{\rho_-}$ provided that such projection is not zero, which is satisfied for a.e. initial condition $\mathbf{F}(0)$.

$\square$

## C.2 COMPARISON WITH EXISTING RESULTS: PROOFS

*Proof of Theorem 4.3.* If we drop the residual connection and simply consider $\mathbf{F}(t+\tau) = \tau \bar{\mathbf{A}} \mathbf{F}(t) \mathbf{W}$, then

$$\mathrm{vec}(\mathbf{F}(m\tau)) = (\tau \mathbf{W} \otimes \bar{\mathbf{A}})^m \mathrm{vec}(\mathbf{F}(0)).$$

Since G is not bipartite, the Laplacian spectral radius satisfies $\rho_{\mathbf{\Delta}} < 2$. Therefore, for each pair of indices $(r, \ell)$ we have the following bound:

$$|\lambda_r^{\mathbf{W}}(1 - \lambda_\ell^{\mathbf{\Delta}})| \leq \max\{\lambda_+^{\mathbf{W}}, |\lambda_-^{\mathbf{W}}|\},$$

and the inequality becomes strict if $\ell > 0$, i.e. $\lambda_\ell^{\mathbf{\Delta}} > 0$. The eigenvalues $\lambda_+^{\mathbf{W}}$ and $\lambda_-^{\mathbf{W}}$ are attained along the eigenvectors $\phi_+^{\mathbf{W}} \otimes \phi_0^{\mathbf{\Delta}}$ and $\phi_-^{\mathbf{W}} \otimes \phi_0^{\mathbf{\Delta}}$ respectively. Accordingly, the dominant terms of the evolution lie in the kernel of $\mathbf{I}_d \otimes \mathbf{\Delta}$, meaning that for any $\mathbf{F}_0$ with non-zero projection in $\ker(\mathbf{I}_d \otimes \mathbf{\Delta})$ – which is satisfied by all initial conditions except those belonging to a lower dimensional subspace – the dynamics is LFD. In fact, without loss of generality assume that $|\lambda_-^{\mathbf{W}}| > \lambda_+^{\mathbf{W}}$, then

$$\mathrm{vec}(\mathbf{F}(m\tau)) = |\lambda_-^{\mathbf{W}}|^m \sum_{r:\lambda_r^{\mathbf{W}} = \lambda_-^{\mathbf{W}}} (-1)^m c_{r,0}(0) \phi_-^{\mathbf{W}} \otimes \phi_0^{\mathbf{\Delta}}$$

$$+ |\lambda_-^{\mathbf{W}}|^m \left( \mathcal{O}(\varphi(m)) \left( \mathbf{I}_{nd} - \sum_{r:\lambda_r^{\mathbf{W}} = \lambda_-^{\mathbf{W}}} (\phi_-^{\mathbf{W}} \otimes \phi_0^{\mathbf{\Delta}})(\phi_-^{\mathbf{W}} \otimes \phi_0^{\mathbf{\Delta}})^\top \right) \mathrm{vec}(\mathbf{F}(0)) \right),$$

with $\varphi(m) \to 0$ as $m \to \infty$, which completes the proof. $\square$

**Gradient flow as spectral GNNs.** We finally discuss Equation (11) from the perspective of spectral GNNs as in Balcilar et al. (2020). Let us assume that $\tilde{\mathbf{W}} = \mathbf{0}$, $\mathbf{\Omega} = \mathbf{0}$. If we let $\mathbf{\Delta} = \mathbf{\Phi}^{\mathbf{W}} \mathbf{\Lambda}^{\mathbf{W}} (\mathbf{\Phi}^{\mathbf{W}})^\top$ be the eigendecomposition of the graph Laplacian and $\{\lambda_r^{\mathbf{W}}\}$ be the spectrum of $\mathbf{W}$ with associated orthonormal basis of eigenvectors given by $\{\phi_r^{\mathbf{W}}\}$, and we introduce $\mathbf{z}^r(t) : \mathsf{V} \to \mathbb{R}$ defined by $z_i^r(t) = \langle \mathbf{f}_i(t), \phi_r^{\mathbf{W}} \rangle$, then we can rewrite the discretized gradient flow as

$$\mathbf{z}^r(t + \tau) = \mathbf{\Phi}^{\mathbf{W}} (\mathbf{I} + \tau \lambda_r^{\mathbf{W}} (\mathbf{I} - \mathbf{\Lambda}^{\mathbf{\Delta}}))(\mathbf{\Phi}^{\mathbf{W}})^\top \mathbf{z}^r(t) = \mathbf{z}^r(t) + \tau \lambda_r^{\mathbf{W}} \bar{\mathbf{A}} \mathbf{z}^r(t), \quad 1 \leq r \leq d. \quad (39)$$

Accordingly, for each projection into the $r$-th eigenvector of $\mathbf{W}$, we have a spectral function in the graph frequency domain given by $\lambda^{\mathbf{\Delta}} \mapsto 1 + \tau \lambda_r^{\mathbf{W}} (1 - \lambda^{\mathbf{\Delta}})$. If $\lambda_r^{\mathbf{W}} > 0$ we have a *low-pass* filter

while if $\lambda_r^{\mathbf{W}} < 0$ we have a *high-pass* filter. Moreover, we see that along the eigenvectors of $\mathbf{W}$, if $\lambda_r^{\mathbf{W}} < 0$ then the dynamics is equivalent to flipping the sign of the edge weights, which offers a direct comparison with methods proposed in Bo et al. (2021); Yan et al. (2021) where some 'attentive' mechanism is proposed to learn negative edge weights based on feature information.

The previous equation simply follows from

$$z_i^r(t + \tau) = \langle \mathbf{f}_i(t + \tau), \phi_r^{\mathbf{W}} \rangle = \langle \mathbf{f}_i(t) + \mathbf{W}(\bar{\mathbf{A}}\mathbf{f}(t))_i, \phi_r^{\mathbf{W}} \rangle$$
$$= z_i^r(t) + \lambda_r^{\mathbf{W}} \sum_j \bar{a}_{ij} z_j^r(t),$$

which concludes the derivation of Equation (39).

## D  PROOFS AND ADDITIONAL DETAILS OF SECTION 5

*Proof of Theorem 5.1.*  First we check that if time is continuous, then $\mathcal{E}_\theta$ in Equation (35) is decreasing. We use the Kronecker product formalism to rewrite the gradient $\nabla_{\mathbf{F}}\mathcal{E}_\theta(\mathbf{F})$ as a vector in $\mathbb{R}^{nd}$: explicitly, we get

$$\nabla_{\mathbf{F}}\mathcal{E}_\theta(\mathbf{F}) = (\mathbf{\Omega} \otimes \mathbf{I}_n - \mathbf{W} \otimes \bar{\mathbf{A}})\text{vec}(\mathbf{F}) + (\tilde{\mathbf{W}} \otimes \mathbf{I}_n)\text{vec}(\mathbf{F}(0)).$$

It follows then that

$$\frac{d\mathcal{E}_\theta(\mathbf{F}(t))}{dt} = (\nabla_{\mathbf{F}}\mathcal{E}_\theta(\mathbf{F}(t)))^\top \text{vec}(\dot{\mathbf{F}}(t)) =$$
$$= (\nabla_{\mathbf{F}}\mathcal{E}_\theta(\mathbf{F}(t)))^\top \sigma(-\nabla_{\mathbf{F}}\mathcal{E}_\theta(\mathbf{F}(t))).$$

If we introduce the notation $\mathbf{Z}(t) = -\nabla_{\mathbf{F}}\mathcal{E}_\theta(\mathbf{F}(t)$, then we can rewrite the derivative as

$$\frac{d\mathcal{E}_\theta(\mathbf{F}(t))}{dt} = -\mathbf{Z}(t)^\top \sigma(\mathbf{Z}(t)) = -\sum_\alpha \mathbf{Z}(t)^\alpha \sigma(\mathbf{Z}(t)^\alpha) \leq 0$$

by assumption on $\sigma$. The discrete case follows similarly. Let us use the same notation as above so we can write $\mathbf{F}(t + \tau) = \mathbf{F}(t) + \tau\sigma(\mathbf{Z}(t))$, with $\mathbf{Z}(t) = -\nabla_{\mathbf{F}}\mathcal{E}_\theta(\mathbf{F}(t))$.

$$\mathcal{E}_\theta(\mathbf{F}(t + \tau)) = \langle \text{vec}(\mathbf{F}(t + \tau)), \frac{1}{2}(\mathbf{\Omega} \otimes \mathbf{I}_n - \mathbf{W} \otimes \bar{\mathbf{A}})\text{vec}(\mathbf{F}(t + \tau)) + (\tilde{\mathbf{W}} \otimes \mathbf{I}_n)\text{vec}(\mathbf{F}(0)) \rangle$$

$$= \langle \text{vec}(\mathbf{F}(t)) + \tau\sigma(\mathbf{Z}(t)), \frac{1}{2}(\mathbf{\Omega} \otimes \mathbf{I}_n - \mathbf{W} \otimes \bar{\mathbf{A}})\text{vec}(\mathbf{F}(t + \tau)) + (\tilde{\mathbf{W}} \otimes \mathbf{I}_n)\text{vec}(\mathbf{F}(0)) \rangle$$

$$= \langle \text{vec}(\mathbf{F}(t)) + \tau\sigma(\mathbf{Z}(t)), \frac{1}{2}(\mathbf{\Omega} \otimes \mathbf{I}_n - \mathbf{W} \otimes \bar{\mathbf{A}})(\text{vec}(\mathbf{F}(t) + \tau\sigma(\mathbf{Z}(t))) + (\tilde{\mathbf{W}} \otimes \mathbf{I}_n)\text{vec}(\mathbf{F}(0)) \rangle$$

$$= \langle \text{vec}(\mathbf{F}(t)), \frac{1}{2}(\mathbf{\Omega} \otimes \mathbf{I}_n - \mathbf{W} \otimes \bar{\mathbf{A}})\text{vec}(\mathbf{F}(t)) + (\tilde{\mathbf{W}} \otimes \mathbf{I}_n)\text{vec}(\mathbf{F}(0)) \rangle$$

$$+ \tau\langle \text{vec}(\mathbf{F}(t)), \frac{1}{2}(\mathbf{\Omega} \otimes \mathbf{I}_n - \mathbf{W} \otimes \bar{\mathbf{A}})\sigma(\mathbf{Z}(t)) \rangle$$

$$+ \tau\langle \sigma(\mathbf{Z}(t)), \frac{1}{2}(\mathbf{\Omega} \otimes \mathbf{I}_n - \mathbf{W} \otimes \bar{\mathbf{A}})\text{vec}(\mathbf{F}(t) + (\tilde{\mathbf{W}} \otimes \mathbf{I}_n)\text{vec}(\mathbf{F}(0)) \rangle$$

$$+ \tau^2\langle \sigma(\mathbf{Z}(t)), \frac{1}{2}(\mathbf{\Omega} \otimes \mathbf{I}_n - \mathbf{W} \otimes \bar{\mathbf{A}})\sigma(\mathbf{Z}(t)) \rangle.$$

By using that $\mathbf{\Omega} \otimes \mathbf{I}_n - \mathbf{W} \otimes \bar{\mathbf{A}}$ is symmetric, we find that

$$\mathcal{E}_\theta(\mathbf{F}(t + \tau)) = \mathcal{E}_\theta(\mathbf{F}(t)) + \tau\langle \sigma(\mathbf{Z}(t), (\mathbf{\Omega} \otimes \mathbf{I}_n - \mathbf{W} \otimes \bar{\mathbf{A}})\text{vec}(\mathbf{F}(t) + (\tilde{\mathbf{W}} \otimes \mathbf{I}_n)\text{vec}(\mathbf{F}(0)) \rangle$$

$$+ \tau^2\langle \frac{1}{\tau}(\mathbf{F}(t + \tau) - \mathbf{F}(t)), \frac{1}{2}(\mathbf{\Omega} \otimes \mathbf{I}_n - \mathbf{W} \otimes \bar{\mathbf{A}})\frac{1}{\tau}(\mathbf{F}(t + \tau) - \mathbf{F}(t)) \rangle$$

$$= \mathcal{E}_\theta(\mathbf{F}(t)) - \tau\langle \sigma(\mathbf{Z}(t)), \mathbf{Z}(t) \rangle + \langle \mathbf{F}(t + \tau) - \mathbf{F}(t), \frac{1}{2}(\mathbf{\Omega} \otimes \mathbf{I}_n - \mathbf{W} \otimes \bar{\mathbf{A}})(\mathbf{F}(t + \tau) - \mathbf{F}(t)) \rangle$$

$$\leq \mathcal{E}_\theta(\mathbf{F}(t)) + C_+||\mathbf{F}(t + \tau) - \mathbf{F}(t))||^2,$$

where again we have used that $\mathbf{Z}^\top \sigma(\mathbf{Z}) \geq 0$. This completes the proof.  $\square$

To further support the principle that the effects induced by $\mathbf{W}$ are similar even in this non-linear setting, we consider a simplified scenario.

**Lemma D.1.** *If we choose $\mathbf{\Omega} = \mathbf{W} = \mathrm{diag}(\boldsymbol{\omega})$ with $\omega^r \leq 0$ for $1 \leq r \leq d$ and $\tilde{\mathbf{W}} = \mathbf{0}$ i.e. $t \mapsto \mathbf{F}(t)$ solves the dynamical system*

$$\dot{\mathbf{F}}(t) = \sigma\left(-\mathbf{\Delta}\mathbf{F}(t)\mathrm{diag}(\boldsymbol{\omega})\right),$$

*with $x\sigma(x) \geq 0$, then the standard graph Dirichlet energy satisfies*

$$\frac{d\mathcal{E}^{\mathrm{Dir}}(\mathbf{F}(t))}{dt} \geq 0.$$

*Proof.* This again simply follows from directly computing the derivative:

$$\frac{d\mathcal{E}^{\mathrm{Dir}}(\mathbf{F}(t))}{dt} = \frac{1}{4}\frac{d}{dt}\left(\sum_{r=1}^{d}\sum_{(i,j)\in\mathsf{E}}\left(\frac{f_i^r(t)}{\sqrt{d_i}} - \frac{f_j^r(t)}{\sqrt{d_j}}\right)^2\right)$$

$$= \sum_{r=1}^{d}\sum_{i\in\mathsf{V}}(\mathbf{\Delta}\mathbf{f}^r)_i\sigma(-\omega^r(\mathbf{\Delta}\mathbf{f}^r)_i) = \sum_{r=1}^{d}\sum_{i\in\mathsf{V}}(\mathbf{\Delta}\mathbf{f}^r)_i\sigma(|\omega^r|(\mathbf{\Delta}\mathbf{f}^r)_i) \geq 0.$$

**Important consequence:** The previous Lemma implies that even with non-linear activations, negative eigenvalues of the channel-mixing induce repulsion and indeed the solution becomes less smooth as measured by the classical Dirichlet Energy increasing along the solution. Generalising this result to more arbitrary choices is not immediate and we reserve this for future work.

$\square$

# E   ADDITIONAL DETAILS ON EXPERIMENTS

## E.1   GENERAL EXPERIMENTAL DETAILS

GRAFF is implemented in PyTorch Paszke et al. (2019), using PyTorch geometric Fey & Lenssen (2019) and torchdiffeq Chen et al. (2018). Code and instructions to reproduce the experiments are available on GitHub. Hyperparameters were tuned using wandbBiewald (2020) and random grid search. Experiments were run on AWS p2.8xlarge machines, each with 8 Tesla V100-SXM2 GPUs.

**Methodology.**   Throughout the experiments and ablations we rely on the following parameterisations. We implement $\psi_{\mathrm{EN}}, \psi_{\mathrm{DE}}$ as single linear layers or MLPs, and we set $\mathbf{\Omega}$ to be diagonal. For the real-world experiments we consider *diagonally-dominant* (DD) and *diagonal* (D) choices for the structure of $\mathbf{W}$ that offer explicit control over its spectrum. In the (DD)-case, we consider a $\mathbf{W}^0 \in \mathbb{R}^{d\times d}$ *symmetric* with zero diagonal and $\mathbf{w} \in \mathbb{R}^d$ defined by $\mathbf{w}_\alpha = q_\alpha\sum_\beta|\mathbf{W}^0_{\alpha\beta}| + r_\alpha$, and set $\mathbf{W} = \mathrm{diag}(\mathbf{w}) + \mathbf{W}^0$. Due to the Gershgorin Theorem the eigenvalues of $\mathbf{W}$ belong to $[\mathbf{w}_\alpha - \sum_\beta|\mathbf{W}^0_{\alpha\beta}|, \mathbf{w}_\alpha + \sum_\beta|\mathbf{W}^0_{\alpha\beta}|]$, so the model 'can' easily re-distribute mass in the spectrum of $\mathbf{W}$ via $q_\alpha, r_\alpha$. This generalizes the decomposition of $\mathbf{W}$ in Chen et al. (2020) providing a justification in terms of its spectrum. For (D) we take $\mathbf{W}$ to be diagonal. To investigate the role of the spectrum of $\mathbf{W}$ on synthetic graphs, we construct three additional variants: $\mathbf{W} = \mathbf{W}' + \mathbf{W}'^\top$, $\mathbf{W} = \pm\mathbf{W}'^\top\mathbf{W}'$ named *sum*, *prod* and *neg-prod* respectively where *prod* (*neg-prod*) variants have only non-negative (non-positive) eigenvalues.

## E.2   ABLATION STUDIES ON SYNTHETIC HOMOPHILY

**Synthetic experiments and ablation studies.**   To investigate our claims in a controlled environment we use the synthetic Cora dataset of (Zhu et al., 2020, Appendix G). Graphs are generated for target levels of homophily via preferential attachment – see Appendix E.3 for details. Figure 3 confirms the spectral analysis and offers a better understanding in terms of performance and smoothness of the predictions. Each curve – except GCN – represents one version of $\mathbf{W}$ as in 'methodology' and we implement Equation (11) with $\tilde{\mathbf{W}} = \mathbf{0}, \mathbf{\Omega} = \mathbf{0}$.

Figure 3 (top) reports the test accuracy vs true label homophily. *Neg-prod* is better than *prod* on low-homophily and viceversa on high-homophily. This confirms Theorem 4.1 where we have shown that the gradient flow can lead to a HFD dynamics – that are generally desirable with low-homophily – through the negative eigenvalues of $\mathbf{W}$. Conversely, the *prod* configuration (where we have an attraction-only dynamics) struggles in low-homophily scenarios *even though a residual connection is present*. Both *prod* and *neg-prod* are 'extreme' choices and serve the purpose of highlighting that by turning off one side of the spectrum this could be the more damaging depending on the underlying homophily. In general though 'neutral' variants like *sum* and (DD) are indeed more flexible and better performing. In fact, (DD) outperforms GCN especially in low-homophily scenarios,

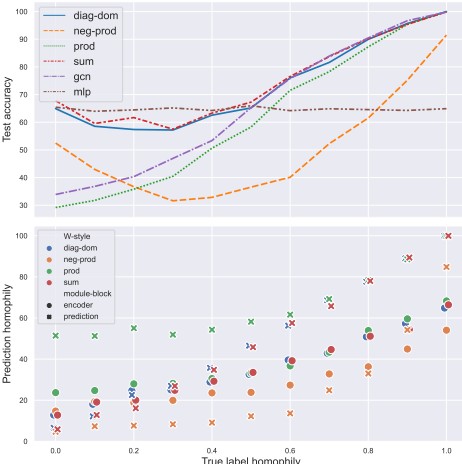

Figure 3: Experiments on synthetic datasets with controlled homophily.

confirming Theorem 4.1 where we have shown that without a residual connection convolutional models are LFD – and hence more sensitive to underlying homophily – irrespectively of the spectrum of $\mathbf{W}$. This is further confirmed in Figure 4.

In Figure 3 (bottom) we compute the homophily of the prediction (cross) for a given method and we compare with the homophily (circle) of the prediction read from the encoding (i.e. *graph-agnostic*). The homophily here is a proxy to assess whether the evolution is *smoothing*, the goal being explaining the smoothness of the prediction via the spectrum of $\mathbf{W}$ as per our theoretical analysis. For *neg-prod* the homophily after the evolution is lower than that of the encoding, supporting the analysis that negative eigenvalues of $\mathbf{W}$ enhance high-frequencies. The opposite behaviour occurs in the case of *prod* and explains that in the low-homophily regime *prod* is under-performant due to the prediction being smoother than the true homophily. (DD) and *sum* variants adapt better to the true homophily. We note how the encoding compensates when the dynamics can only either attract or repulse (i.e. the spectrum of $\mathbf{W}$ has a sign) by decreasing or increasing the initial homophily respectively.

### E.3    ADDITIONAL DETAILS ON SYNTHETIC ABLATION STUDIES:

The synthetic Cora dataset is provided by (Zhu et al., 2020, Appendix G). They use a modified preferential attachment process to generate graphs for target levels of homophily. Nodes, edges and features are sampled from Cora proportional to a mix of class compatibility and node degree resulting in a graph with the required homophily and appropriate feature/label distribution. To validate the provided data before use we provide Table 2 summarising the properties of the synthetic Cora dataset. All rows/levels of homophily have the same number of nodes (1,490), edges (5,936), features (1,433) and classes (5).

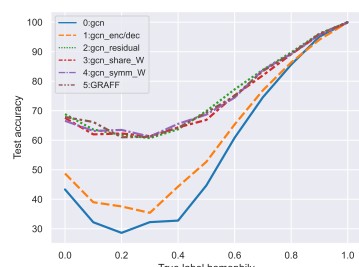

Figure 4: Experiments on synthetic Cora - GCN ablation

As well as the ablation shown in Figure 3 we used this dataset to perform an ablation using GCN as the baseline. We asses the impact of each of the steps necessary to augment a standard GCN model to GRAFF. This involves 5 steps; 1) add an encoder/decoder. 2) add a residual connection. 3) share the weights of $\mathbf{W}$ and $\mathbf{\Omega}$ across time/layers. 4) symmetrize $\mathbf{W}$ and $\mathbf{\Omega}$. 5) remove the non-linearity between layers. The results are shown in Figure 4 and corroborate Theorem 4.1 that adding a residual term is beneficial especially in low-homophily scenarios. We also note augmentations 3,4 and 5 are not "costly" in terms of performance.

### E.4    ADDITIONAL DETAILS ON REAL-WORLD ABLATION STUDIES

For the real-world experiments in Table 1 we performed 10 repetitions over the splits taken from Pei et al. (2020). For all datasets we used the largest connected component (LCC) apart from Citeseer where the 5th and 6th split are LCC and others require the full dataset. For Chameleon and Squirrel

| homophily | max_degree | min_degree | av_degree | density | edge_homoph | node_homoph |
|---|---|---|---|---|---|---|
| 0.00 | 84.33 | 1.67 | 3.98 | 0.0027 | 0.00 | 0.00 |
| 0.10 | 71.33 | 2.00 | 3.98 | 0.0027 | 0.10 | 0.10 |
| 0.20 | 73.33 | 1.67 | 3.98 | 0.0027 | 0.20 | 0.20 |
| 0.30 | 70.00 | 2.00 | 3.98 | 0.0027 | 0.29 | 0.30 |
| 0.40 | 77.67 | 2.00 | 3.98 | 0.0027 | 0.39 | 0.39 |
| 0.50 | 76.33 | 2.00 | 3.98 | 0.0027 | 0.49 | 0.49 |
| 0.60 | 76.00 | 1.67 | 3.98 | 0.0027 | 0.59 | 0.60 |
| 0.70 | 67.67 | 2.00 | 3.98 | 0.0027 | 0.70 | 0.70 |
| 0.80 | 58.00 | 1.67 | 3.98 | 0.0027 | 0.78 | 0.79 |
| 0.90 | 58.00 | 1.67 | 3.98 | 0.0027 | 0.89 | 0.89 |
| 1.00 | 51.00 | 2.00 | 3.98 | 0.0027 | 1.00 | 1.00 |

Table 2: Summary of properties of synthetic Cora dataset

we added self loops and made the edges undirected as a preprocessing step. All other datasets are provided as undirected but without self loops. Each split uses 48/32/20 of nodes for training, validation and test set respectively. Table 6 summarises each of the datasets.

| dataset | nodes | edges | features | classes | max_degree | min_degree | av_degree | density | edge_homoph | node_homoph |
|---|---|---|---|---|---|---|---|---|---|---|
| Texas | 183 | 558 | 1,703 | 5 | 104 | 1 | 3.05 | 0.0167 | 0.06 | 0.06 |
| Wisconsin | 251 | 900 | 1,703 | 5 | 122 | 1 | 3.59 | 0.0143 | 0.18 | 0.16 |
| Cornell | 183 | 554 | 1,703 | 5 | 94 | 1 | 3.03 | 0.0165 | 0.3 | 0.3 |
| Film | 7,600 | 53,318 | 932 | 5 | 1,303 | 1 | 7.02 | 0.0009 | 0.22 | 0.22 |
| Squirrel | 5,201 | 401,907 | 2,089 | 5 | 1,904 | 2 | 77.27 | 0.0149 | 0.23 | 0.29 |
| Chameleon | 2,277 | 65,019 | 2,325 | 5 | 733 | 2 | 28.55 | 0.0125 | 0.26 | 0.33 |
| Citeseer * | 3,327 | 9,104 | 3,703 | 6 | 99 | 0 | 2.74 | 0.0008 | 0.74 | 0.71 |
| Citeseer | 2,120 | 7,358 | 3,703 | 6 | 99 | 1 | 3.47 | 0.0016 | 0.73 | 0.71 |
| Pubmed | 19,717 | 88,648 | 500 | 3 | 171 | 1 | 4.5 | 0.0002 | 0.8 | 0.79 |
| Cora | 2,485 | 10,138 | 1,433 | 7 | 168 | 1 | 4.08 | 0.0016 | 0.8 | 0.81 |

Table 3: Summary of properties of real-word datasets. All LCC except *

We perform an ablation study to further corroborate the behaviour seen in Figure 3. For heterophilic datasets we used the splits from Pei et al. (2020). For homophilic datasets we used the methodology in Shchur et al. (2018), each split randomly selects 1,500 nodes for the development set, from the development set 20 nodes for each class are taken as the training set, the remainder are allocated as the validation set. The remaining nodes outside of the development set are used as the test set. This gives a lower percentage (3-6%) of training nodes. This approach was taken because less training information is needed in the homophilic setting and performance can become less sensitive to other factors, meaning less signal from the controlled variable. We tested the structures of $\mathbf{W}$ against the real-world datasets with known homophily, again *neg-prod* outperforms *prod* in the heterophilic setting and vice-versa due the sign of their spectra.

| dataset | neg_prod | prod | sum |
|---|---|---|---|
| Chameleon | 67.32 | 58.86 | 68.36 |
| Squirrel | 51.39 | 42.11 | 51.29 |
| Cora | 31.80 | 79.65 | 81.17 |
| Citeseer | 32.47 | 67.31 | 67.53 |

Table 4: Ablation with controlled spectrum of $\mathbf{W}$ on real-world datasets

To validate the complexity analysis in Section 6 we performed a runtime ablation for the models between standard GCN and GRAFF described in the GCN ablation Figure 4. The average inference runtime over 100 runs for 1 split of Cora was recorded. We also include runtimes for the provided dense and sparse implementations of GGCN Yan et al. (2021). Adding the encoder/decoder (step 1) speeds up the model due to dimensionality reduction. Subsequent steps also reduce complexity and offer speedup with GRAFF performing the fastest.

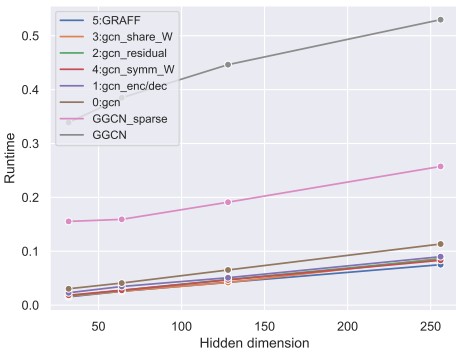

Figure 5: Runtime ablation for inference on Cora dataset

## E.5 DETAILS ON HYPERPARAMETERS

Using wandb Biewald (2020) we performed a random grid search with uniform sampling of the continuous variables. We provide the hyperparameters that achieved the best results from the random grid search in Table 5. An implementation that uses these hyperparameters is available in the provided code with hyperparameters provided in `graff_params.py`. Input dropout and dropout are the rates applied to the encoder/decoder respectively *with no dropout applied in the ODE block*. Further hyperparameters decide the use of non-linearities, batch normalisation, parameter vector $\omega$ and source term multiplier $\beta$ which are specified in the code.

| | w_style | lr | decay | dropout | input_dropout | hidden | time | step_size |
|---|---|---|---|---|---|---|---|---|
| chameleon | diag_dom | 0.0050 | 0.0005 | 0.36 | 0.48 | 64 | 3.33 | 1 |
| squirrel | diag_dom | 0.0065 | 0.0009 | 0.17 | 0.35 | 128 | 2.87 | 1 |
| texas | diag_dom | 0.0041 | 0.0354 | 0.33 | 0.39 | 64 | 0.6 | 0.5 |
| wisconsin | diag | 0.0029 | 0.0318 | 0.37 | 0.37 | 64 | 2.1 | 0.5 |
| cornell | diag | 0.0021 | 0.0184 | 0.30 | 0.44 | 64 | 2.0 | 1 |
| film | diag | 0.0026 | 0.0130 | 0.48 | 0.42 | 64 | 1.5 | 1 |
| Cora | diag | 0.0026 | 0.0413 | 0.34 | 0.53 | 64 | 3.0 | 0.25 |
| Citeseer | diag | 0.0001 | 0.0274 | 0.22 | 0.51 | 64 | 2.0 | 0.5 |
| Pubmed | diag | 0.0039 | 0.0003 | 0.42 | 0.41 | 64 | 2.6 | 0.5 |

Table 5: Selected hyperparameters for real-world datasets

## E.6 EXPERIMENTS ON LARGER HETEROPHILIC GRAPHS

Since our frameworks GRAFF and GRAFF$_{\text{NL}}$ are fast and efficient, we test them on two larger heterophilic datasets arXiv-year (Hu et al., 2021) and snap-patents (Leskovec & Krevl, 2014; Leskovec et al., 2005).

| dataset | nodes | edges | features | classes | edge_homoph |
|---|---|---|---|---|---|
| arxiv-year | 169,343 | 1,166,243 | 128 | 5 | .222 |
| snap-patents | 2,923,922 | 13,975,788 | 269 | 5 | .073 |

Table 6: Statistics of the larger heterophilic datasets.

Both the datasets concern predicting 'publication time' in a citation network. We report baselines as in Lim et al. (2021) and the best performing number among GRAFF and GRAFF$_{\text{NL}}$ in Table 7. Before commenting on the results, we observe that LINK (Zheleva & Getoor, 2009) is a method that only acts on the input adjacency and ignores completely the node features. The facts that LINK is such a strong baseline on the datasets and that the MLP is instead very poor denote that the features on these datasets actually carry very little valuable information. Indeed, most of the MPNNs struggle

|             | arXiv-year        | snap-patents         |
|-------------|-------------------|----------------------|
| MLP         | 36.7 ± 0.21       | 31.34 ± 0.05         |
| L Prop 1-hop | 43.42 ± 0.17     | 30.28 ± 0.09         |
| L Prop 2-hop | 46.07 ± 0.15     | 38.61 ± 0.07         |
| LINK        | 53.97 ± 0.18      | 60.39 ± 0.07         |
| SGC 1-hop   | 32.83 ± 0.13      | 30.31 ± 0.06         |
| SGC 2-hop   | 32.27 ± 0.06      | 29.09 ± 0.09         |
| C&S 1-hop   | 44.51 ± 0.16      | 35.55 ± 0.05         |
| C&S 2-hop   | 49.78 ± 0.26      | 49.08 ± 0.04         |
| GCN         | 46.02 ± 0.26      | 45.65 ± 0.04         |
| GAT         | 46.05 ± 0.51      | 45.37 ± 0.44 (M)     |
| GCNJK       | 46.28 ± 0.29      | 46.88 ± 0.13         |
| GATJK       | 45.8 ± 0.72       | 44.78 ± 0.5          |
| APPNP       | 38.15 ± 0.26      | 32.19 ± 0.07         |
| H2GCN       | 49.09 ± 0.1       | (M)                  |
| MixHop      | 51.81 ± 0.17      | 52.16 ± 0.09         |
| GPR-GNN     | 45.07 ± 0.21      | 40.19 ± 0.03         |
| GCNII       | 47.21 ± 0.28      | 37.88 ± 0.69         |
| LINKX       | 56 ± 1.34         | 61.95 ± 0.12         |
| GRAFF       | 50.52 ± 0.23      | 55.77 ± 0.27         |

Table 7: Performance on larger heterophilic datasets. (M) stands for out of memory.

significantly, with the partial exception of MixHop – which accounts directly for 2-hop information at each layer resulting in worse complexity that GRAFF. The reason why MixHop is more competitive than other MPNNs is because these datasets have some strong 'monophily' type of bias such that architectures that directly access the more homophilic 2-hop are at an advantage; indeed, it has already been noted that LINK is indeed meant to work well in this setting (Altenburger & Ugander, 2018). We also note that the labelling in arXiv-year was introduced in Lim et al. (2021). We point out how GRAFF manages to stay competitive with other MPNNs and almost consistently outperforms them. This confirms the validity of our theoretical analysis and motivates investigating energy functionals that would allow to incorporate higher-order terms as in MixHop in a gradient flow framework. We also note that by losing the inductive bias of message-passing LINKX is not an optimal framework for dealing with homophilic graphs as reported in Lim et al. (2021); on the other hand, an advantage of the gradient flow graph convolutional equations is that they manage to adapt to the underlying homophily of the graph in a way that is provable and justifiable. The latter point is also in stark contrast with LINKX which instead works as a 'black box' on both adjacency and features.

**Remark.** Both arXiv-year and snap-patents graphs come as directed and such information is essential for the task as already noted in Lim et al. (2021). Strictly speaking, by using the directed (and hence non symmetric) adjacency matrix $\bar{\mathbf{A}}$ in Equation (17) and Equation (18) we are no longer a gradient flow due to the lack of symmetry of the gradient of the quadratic energy $\mathcal{E}_\theta$ as noted in Appendix B.1. It is still interesting to observe how symmetrizing and sharing the weights have enabled our framework to beat both GCN and GCNII by a margin even though we are no longer a gradient flow. We reserve a more thorough investigation of gradient flows for directed non symmetric relations to future work.

### E.7  FURTHER COMPARISONS WITH BASELINES

In this subsection we provide further comparison with recent baselines (Luan et al., 2021; Lingam et al., 2021; Maurya et al., 2021) that try to specifically target heterophily. We report their best numbers without reproducing them.

A few comments are in order. First, we have reported the best numbers over all (several) configurations of the reported baselines. We note that despite its simplicity, our framework is very competitive on

| | Texas | Wisconsin | Cornell | Film | Squirrel | Chameleon | Citeseer | Pubmed | Cora |
|---|---|---|---|---|---|---|---|---|---|
| Hom level | **0.11** | **0.21** | **0.30** | **0.22** | **0.22** | **0.23** | **0.74** | **0.80** | **0.81** |
| #Nodes | 183 | 251 | 183 | 7,600 | 5,201 | 2,277 | 3,327 | 18,717 | 2,708 |
| #Edges | 295 | 466 | 280 | 26,752 | 198,493 | 31,421 | 4,676 | 44,327 | 5,278 |
| #Classes | 5 | 5 | 5 | 5 | 5 | 5 | 7 | 3 | 6 |
| $\text{ACM}_{best}$ | $87.84 \pm 3.87$ | $88.43 \pm 3.22$ | $85.95 \pm 5.64$ | $36.89 \pm 1.18$ | $54.4 \pm 1.88$ | $67.08 \pm 2.04$ | $77.15 \pm 1.45$ | $90.00 \pm 0.52$ | $88.01 \pm 1.08$ |
| $\text{HLP}_{best}$ | $87.57 \pm 5.44$ | $86.67 \pm 4.22$ | $84.05 \pm 4.67$ | $34.59 \pm 1.32$ | $74.17 \pm 1.83$ | $77.48 \pm 0.80$ | $NA$ | $NA$ | $NA$ |
| FSGNN | $87.30 \pm 5.55$ | $88.43 \pm 3.22$ | $87.03 \pm 5.77$ | $35.67 \pm 0.69$ | $73.48 \pm 2.13$ | $78.14 \pm 1.25$ | $77.19 \pm 1.35$ | $89.73 \pm 0.39$ | $87.73 \pm 1.36$ |
| GRAFF | $88.38 \pm 4.53$ | $88.83 \pm 3.29$ | $84.05 \pm 6.10$ | $37.11 \pm 1.08$ | $58.72 \pm 0.84$ | $71.08 \pm 1.75$ | $77.30 \pm 1.85$ | $90.04 \pm 0.41$ | $88.01 \pm 1.03$ |
| $\text{GRAFF}_{\text{NL}}$ | $86.49 \pm 4.84$ | $87.26 \pm 2.52$ | $77.30 \pm 3.24$ | $35.96 \pm 0.95$ | $59.01 \pm 1.31$ | $71.38 \pm 1.47$ | $76.81 \pm 1.12$ | $89.81 \pm 0.50$ | $87.81 \pm 1.13$ |

Table 8: Node-classification results.

| | **Cora** | **Citeseer** | **Chameleon** | **Squirrel** |
|---|---|---|---|---|
| Hom level | **0.81** | **0.74** | **0.23** | **0.22** |
| #Nodes | 2,708 | 3,327 | 2,277 | 5,201 |
| #Edges | 5,278 | 4,676 | 31,421 | 198,493 |
| #Classes | 6 | 7 | 5 | 5 |
| Test accuracy | $87.69 \pm 0.86$ | $76.75 \pm 1.33$ | $70.44 \pm 1.47$ | $59.60 \pm 1.49$ |
| $\mathcal{E}_{\text{in}}^{\text{Dir}}$ | 0.41 | 0.22 | 0.94 | 0.80 |
| $\mathcal{E}_{\text{fin}}^{\text{Dir}}$ | 0.16 | 0.10 | 0.73 | 0.77 |
| $\lambda_+^{\mathbf{W}}/|\lambda_-^{\mathbf{W}}|$ | 1.93 | 1.02 | 0.98 | 1.02 |
| $\lambda_-^{\mathbf{W}}$ | $-0.40$ | $-0.95$ | $-5.05$ | $-4.80$ |

Table 9: Spectral analysis.

the small heterophilic graphs and stronger on the larger dataset Film and on *all the homophilic ones*. On the other hand, HLP and FSGNN are much stronger baselines on Squirrel and Chameleon. This is also mainly because both the architectures handle the graphs as directed without self-loops which helps the performance massively and we suspect effectively reduces the heterophily of the graph hence making tha task easier. We reserve the investigation of heterophily in the context of directed graphs for future work.

### E.8 SPECTRAL PROPERTIES OF THE LEARNT CHANNEL-MIXING

In Table 9 we report a few spectral properties of the channel-mixing $\mathbf{W}$ learnt on 4 real datasets – two homophilic and two heterophilic – along with the value of the normalized Dirichlet energy $\mathbf{X} \mapsto \mathcal{E}^{\text{Dir}}(\mathbf{X})/\|\mathbf{X}\|^2$ used for characterizing LFD and HFD dynamics.

Some important comments are in order. The quantities $\mathcal{E}_{\text{in}}^{\text{Dir}}$ and $\mathcal{E}_{\text{fin}}^{\text{Dir}}$ are the normalized Dirichlet energy of the encoded node features before and after the diffusion layers respectively; recall that these are values between $0$ and $2$ we introduced in Section 3 to characterize whether a given GNN is mostly smoothing or not. $\lambda_+^{\mathbf{W}}/|\lambda_-^{\mathbf{W}}|$ instead is the ratio between most positive and most negative eigenvalue. (i) we note how the initial encoder provides us with node-wise features that are increasingly less smooth depending on the underlying graph heterophily – as we derive from $\mathcal{E}_{\text{in}}^{\text{Dir}}$. (ii) The model learns to adapt to the underlying homophily as we see from the fact that the normalized Dirichlet energy decreases much more after the message-passing for Cora and Citeseer. (iii) The ratio $\lambda_+^{\mathbf{W}}/|\lambda_-^{\mathbf{W}}|$ – which is a partial indicator of the amount of attraction vs repulsion exerted by the diffusion – is also an indicator of performances as we read when comparing the case of CORA with the other datasets. (iv) As we increase the heterophily, the most negative eigenvalue increases in absolute value since most likely more repulsion is needed. We argue that the model might benefit from inducing fast attraction along some directions and fast repulsion along others – since on heterophilic graphs we have larger positive and negative eigenvalues. Connected to this last point though, we note that at the end of the architecture we have a decoder that can also potentially discard some node feature projections hence keeping only the smoother (or less smooth) components.

