# OpenReview forum: "Graph Neural Networks as Gradient Flows: understanding graph convolutions via energy"
_ICLR.cc/2023/Conference — Submitted to ICLR 2023_

### Official Review · Reviewer_Sead · 2022-10-23

**Confidence:** 3
**Correctness:** 4
**Technical Novelty And Significance:** 4
**Empirical Novelty And Significance:** 2
**Recommendation:** 6

**Clarity, Quality, Novelty And Reproducibility:**

- Paper clarity and quality is great
- The work is original and novel to my knowledge
- The presented GNN architecture seems easy to reproduce with additional content given in suplementals

**Strength And Weaknesses:**

Strengths:
- The presented theoretical framework is novel and interesting to read. It provides a new perspective on GNNs.
- The framework unifies many existing GNN architectures and is able to derive conclusions about their behaviour with respect to favoring high or low frequencies. These conclusions are partially supported by proof of concept experiments.
- The paper exposes a guideline of how to design GNNs that favor high frequency encodings, which might be useful to certain practical tasks, as most existing GNNs favor low frequency.
- The paper is very well written. I am not an expert in ODEs and only roughly familiar with the most recent GNN theory and I was able to follow the paper mostly.

Weaknesses:
- The experiments are on the weak side of the paper. While theoretically shown effects are supported by proof of concept experiments, real applications of the presented phenomena are lacking.
- In general, it would be great if the authors make stronger connections to practice, e.g. by zooming in on a specific application/graph type that would benefit from the presented method in a special way. That would also give a good intuition about how the knowledge can be applied to other tasks as well.

**Summary Of The Paper:**

This mostly theoretical work presents a framework of GNNs in the context of Dirichlet energy minimization/maximization. GNNs are framed as estimators for gradients of energy functionals that point to either of those two directions. It shows that for certain classes of graphs and GNNs, attractive or repulsive forces between node features overweigh the other, leading to dominance of low or high frequencies, respectively, after convergence.  Based on this framework, the authors are able to design a GNN architecture that works well for low-homophily and high-homophily, depending on parameterization.

**Summary Of The Review:**

In summary, I think this paper should be accepted, as it provides a new perspective of GNNs as gradient estimators for energy functionals. The paper, while reading, gave me a few ideas and a mental framework to work with, when thinking about GNN solutions for specific problems.

Questions:
- This work focusses on architectures favoring low or high frequency content. I was wondering if it wouldn't be best to converge to a range of different frequency content, e.g. in different feature groups, to extract most from the given graph. What do the authors think here?
- As far as I understand, the architecture only allows for switching between low and high frequency convergence with sign flipping. Could you think of a way of providing more explicit control over the frequency in the given framework?

---

> ### Author Response · Authors · 2022-11-08
> **Thank you for the review: rebuttal**
>
> Thank you for your analysis and the stimulating final questions. We address your concerns below individually.
>
> "_This mostly theoretical work_"
>
> Thank you for recognizing that the work is indeed mostly theoretical and of understanding.
>
> "_The experiments are on the weak side of the paper. While theoretically shown effects are supported by proof of concept experiments, real applications of the presented phenomena are lacking._"
>
> Thank you for your feedback. We are actually unsure why you refer to our experiments as proof of concept. This is a list of references included in our paper that, for the great part, have the specific and unique goal of proposing models that work well with heterophily: _Beyond low-frequency information in graph convolutional networks_, _Neural sheaf diffusion: A topological perspective on heterophily and oversmoothing in Gnns_, _Two sides of the same coin: Heterophily and oversmoothing in graph convolutional neural networks_,  _Dirichlet energy constrained learning for deep graph neural networks_, _Beyond homophily in graph neural networks: Current limitations and effective designs_, _Simple and deep graph convolutional networks_,  _Is Heterophily A Real Nightmare For Graph Neural Networks To Do Node Classification?_, _Simple Truncated SVD based Model for Node Classification on Heterophilic Graphs_ and _Improving Graph Neural Networks with Simple Architecture Design_. All these references _have experiments on the node classification task only and using the very same datasets we have tested on, so we believe our evaluation to be exhaustive and not just a proof of concept_. Since this seems to be the only and main critical point you raised, if you have further questions about that please let us know otherwise we would greatly appreciate you considering raising the score accordingly given that we have in fact performed experiments on very common benchmarks adopted by similar papers in the literature that cannot be deemed as proof of concept.
>
> "_In general, it would be great if the authors make stronger connections to practice (...)_"
>
> Thank you for the input. Again the application here is semi-supervised node classification which is very typical in GNN literature (see previous response).
>
> "_This work focusses on architectures favoring low or high frequency content. (...) Could you think of a way of providing more explicit control over the frequency in the given framework?_"
>
> These are two extremely valuable questions we have decided to group here and address jointly. We agree with you that in principle one would need more expressive power in the frequency domain by also being able to hit the medium frequencies. We propose here two answers, one that pertains to GRAFF and one that goes beyond the proposed framework and extends it. If you consider the explicit expansion of a GNN solution in GRAFF provided in eq. (12), note that once we are given a number of layers m, the contributions associated with a frequency $\lambda_{\ell}^{\Delta}$ are also weighted by $c_{r,\ell}(0)$ meaning that the initial (learnable) encoder – for a given fixed number of layers m – can in principle learn to weight that frequency more. Note that Theorem 4.1 is also a valuable theoretical contribution precisely because of your comments, since it shows that graph convolutional models may fail to properly enhance the medium frequencies (when and if they are needed). The second answer instead is that if one replaces the term $\beta$ in eq. (17) with a time-dependent scalar $\beta(t)$ (i.e. one for each layer) then we can show that the m-degree GRAFF can be equal to any m-degree polynomial in the graph frequency domain, meaning that we can indeed have a more explicit control over the medium frequencies as well. We reserve a more detailed investigation of this in future works where we extend GRAFF. Once again these are very insightful questions resonating with things we have already been thinking about and hope to have addressed them here.
>
> _Empirical Novelty And Significance:  2_
>
> Thank you for your feedback. We hope that in light of our response we have convinced you why we disagree with the score. We hope that we have addressed your concerns in regard to empirical evaluation by showing how many recent papers targeting the same problems of heterophily and frequency response in GNNs have tested on our same benchmarks. We actually believe that the empirical contribution is not marginal given that we show how a subclass of graph convolutional models that is motivated by the gradient flow (and indeed results in an architecture with shared,symmetric weights) can beat much slower baselines (see Figure 5 where we compare with GCN and GGCN in terms of runtime) in the heterophilic scenario. If we have sufficiently supported that our experiments are indeed exhaustive and common in GNN literature as we believe, we hope you would consider raising the score.

---

### Official Review · Reviewer_Hwvc · 2022-10-23

**Confidence:** 4
**Correctness:** 3
**Technical Novelty And Significance:** 2
**Empirical Novelty And Significance:** 2
**Recommendation:** 3

**Clarity, Quality, Novelty And Reproducibility:**

*The paper is not the easiest to follow.

*The paper offers a quality analysis of GNNs through their energy functions.

*The paper continues the recent efforts in GNNs (as properly cited by the authors) and adopts insights and ideas from the Image Processing world (e.g., Scale-space and edge detection using anisotropic diffusion, as properly cited by the authors)

*The authors claim to include the code and hyperparameters, but I could not find it anywhere. I therefore cannot say that it is reproducible.

**Strength And Weaknesses:**

Strengths:

+This work takes inspiration from recent works like GRAND, PDE-GCN and GraphCON and extends the analysis of GNNs and their interpretation as dynamical systems that can be explored through their energies.

+The authors sufficiently cite and discuss relevant background.

Weaknesses:

-The authors discuss the possibility of 'negative edge weight' in page 7. However, it is not clear what is the relation between allowing negative edge weights and recent works that propose designated architectures for this kind of data, for instance "Learning signed network embedding via graph attention" (AAAI 2020), "Signed graph attention networks" (ICANN 2019).

-The authors show in Theorem 5.1 that the considered energy decreases from layer to layer which as discussed in the paper it is not anticipated to work well for heterophilic datasets. Then, I do not understand the obtained accuracy on such datasets. What causes the method to behave well? did the authors use only a small number of layers?

-The phenomenon of oversmoothing is widely spread in GNNs and many works (that are properly cited in the paper) propose various ways to tackle this issue. The authors also discuss this issue but no study was done in this important aspect.

-In page 8, paragraph "The model and the parameterisation" the authors suggest two models for the learned matrix W, but it is not clear what is the real influence of each of them. According to the appendix, I can only know what model was used, but as a reader I cannot infer what is the impact of those models on the different datasets. I think that the authors should add an experiment that studies this aspect.

-It is not reported how many layers are used for the different experiments. I think that in this aspect it is important also given the theorems that regard to the energy change between layers.

-Regarding the results in table 1, I believe that the authors are missing comparison with recent works with better performance. For instance "Is Heterophily A Real Nightmare For Graph Neural Networks To Do Node Classification?", "Simple Truncated SVD based Model for Node Classification on Heterophilic Graphs" and "Improving Graph Neural Networks with Simple Architecture Design" obtain significantly better accuracy on some of the datasets (e.g., Chameleon and Squirrel)

-I appreciate the number of experiments but I think that to allow a broader comparison with many existing methods the authors should also report the accuracy on the semi-supervised Cora, Citeseer and Pubmed datasets.

-It is claimed that the code and hyperparameters are provided but I cannot find them.

*A question to the authors: can this method be used for graph classification?


**Summary Of The Paper:**

This work suggests an energy based perspective on GNNs, and by an analysis of the application of GNN layers and the channel-mixing operator sheds light on the behavior of GNNs.
Several experiments are conducted showing improvement over the considered methods in some cases.

**Summary Of The Review:**

The paper continues the interesting direction of interpreting GNNs through their energy function. I think that it has a merit but the experiments can be extended and compared with more methods, and many of the details are missing.

---

> ### Author Response · Authors · 2022-11-08
> **Thank you for the review: rebuttal part 1**
>
> Thank you for the review and the various input concerning evaluation. Before we address each comment individually we kindly ask you to consider reading our general response with emphasis on the technical (theoretical) contributions that we feel have been a little ignored here despite them representing our main contributions. We hope that as a consequence of our rebuttal you will consider revising your score, in particular based on the novelty and significance of our theoretical (technical) contributions.
>
> "_The authors discuss the possibility of 'negative edge weight' in page 7 (...) "Learning signed network embedding via graph attention" (AAAI 2020), "Signed graph attention networks" (ICANN 2019)._"
>
> Thank you for raising this valuable point – we have now also included the reference. Due to space constraints, in the main body of the paper we could not expand further the point raised in pag. 7. However, we have done so in more detail in the appendix, see eq. (39) pag. 26. Here we derive how along a $\mathbf{W}$-eigenvector with negative eigenvalue the GNN is equivalent to flipping the sign of the edge weights and hence establish a nice connection with existing methods. We kindly invite you to read this part of the appendix and let us know if it answers your concerns. In the meantime, based on your feedback we have revised pag. 7 adding a pointer to eq. (39).
>
> "_The authors show in Theorem 5.1 that the considered energy decreases from layer to layer (...)_"
>
> Thank you for the important question. What we argue in the paper is that generally if you run the dynamics for too long, you would risk losing too much information due to dominant effects as precisely derived in Theorem 4.1. The accuracy is due to the fact that by finding the right energy (i.e. the right forces via the spectra of the channel-mixing) we can have a small number of layers and still induce repulsion. The ability to generate repulsion is an important feature of our framework to deal with heterophily; indeed we have proved in Theorem 3.3 there are existing GNN frameworks that fail to induce repulsion independently of the number of layers (and hence struggle with heterophily, as validated in our Table 1).
>
> "_The phenomenon of oversmoothing is widely spread in GNNs and many works (...)._"
>
> Thank you for raising this issue which we hope to clarify. On a theoretical level, we have actually proved that residual graph convolutions that follow the gradient flow paradigm, can induce an HFD dynamics and hence can learn to be dominated by the high-frequency dynamics as per Theorem 4.1. This rigorously proves that our framework can avoid over-smoothing. In terms of architectures with a very large number of layers, Theorem 4.1. actually shows that residual graph convolutions will incur in over-separation depending on negative eigenvalues when the depth is too large. This is a theoretical contribution of its own and explains why for a simple convolutional model one needs to halt the diffusion at some point. However, note how the experiments show that if you can also induce repulsion from the get-go, you do not need arbitrarily many layers to reach competitive performance on heterophilic datasets.
>
> "_In page 8, paragraph "The model and the parameterisation" the authors suggest two models for the learned matrix W, but it is not clear what is the real influence of each of them. (...)_"
>
> Thank you for the question. Please note that we have reported in Table 5 the different parameterisation used on the real datasets.
>
> "_It is not reported how many layers are used for the different experiments (...)_"
>
> Thanks for the question. The number of layers is actually reported for all the datasets in table 5 (it can be computed as time/step_size).
>
> "_Regarding the results in table 1, I believe that the authors are missing comparison with recent works with better performance. (...)_"
>
> Thanks for the references, which we have now included. We first note that the papers shared are arguably more on the empirical side and lack the theoretical analysis performed here that instead would grant us a better understanding of _why and how_ convolution on graphs may help in _any architecture_ even with heterophily (see thm 4.1 and thm 5.1). We report the numbers you asked in Table 8 and have added a pointer in the main text. A few comments. We note that despite its simplicity, our framework is very competitive on the small heterophilic graphs and stronger on the larger dataset Film and on _all the homophilic ones_. On the other hand, HLP and FSGNN are much stronger baselines on Squirrel and Chameleon. This is also mainly because _both the architectures handle the graphs as directed without self-loops_ which helps the performance massively and we suspect effectively reduces the heterophily of the graph. We reserve the investigation of heterophily in the context of directed graphs for future work and thank you for pointing us to these references.

---

> > ### Author Response · Authors · 2022-11-08
> > **Thank you for the review: rebuttal part 2**
> >
> > "_I appreciate the number of experiments but I think that to allow a broader comparison with many existing methods the authors should also report the accuracy on the semi-supervised Cora, Citeseer and Pubmed datasets._"
> >
> > Thanks for the input. This is a list of references included in our paper that, for the great part, have the specific and unique goal of proposing models that work well with heterophily: _Beyond low-frequency information in graph convolutional networks_, _Neural sheaf diffusion: A topological perspective on heterophily and oversmoothing in Gnns_, _Two sides of the same coin: Heterophily and oversmoothing in graph convolutional neural networks_,  _Dirichlet energy constrained learning for deep graph neural networks_, _Beyond homophily in graph neural networks: Current limitations and effective designs_, _Simple and deep graph convolutional networks_, along with all the references you recommended in a previous comment that have experiments on node classification task only and using the very same datasets we have tested on, _so we believe our evaluation to be exhaustive_. Note that in Appendix E we have also conducted experiments on larger heterophilic datasets.
> >
> > "_It is claimed that the code and hyperparameters are provided but I cannot find them._"
> >
> > Hyperparameters have been reported in Table 5 (note that the number of layers can be computed as time/step-size). Concerning the code we are currently not at liberty of releasing it but will do so for the camera ready.
> >
> > "_A question to the authors: can this method be used for graph classification?_"
> >
> > This is a valuable question. The answer is of course affirmative for the same reason why methods like GCN can indeed be used for graph classification as well – given that our architecture is a gradient flow instance of residual GCN. Since the main goal of the paper amounts to a theoretical analysis of the frequency response of GNNs we focused on node-level tasks as similar references have done on the same datasets (see previous response).
> >
> > "_The paper is not the easiest to follow._"
> >
> > Thank you for the input. Let us know if there are specific parts that you think should be improved. Note that we have a notation paragraph at the end of pag. 2 and at pag. 14 to facilitate the reader along with outlines of the most theoretical Sections 3,4,5 at pages 5,7,8 respectively.
> >
> > "_The authors claim to include the code and hyperparameters, but I could not find it anywhere. I therefore cannot say that it is reproducible._"
> >
> > Please see our previous response.
> >
> > "_Technical Novelty And Significance: 2: The contributions are only marginally significant or novel."_
> >
> > Thank you for your feedback. We believe that the technical (theoretical) novelty of the paper is not marginally significant given that we are the first (i) To realise that graph convolutions can be derived through a physics-inspired framework, leading to more interpretable architectures that can be quite competitive on heterophilic graphs as well and be a building point for researchers in the future (ii) To prove that the dynamics of GCNs is controlled by the interactions between the laplacian and channel-mixing spectra (iii) To prove that even non-linear GCNs can be interpreted using the same energy (something that generalise existing theoretical work that focus on GCNs wrt the Dirichlet Energy and only activated with ReLU) (iv) to introduce LFD/HFD characterizations to monitor over-smoothing that can be used in any future work on the topic. We are confident that our approach could stem several new works that aim at both understanding GNNs via energies and developing new frameworks that are more physics-oriented. We hope that based on our responses you would consider raising the technical novelty and significance of our paper given that we consider it to be not at all marginal for all the reasons described here. If you disagree about these points, please let us know.
> >
> > $\textbf{Final comment}$: We hope that you find our rebuttal helpful and that addresses your points  – mainly concerning technical novelty and evaluation which we have all addressed both in the rebuttal and in the revised version. If that is not the case, please let us know of other standing doubts/concerns and we will do our best to address them. Otherwise, we kindly ask to revise your score in light of our responses and clarifications, especially considering the technical novelty and significance of our submission.

---

> > > ### Comment · Reviewer_Hwvc · 2022-11-18
> > > **Discussion**
> > >
> > > Thank you for your rebuttal.
> > > I read it all (including the comments to other reviewers), and also your revised paper.
> > >
> > > I am still left with a few concerns:
> > >
> > > 1. Regarding finding the right energy: how can you be certain that you can find the right energy using your method? Also, what does 'running the dynamics for too long' mean ? is it 16 layers ? 32 ? 64 ? It is not very clear to me. If the proposed method can create repulsive forces then why should it lose too much information?
> > >
> > > 2. Regarding missing experimental results of oversmoothing prevention: I read your answer but it still doesn't tell me if in practice your method prevents oversmoothing. I do appreciate the theorem but in the end this is a practical theorem can be easily validated. So I think that to really verify this theorem, experimental results are required.
> > >
> > > 3. Regarding types of parameterizations: Thank you for your answer. What you show in table 5 is exactly what makes me not understand the influence of the parameterizations. In my review I meant that I would have liked to see some sort of an ablation study where the reader can understand the influence of the proposed parameterizations on a few datasets. However in your report I can only learn what worked the best, I assume.
> > >
> > > 4. Regarding other methods: I appreciate the discussion of other methods with high accuracy. In the context of why and how methods work I think that additional recent papers that also achieve high accuracy can be considered, for example: "How Powerful are Spectral Graph Neural Networks".
> > >
> > > 5. Regarding semi-supervised experiments: Thank you for your answer. I am still holding the opinion that showing the additional type of experiment is valuable.

---

> > > > ### Author Response · Authors · 2022-11-18
> > > > **Thanks: discussion**
> > > >
> > > > Thanks for getting back to us. Given the time, at this point we are not running further experiments since we also believe that our evaluation has been exhaustive as amply demonstrated by the references shared in our previous response.
> > > >
> > > > 1. "_Regarding finding the right energy (...)_" We hope you can agree that this question is like asking how one can be sure that a neural network can find the right weights. The energy is parametric and as justified in our submission the gradient flow amounts to residual graph convolution with symmetric weights (which has same expressive power as residual graph convolutions) so in this regard this is a standard deep learning framework. We think an important point  is that _almost any_ square matrix $\mathbf{W}$ is diagonalizable; for any such matrix the same spectral analysis of attraction and repulsion would apply. One should read our approach as: if the graph convolution is symmetric, then the energy decreases and hence all the spectral analysis applies. From a practical point of view though, this is no different than using residual graph convolution with symmetric weights so there is no serious constraint on our side that would compromise the expressive power.
> > > >
> > > > "_Also, what does 'running the dynamics for too long' mean? (...)_"
> > > >
> > > > Thanks for the question. In the current formulation, since we are _sharing_ the weights and hence we cannot dampen the evolution when stacking many layers, generally the performance starts dropping around 12 layers -- we have not conducted extensive experiments in this regard. Your question seems to be based on a point that for us was actually crucial to clarify precisely thanks to our theory. In the literature there is an understanding that generally graph neural networks may "over-smooth". This has been empirically characterized as performance dropping with many layers; on the other hand, this has a more precise theoretically characterization of "the solution becomes too smooth" as it can be measured via the Dirichlet energy.
> > > > Our paper shows something more general: if you have residual convolutional models, then Theorem 4.1 shows that either (i) the positive eigenvalues of $\mathbf{W}$ dominate so that attractive forces in the long run lead to over-smoothing or (ii) the negative eigenvalues of $\mathbf{W}$ dominate so that repulsive forces in the long-run lead to over-separation. So even if you have repulsion, if you keep repelling at the end the performance will drop even though this is _not_ over-smoothing. The goal of our submission was not to introduce new techniques to build very deep architectures but to provide a better theoretical framework to explain that (i) drops in performance occur for graph convolutions because you either have too attraction or too repulsion in the long term and (ii) to properly study over-smoothing on a theoretical level one needs to normalize the Dirichlet energy as per Definition 3.1.
> > > >
> > > > 2. We hope we have clarified your point above. As already argued in the paragraph below Corollary 4.2, Theorem 4.1. actually tells you that graph convolution models (with shared weights) will either over-smooth or over-separate in the limit. So the drop in performance is _bound to happen_. However, as argued in the same paragraph, given the initial encoder and a finite number of layers (<= 10) one can achieve competitive performance already. Again, the main goal of our submission though was specifying that the drop in performance in deep GNNs has to do with the forces generated by the spectrum of $\mathbf{W}$ -- if you don't have repulsion you can fail on heterophilic graphs, but if you do have repulsion and run the dynamics for too long then you will inevitably over-separate which is still bad. The design of architectures to achieve similar performances even with many layers was not part of the scope.
> > > >
> > > > 3. Thanks for your feedback -- unfortunately such an ablation at this stage would not be possible  but we would be more than happy to include it in the camera ready.
> > > >
> > > > 4. Thanks for the reference, we will include it.
> > > >
> > > > 5. Thanks for the point. Again, we believe our experiments are extensive when compared to all the previous literature mentioned in our rebuttal and we hope you can see how this can not be a big point of criticism in light of that.
> > > >
> > > > We kindly wish to note how all your points pertain the empirical side. We again emphasize that our main contributions are theoretical. In light of our general response about technical novelty, we strongly believe that _the theory of this submission $\textbf{cannot}$ be deemed "marginal"_ and if you disagree (about the technical/theoretical contributions) we kindly ask you to point us why, otherwise we hope you could revise your score because of that. Note how our theory has indeed also addressed something important about the drop of performance of graph convolutions even when we have repulsion due to over-separation.

---

### Official Review · Reviewer_zvZU · 2022-10-24

**Confidence:** 2
**Correctness:** 2
**Technical Novelty And Significance:** 2
**Empirical Novelty And Significance:** 2
**Recommendation:** 3

**Clarity, Quality, Novelty And Reproducibility:**

Mathematically, gradient flow (in a linear space) -- or steepest descent curve -- is a smooth curve x : R --> X (a linear space) such that $x'(t) = - \nabla E(x(t))$ for a linear function $E: X-->R$.

I assume that in the paragraph "What is a gradient flow" in Section 2, F(t) is the gradient flow, and ODE(F(t) indicates $\dot{F}(t)$?

Can you provide some justification for eq.(5) (and equivalently eq.(11)), and why it represents the gradient flow for the "energy functional" defined in eq.(6)?    In other words, given eq.(6), why does eq.(11) represents its gradient flow?

The same question applies to eq.(10).

Furthermore, how does the "graph" (or adjacency matrix A) come to the picture? What is the underlying space/domain X that the energy function is defined, and how it evolves across the space/domain over time?

There are a lot of "hand-wavy" statements. What are "features"  $f_i$ and $f_j$?

In practice, what do you actually try to learn, the "energy functional" or "gradient flows"? I assume the former.
In terms of experiments, your eqs. (17) and (18) basically provide an "evolution" equation. How can you ensure that it gives the right "gradient flow" for the energy functional that you try to learn?  With the specific forms you used, basically the second part in the right hand side of eq.(17) (or eq.(18) gives us $\nabla E$. But since it involves the parameters that you need to learn, how do you ensure that it provides the "correct" gradient of the energy function?  In the end, it seems that you are just blindly using GNNs to learn some function based on some loss functional. We end up in square one --- we still do not necessary learn what the GNN actually learns.

A more basic question: can you clearly define what you mean by the gradient operator $\nabla$,  e.g., $\nabla F$ (or $\nabla E_{\theta}$) that you used in the paper? Note that in general, $\nabla$ operator is defined using a (local) Euclidean coordinate system.  Here you are talking about a graph. What is the underlying domain in which either the energy function $E_{\theta}$ or the gradient flow $F$ is defined? If the nodes in the "graph" represent "particles" in an N-particle system, these particles still "reside" within, say, a 3-dim Euclidean space.

**Details Of Ethics Concerns:**

There are  no ethics concerns.

**Strength And Weaknesses:**

Strengths
   + Provides an interesting (albeit not completely new) perspective to view GNNs from the lens of gradient flows (of a energy functional).

    + Detailed proofs of the mathematical claims made in the paper

Weaknesses:
   - There are implicitly assumptions that are not clearly stated and some of the terminologies seem to be loosely used (see the next section for more details);
   -  There also appear to be gaps between the "theoretical framework" and how it is applied in practice (with discrete datasets).

**Summary Of The Paper:**

Inspired by gradient flows as (solution curves to) differential equations that minimize an energy functional, the authors view GNNs as a gradient flow equation of a parametric energy. The authors show that in graph convolutional models (GCN), the positive/negative eigenvalues of the channel mixing matrix correspond to attractive/repulsive forces between adjacent features. They also demonstrate how the channel-mixing can learn to steer the dynamics towards low or high frequencies, which allows to deal with heterophilic graphs.
The authors provide a use case to show experimentally how  the gradient flow framework is more efficient than GCN, and achieves competitive performance on graph datasets of varying homophily often outperforming recent baselines specifically designed to target heterophily.

**Summary Of The Review:**

The paper presents an  interesting (albeit not completely new) perspective to view GNNs from the lens of gradient flows (of a energy functional).  While I appreciate this new lens to view GNNs, there are a lot of implicit assumptions, confusing notions, vague definitions and various formulas that are not well explained or justified. As such, it makes decipher the mathematical statements and proofs hard to follow and ascertain. There are a number of ad hoc constructions. It is unclear the proposed framework really provides any deeper insight into how GNNs work.

---

> ### Author Response · Authors · 2022-11-08
> **Thank you for the review: rebuttal part 1**
>
> Thank you for the review. Below we address each concern individually.
>
> First, we note you claim on two occasions that our paper is _not completely new_. To the best of our knowledge, we are the first to present this physics-inspired approach to GNNs as starting from an energy rather than the equations: this has led to 1. an in-depth theoretical analysis of the learned node presentations (Theorem 4.1) and to 2.  an efficient architecture that can be competitive on heterophilic graphs that were thought of as too challenging for graph convolutional networks. If you know of references arriving to the same ideas and or theory, could you to share them?
> Otherwise, we feel that the statement is not entirely backed and hence our contributions are actually novel.
>
> "_There are implicitly assumptions that are not clearly stated and some of the terminologies seem to be loosely used (see the next section for more details) (...)._"
>
> Thank you for expressing your concerns. However we respectfully disagree and feel this is quite undeserved and hope we will convince you why in the rebuttal. Any statement refers to its assumptions explicitly, there are 14 pages of mathematical details and proofs in the appendix which we stand by that confirm the rigour and accuracy of the paper: if you have questions about part of the theory you are unsure about, we would be more than happy to address those. Most importantly, the framework we have used for validation, is the same we have developed our theory on and there are no gaps in this regard.
>
> "_Mathematically, gradient flow (in a linear space) -- or steepest descent curve -- is a smooth curve x : R --> X (a linear space) such that $x^{\prime}(t)=−\nabla E(x(t))$ for a linear function $E:X−−>R$._"
>
> There is no need to require the space to be linear (it works for manifolds too) nor to ask the function to be linear and indeed the energy $\mathcal{E}_{\theta}$ used throughout our paper (see eq. (10)) is non-linear.
>
> "_assume that in the paragraph "What is a gradient flow" in Section 2, F(t) is the gradient flow, and ODE(F(t) indicates $\dot{F}(t)$?_"
>
> To be slightly more precise, $F(t)$ is the solution to the gradient flow and gradient flow generally indicates the whole differential equation $\dot{F}(t) = -\nabla E(F(t))$.
>
> "_Can you provide some justification for eq.(5) (and equivalently eq.(11)), and why it represents the gradient flow for the "energy functional" defined in eq.(6)? In other words, given eq.(6), why does eq.(11) represents its gradient flow?_"
>
> Thanks for your questions.
>
> Justification for eq.(5): as explained in the paper, in eq. (5) we have considered a general family of GNNs that include very common and important baselines like (residual) GCN, GraphSAGE, and GCNII to name a few and generally pertain to the class of GNNs with convolutional flavour as they are referred to in the _Geometric Deep Learning_ book by Bronstein et al.
>
> About  eq. (11): this is the Euler discretization of eq. (9). The latter is the gradient flow of eq. (6) since the right hand side of (9) is minus the gradient of the energy in (6): this follows from $\textbf{direct computation}$ $\textbf{by simply}$ $\textbf{differentiating}$ $\textbf{eq. (7) with respect to F}$. We hope that this clarifies your question, and that you see how we have already justified in our paper why (11) is the discretized gradient flow of the energy in eq. (6). If you have some input to better rephrase this explanation, please let us know.
>
> "_The same question applies to eq.(10)._"
>
> We are not sure what question you are referring to here. In (10) we simply rewrite the eq. in (6) by showing how the positive eigenvalues of $\mathbf{W}$ contribute to attractive forces in the energy while negative eigenvalues lead to repulsive forces.
>
> "_Furthermore, how does the "graph" (or adjacency matrix A) come to the picture? What is the underlying space/domain X that the energy function is defined, and how it evolves across the space/domain over time?_"
>
> Thanks for your question. The energy is defined on the space of node features i.e. $\mathbb{R}^{n\times d}$: indeed, $\nabla \mathcal{E}$ refers to the gradient of $\mathcal{E}$ wrt the features $\mathbf{F}\in\mathbb{R}^{n\times d}$. The graph enters the picture in (6) via the (normalised) adjacency $\bar{\mathbf{A}}$ since the pairwise terms in the energy are restricted to the edges – see $\mathcal{E}^{\mathrm{pair}}_{\mathbf{W}}$ in eq. (6). Indeed, in the gradient flow equation in (11) you can see that there is a message-passing block of the form $\bar{\mathbf{A}}\mathbf{F}(t)\mathbf{W}$ as is typical in classical architectures like GCN. We hope that this clarifies your questions and you agree that this information was already present in the manuscript.

---

> > ### Author Response · Authors · 2022-11-08
> > **Thank you for the review: rebuttal part 2**
> >
> > "_There are a lot of "hand-wavy" statements. What are "features" $\mathbf{f}_i$ and $\mathbf{f}_j$_"
> >
> > Thank you for the question. We respectfully believe this comment to be inaccurate though, and we hope you can see that our paper is very rigorous and all but hand-wavy. In fact, for the convenience of the reader we have already written a notations paragraph at the end of pag. 2 _explaining all our notations including your question_ i.e. $f_i$ is the feature of node i. Furtherher when we write $f_i(t)$, then we refer to the feature of node i after time t (i.e. after t layers) something that is common in any GNN paper we are aware of. What other statements do you believe are ‘hand-wavy’?
> >
> > "_In practice, what do you actually try to learn, the "energy functional" or "gradient flows"? I assume the former._"
> >
> >  In practice one learns the gradient flows which are the update equations. Note however that the equations are minus the gradient of the energy, so by learning the gradient flow you are also learning the right energy.
> >
> > "_In terms of experiments, your eqs. (17) and (18) basically provide an "evolution" equation. How can you ensure that it gives the right "gradient flow" for the energy functional that you try to learn?_"
> >
> > (17) is a special form of (11) that we explained in one of our previous comments above to be the  gradient flow of the energy in (6). If there are mathematical details that are unclear please let us know.
> >
> > "_With the specific forms you used, basically the second part in the right hand side of eq.(17) (or eq.(18) gives us
> > $\nabla E$ (...) we still do not necessary learn what the GNN actually learns._"
> >
> > Thanks for your questions. There are no multiple notions of gradient for the energy functional. There is only one, and this is the one used to derive the equations (17), (18). Our application is not blind. _First note that of course you want your energy to depend on some parameters otherwise there is no learning_: think of the examples in Section 2 where if we used the Dirichlet energy (without parameters) our gradient flow would be the heat equation and we would do smoothing no matter what (even when we might not need it due to heterophily). Second, our theory points to the fact that we know what the GNN is learning: it is learning the energy forces which are represented by the eigenvalues of the channel-mixing matrices. Note how we validated our theory in the synthetic Cora experiments in Figure 2 by justifying how positive and negative eigenvalues of the channel-mixing are being used -- see also the new Table 9 in Section E.8 in the appendix. So we actually have a better understanding compared to GNNs where there is no energy and hence there are no attractive/repulsive forces generated by the eigenvalues of the channel-mixing. We hope that this clarifies your concerns.
> >
> > "_A more basic question: can you clearly define what you mean by the gradient operator (...)_"
> >
> > Thanks for your questions. Although we think we have already addressed these doubts in a separate comment, let’s emphasize this point here as well. The energy is defined in feature space i.e. $\mathbb{R}^{n\times d}$ –  as explained in a comment above the graph structure enters the picture since we sum pairwise terms only on the edges of the graph. So the gradient of the energy is the _classical Euclidean gradient_ in $\mathbb{R}^{n\times d}$. When instead we are talking about node features, as explained on pag. 3 (and how customary done in other papers like GRAND, BLEND, SHEAF), $\nabla \mathbf{f}(i,j)$ is the gradient of the node features along the edge (i,j) i.e. simply the difference $\mathbf{f}_i / \sqrt{d_i} - \mathbf{f}_j / \sqrt{d_j}$. We hope this again clarifies one of your main concerns regarding the definition of the energy and its gradient.
> >
> > "_there are a lot of implicit assumptions, confusing notions, vague definitions and various formulas that are not well explained or justified. (...) There are a number of ad hoc constructions._"
> >
> > We hope that our replies actually clear out any doubt. We believe that our paper does not have vague definitions, confusing notations nor ad hoc constructions: if there are *specific parts* you are concerned about, please let us know? We have reported a notations paragraph at the end of Section 2 to guide the reader and an additional notation paragraph at pag. 14 for the appendix.
> >
> > $\textbf{Final comment}$:
> > In light of all our comments and responses, we kindly ask to revise your score since we feel that the low score was motivated by concerns regarding definitions and approaches that we hope have been resolved. _If you feel that there are questions still left open please let us know (with specific pointers) and we will do our best to address them since we truly believe that our submission is all but “hand-wavy” or “vague” and indeed believe it to be extremely rigorous_. If you also have input to improve certain parts of the manuscript, let us know and we will do our best to revise.

---

### Official Review · Reviewer_DfeD · 2022-10-25

**Confidence:** 4
**Correctness:** 2
**Technical Novelty And Significance:** 3
**Empirical Novelty And Significance:** 3
**Recommendation:** 5

**Clarity, Quality, Novelty And Reproducibility:**

The paper is clearly written, and of good quality. Aspects of the paper are novel and contribute to the field of building physics-inspired GNNs. The code is not available for anonymity reasons but I believe that it will be provided if the paper is accepted. There are some implementation details that are not clear such as the choice of the ODEsolver and the hyperparameters but I believe that these issues can be easily fixed.

**Details Of Ethics Concerns:**

No specific concerns to report.

**Strength And Weaknesses:**

Strength: The topic is definitely timely and interesting, and the general approach relevant. The paper is generally well-written. There are multiple concepts of interest, as using Dirchlect energy to quantify HFD/LFD nature of node features, emphasising the importance of residual connections, the attractive-repulsive multi-particle perspective.

Weaknesses: The motivation, particularly with respect to the core idea of energy is unclear. Out of oversquashing, over-smoothing, and heterophily, the authors tell us that their approach is suited to arrive at principled GNNs, guided by the perspective of energy functional/gradient flow. However, the relevance of the energy function/ gradient flow to this is not entirely clear - in fact most of their analysis, proofs, and lemmas, continue to be based on the evolution function, and on the Dirichlect energy, that is used to analyse high-frequency and low-frequency asymptotic behaviour of the latent feature evolution.


**Summary Of The Paper:**

The authors present an approach of viewing the (residual) GNNs / graph ODEs through the point of an energy and corresponding gradient flow. In other words, the residual GNNs are viewed as derived from taking the gradient of a parameterised energy function. One of the ‘take-home’ points emphasised throughout the paper is the aspect that one should parameterise the energy and not the evolution equation of the GNNs.


**Summary Of The Review:**

Below are a few comments that could hopefully improve the quality of the paper. They are mainly related to the intuition, motivation, and validation of the approach.

Motivation of the energy view:

- The energy functional itself is arrived in a sort of reverse-engineered manner by starting from the evolution equation (5), without the non-linearity $\sigma$. Equation (5) itself seems to be presented without explaining why such a specific form alone would be necessary, but that does not affect the analysis seriously. Once the energy functional is defined, we are shown how the mixing matrix and its spectra regularizes the frequency spectra asymptotically — that such an energy helps mediate both smoothness (through positive eigenvalues of the mixing matrix W) and non-smoothness or high-frequency behaviour (through negative spectra of W), which is often connected to heterophily. This is an interesting concept, but not developed fully.  From Section 3 onwards, we see that the rest of the analysis takes the form of a regular graph/neural ODE such as GRAND, and does not directly make use of the energy except in Section 5.

- In the end of Section 3, the authors argue ‘We argue that energies rather than evolution equations should be the object to parameterise for deriving more principled GNNs that are easier to interpret and analyse. ‘ This argument continues throughout without really being made clear of its necessity or advantages.

- I like the perspective of the parameterised inspired from ‘finding the right notion of smoothness for our task’ as mentioned on page 4. However, beyond this minor intuition, I am unable to see the importance or the unique bias the energy view of GNNs brings over the evolution equations.

- Lot of the later arguments regarding LFD/HFD is done using Dirichlect energy and seems disconnected from the energy functional we are introduced to before. This characterisation is nonetheless interesting and of value. However, the connection between HFD and heterophily is not obvious, particularly given that we are not shown how these properties pop up in the actual benchmark datasets, specially in low-homophily ones.

Experimental validation

- The authors present to us the interesting case of a synthesized Cora dataset at different levels of homophily. We are shown comparison with cases where the mixing matrix W is strictly negative in spectrum (neg-prod), positive in spectrum (‘prod’), and a general W. We are shown that in low homophile cases, the ‘neg-prod’ seems to have better performance over ‘prod’, and the other way round for high-homophily. However, in all cases, the individual performances never exceeds having a general W which means even in extremely low homophily situations, the energy functional learns to retain a component which promotes smoothness across features of LFD nature. This is interesting, and not entirely clear why - given the argument that heterophily essentially would be better off to have ‘repulsion’ amongst the particles.

- The authors should better explain why the performance still remains lower than an MLP. Further, it would be advisable to show the reader the relative spectral components (signs of eigenvalues) of the learned W for different homophily levels — this would help validate the hypothesis that indeed the attraction repulsion dynamic is in play (as discussed in section 3.2). Also given that Graph isomorphism convolution layer (GINConv) [R0] is known to be one of the most expressive graph convolution layers available, I suggest to make comparisons with that also as a baseline in the experiments.

- Similarly, it would be interesting to see the spectral aspects of the mixing matrices learnt for all of the real-world  heterophily datasets reported in Table 1 - do the learnt W and their eigenvalues mostly agree with the sign that is expected from the proposed analysis through Section 3.2?


Other minor comments:

- Theorem 5.1, I believe should be seen as Lipschitz continuity and not monotonicity as the authors claim on page 7.

- The performance of GRAFF and GRAFF_{NL} is often fairly close — does this mean that the non-linearity  $\sigma$ has relatively no role to play? Alternatively, how does the spectrum of the learnt W matrices differ for both the cases for a given dataset?

- Before Section 5, the authors statement ‘Convolutional GNN models can deal with heterophily if the channel mixing matrix has negative eigenvalues. ‘ — does not seem clear or substantiated. The authors should make this clear, or explain this through examples.

- In the experiments on benchmark data, we observe that in many cases, GGCN or even GCNII (that are known to be special cases of equation (5)) outperform GRAFF even at low homophily levels. Can this be explained in some way? Perhaps through the associated learnt energy functionals — currently I don’t see how the energy functional view has helped us in understanding these observations.


- Finally, since the entire framework is based on dynamical systems, it would be nice to see if the model is able to capture actual dynamic systems, as considered in [R1] for example.



[R0] How Powerful are Graph Neural Networks? K. Xu, W. Hu, J. Leskovec, S. Jegelka. ICLR 2019

[R1] Chengxi Zang and Fei Wang. 2020. Neural Dynamics on Complex Networks. In Proceedings of the 26th ACM SIGKDD International Conference on Knowledge Discovery & Data Mining (KDD '20). Association for Computing Machinery, New York, NY, USA, 892–902. https://doi.org/10.1145/3394486.3403132

---

> ### Author Response · Authors · 2022-11-08
> **Thank you for the review: rebuttal part 1**
>
> First, we would like to thank you for the detailed feedback and for the valuable comments. We address them below one by one and kindly ask to let us know if there are further doubts or clarifications required. Thanks to your input we have also already revised the submission (see general comment for a list of revisions).
>
> You have assigned a _Correctness = 2_ score: we hope that this rebuttal will convince you that our submission does not satisfy the description “Several of the paper’s claims are incorrect or not well-supported” and that in light of this you might consider revising the score.
>
> "_However, the relevance of the energy function/ gradient flow to this is not entirely clear - in fact most of their analysis, proofs, and lemmas, continue to be based on the evolution function, and on the Dirichlect energy, that is used to analyse high-frequency and low-frequency asymptotic behaviour of the latent feature evolution._"
>
> Thank you for such observations. This is a crucial point: we clarify this (and most of the related doubts raised) here. We note that all theorems and proofs will completely break if the evolution equations we are considering are not gradient flows of the energy $\mathcal{E}_{\theta}$ used throughout the paper. Indeed, the theoretical results require symmetry of the weights entering (9) and (11), and the symmetry holds $\textbf{if and only if}$ the equations (9) and (11) are gradient flows of the energy in (6).
>
> The relevance of the energy is to provide the equations with a specific form – i.e. they are minus the gradient of the energy. The very design of the energy – as explained in eq. (10) – was to enable both pairwise attraction and repulsion along edges: one then uses the evolution equations (i.e. the forces generated by such energy) to prove more rigorous asymptotic analysis. What is the advantage of the energy approach in practice? Convolutional architectures that are not symmetric and do not satisfy the gradient flow paradigm like GCN and GCNII have been consistently outperformed by the (more efficient, as per Figure 5) gradient flow we have proposed – which provably induces repulsion -- see also the new Section E.8.
>
> "_The energy functional itself is arrived in a sort of reverse-engineered manner by starting from the evolution equation (5), without the non-linearity $\sigma$. Equation (5) itself seems to be presented without explaining why such a specific form alone would be necessary, but that does not affect the analysis seriously._"
>
>
> Thanks for the feedback, this is a point that is very important for us and wish to clarify. The energy is not derived in a reverse-engineered manner. While due to space constraints in the main body of the paper we decided to present the easiest case of an energy that generalises the Dirichlet one (hence leading to gradient flow equations that are convolutional), in the appendix we present our true motivation of extending the harmonic map flow from manifolds (images) to graphs – see pag. 23 and Proposition B.6. In light of your comment, we have added an extra paragraph at pag. 24 titled “A more general energy” that explains how the energy in eq. (6) can in fact be derived from the harmonic map energy in Proposition B.6 which we have adapted from image pde analysis. Concerning instead eq. (5), note that we have observed how this is a sufficiently general class of GNNs including common architectures like GCNs, GraphSAGE, GCNII and their residual counterparts.
>
> "_This is an interesting concept, but not developed fully. From Section 3 onwards, we see that the rest of the analysis takes the form of a regular graph/neural ODE such as GRAND, and does not directly make use of the energy except in Section 5._"
>
> Thank you for your concerns. We have addressed this above – our in depth-analysis is made possible precisely because the graph/neural ODE are the gradient of an energy.
>
> "_the authors argue ‘We argue that energies rather than evolution equations should be the object to parameterise for deriving more principled GNNs that are easier to interpret and analyse. ‘ This argument continues throughout without really being made clear of its necessity or advantages._"
>
> Thank you for raising this question. In this paper we show that if you start from an energy (6) you can write down evolution equations (11) that satisfy nice theoretical properties (mainly reported in Theorem 4.1 and for the non-linear case in Theorem 5.1) and lead to an architecture that is more efficient than GCN (see the runtime in Figure 5) and outperforms recent baselines specifically designed to target heterophily. Without an energy, one would have to come up with evolution equations by guess. Note that if your guess is `right’ (i.e. you come up exactly with (9) and (11)) this is $\textbf{ equivalent to asking that there exists an energy as in (6) being minimized}$.

---

> > ### Author Response · Authors · 2022-11-08
> > **Thank you for the review: rebuttal part 2**
> >
> > "_I like the perspective of the parameterised inspired from ‘finding the right notion of smoothness for our task’ as mentioned on page 4. (...)_ "
> >
> > Thank you for your feedback. As above, we note how the existence of energy grants a form to the equations where the channel-mixing is symmetric and shared so that its eigenvalues are now powerful enough to induce the LFD/HFD dynamics derived in Theorem 4.1 and Corollary 4.2.
> >
> > "_Lot of the later arguments regarding LFD/HFD is done using Dirichlect energy and seems disconnected from the energy functional we are introduced to before (..)_"
> >
> > Thank you for the comments. While the characterization of LFD/HFD is done wrt a Dirichlet energy, the property that the gradient flow can be LFD/HFD (depending on the eigenvalues of $\mathbf{W}$) is precisely a benefit of the introduced energy functional, that can adaptively learn to be mostly attractive/repulsive along edges. We believe that your suggestion of monitoring the frequency in the heterophilic case is valuable and in this regard we have added two rows in the new Table 9 reporting the normalized Dirichlet energy (used for the LFD/HFD definition) of the encoded node features and of the learned node features after the diffusion layers.
> >
> > "_However, in all cases, the individual performances never exceeds having a general W (...)_"
> >
> > We agree with you that this is an interesting observation – note how a general $\mathbf{W}$ can still learn to have mostly repulsion but the existence of positive eigenvalues may help with minor order effects.
> >
> > "_The authors should better explain why the performance still remains lower than an MLP (...)_"
> >
> > Thank you for analysing in detail our synthetic experiments which we believe to be of value. The fact that the MLP may be better under low (but not too low homophily) is expected, since in that range of homophily the feature information is more valuable than the graph structure. Put it differently, a random label distribution is heterophilic and makes the graph useless (i.e. MLP is good enough). A high-frequency label distribution is more heterophilic than a random one, but more correlated with the graph structure (this is where enhancing high frequencies is beneficial and hence GNN are useful). Note that in the synthetic case the prod and neg-prod parameterisations only contain positive and negative eigenvalues respectively so their behaviour validates our spectral analysis. We will consider including GINconv although we note how this is typically a baseline for graph-level tasks and all baselines reported in the experiments do not include it. About reporting the learnt spectra we think this a very valuable suggestion so we have decided to report such numbers for real-world datasets (see below).
> >
> > "_Similarly, it would be interesting to see the spectral aspects of the mixing matrices learnt for all of the real-world heterophily datasets (...)_"
> >
> > Thank you for this suggestion. We have now added Table 9 along with several comments to discuss the results concerning normalized Dirichlet energy and learnt eigenvalues.
> >
> > "_Theorem 5.1, I believe should be seen as Lipschitz continuity and not monotonicity as the authors claim on page 7._"
> >
> > Thanks for the detailed analysis. The monotonicity we refer to is in regard of the continuous dynamical system (i.e. step size
> > $\tau \rightarrow 0$):  the time derivative of the energy is negative along the evolution -> we have decreasing monotonicity. We have rephrased it in the discrete setting (finite positive step size) below the statement of Theorem 5.1 and argued that it can indeed be seen more as a Lipschitz continuity property.
> >
> > "_The performance of GRAFF and GRAFF(NL) is often fairly close, does this mean that the non-linearity $\sigma$ has relatively no role to play?_"
> >
> > This is a correct observation. Yes on these datasets removing non-linearity is not crucially important – something observed for example in _Simplifying graph convolutional networks_.
> >
> > "_Before Section 5, the authors statement ‘Convolutional GNN models can deal with heterophily if the channel mixing matrix has negative eigenvalues. ‘ — does not seem clear or substantiated._"
> >
> > Thank you for asking for clarifications.  On a theoretical level, we believe that this has been substantiated since Theorem 4.1 and Corollary 4.2 show that if the channel-mixing has enough negative eigenvalues, the system will induce repulsion meaning that the GNN will also enhance the high-frequency. On an empirical level, this has also been substantiated by the synthetic CORA given what happens to the prod configuration for example. In terms of real-world experiments, we invite you to compare the performances of GRAFF with those of GRAND and CGNN in Table 1: in Theorem 3.3 we have proved that GRAND and CGNN in fact cannot learn to induce repulsion and hence will provably fail on heterophilic graphs. We have also now added Section E.8 as per your previous suggestion.

---

> > > ### Author Response · Authors · 2022-11-08
> > > **Thank you for the review: final part**
> > >
> > > "_In the experiments on benchmark data, we observe that in many cases, GGCN or even GCNII (that are known to be special cases of equation (5)) outperform GRAFF even at low homophily levels. (...)_"
> > >
> > > We are not entirely sure we understand your question. Where is GRAFF being outperformed by GGCN and GCNII? If you are referring to the fact that GRAFF *outperforms* them, then we point back to the previous comment where we have explained how the existence of an energy leads to shared, symmetric channel-mixing matrices that can induce repulsion thanks to their negative eigenvalues as per Theorem 4.1.
> > >
> > > "_Finally, since the entire framework is based on dynamical systems, it would be nice to see if the model is able to capture actual dynamic systems, as considered in [R1] for example._"
> > >
> > > This is a valuable reference we have included in the revised version, thank you for pointing this to us. Their setup is quite different from ours and unfortunately it will  be beyond the scope of the rebuttal to compare with them, and may be inspiration for future work.
> > >
> > > _Correctness 2:_
> > >
> > > $\textbf{Final comment}$: Thanks for the feedback and the many comments and questions which we believe have helped to improve the revised version. The paper has 14 pages of in-depth mathematical proofs in the appendix whose correctness we stand by. If there are particular concerns we would be more than happy to clarify. In terms of statements about the usefulness of the energy vs evolution equations raised above, we hope we have clarified them. We have also reported learnt spectra as suggested to improve the message. In light of the rebuttal, we would like to ask to revise the correctness score which we feel is undeserved for a paper we believe of mathematical rigour and sufficient empirical evaluation. We also hope that based on this, you will consider raising the final score.

---

### Author Response · Authors · 2022-11-08
**Thank you for the reviews: general comment + list of revisions**

We thank the reviewers for their feedback and comments which we have done our best to address below and that have already been taken into account in our revised version. We also thank them for finding important pros in our submission, including that _the topic is definitely timely and interesting_ that _there are multiple concepts of interest, which contribute to the field of building physics-inspired GNNs_ , _ provides an interesting perspective to view GNNs_,  _the paper offers a quality analysis of GNNs through their energy functions_ and that _the presented theoretical framework is novel and interesting to read and provides a new perspective on GNNs_. We would like to comment on some important points that we believe to be of general interest for all reviewers and then address any question/comment in detail separately.

$\textbf{About correctness}$: Two reviews have assigned a correctness score of 2. We have tried to clarify any doubt/concern below but would also like to point out that a score of 2 in correctness should be motivated by incorrect or unsupported statements. This is a theoretical paper where each result is fully proved and justified. Which of our results are incorrect/unsupported?
We hope that this rebuttal will convince the reviewers that the paper cannot be labelled as “Several of the paper’s claims are incorrect or not well-supported” and we are more than happy to address any further standing doubt.

$\textbf{About technical novelty}$:  We are the first to:
- realise the connection between graph convolutions and gradient flows. This has led to a rich theory about spectral analysis of the GNN and the learned node representations _along with_ convolutional architectures that can perform competitively on heterophilic graphs which were traditionally thought of as being too challenging for such models. Our theory in thm 4.1 and 5.1 fully explains why this is the case.
- introduce scale-aware way of assessing over-smoothing and in general the frequency response of a GNN via the LFD/HFD characterization provided in Definition 3.1 that can be borrowed by any other follow-up theoretical work.
- prove that many existing ODE-based GNNs cannot be HFD and hence will struggle with heterophily (Theorem 3.3) as validated empirically (Table 1)
- prove that the interaction of the graph and channel-mixing spectra is key for the GNN dynamics (Theorem 4.1)
- Extend this analysis to the non-linear case by proving monotonicity (Lipschitz regularity) of an energy more general than Dirichlet and wrt to an infinite class of non-linear maps (Theorem 5.1).

We believe that the technical novelty of the submission cannot be labelled as “only marginally significant or novel” and have done our best in the rebuttal to address any of the reviewers’ questions and doubts in this regard.

$\textbf{About empirical novelty}$: A consequence of our theoretical analysis is the construction of a gradient flow architecture that
- Is faster than GCN (see ablation study in Figure 5 concerning runtime)
- Can provably induce repulsion through the negative eigenvalues of the channel-mixing as per Theorem 4.1 and hence deal with heterophily. We kindly invite the reviewers to read the methodology paragraph at pag. 28 in the Appendix and see the new Table 9.
- Can be competitive and in fact better than recent baselines specifically designed to target heterophily by often relying on slower edge-wise (attention) operations. This has the value of showing that simple graph convolutional models are stronger baselines if we choose parameterisations (mainly symmetry and control of the eigenvalues) that are inspired by physics.

$\textbf{What’s new in the modified version of the manuscript?}$ In light of feedback from the reviewers the submission has been revised as follows:
- Included references suggested by DfeD and Hwvc.
- Rephrased slightly how to interpret Theorem 5.1 as per DfeD’s comment (Lipschitz regularity).
- Added an extra paragraph at pag. 24 titled “A more general energy” that explains how the energy in eq. (6) can in fact be derived from the harmonic map energy in Proposition B.6 which we have adapted from image pde analysis (as per DfeD’s question).
- Added Section E.8 along with Table 9 in the appendix where we report and discuss spectral values of the learnt channel-mixing and normalized Dirichlet values of real datasets (2 homophilic and 2 heterophilic) as per DfeD’s comment to emphasize connections with the theory.
- Added a pointer to eq. (39) at pag. 7 based on feedback from Hwvc.
- Rephrased the paragraph below eq.(17) to point at where the reader can find more details about methodology and parameterisations used.
- Added Section E.7 and Table 8 to include further baselines and comments on the results as suggested by Hwvc.

_We now address each of the reviews individually below._

---

### Author Response · Authors · 2022-11-18
**About engagement**

Dear ACs and reviewers,

We would like to kindly point out that we have got no engagement concerning our rebuttal despite getting to the end of the rebuttal window and our rebuttal being posted 10 days ago.

We believe we have addressed all the questions and concerns raised in the reviews and argued why we think our submission to be a valuable contribution especially in terms of technical novelty and correctness.

We hope you can have a chance to go through our rebuttal as well as our revisions listed below; we also hope that you do appreciate how any chance of engaging in discussion and conversation will be quite limited now given the time at disposal.

---

### Decision · Program_Chairs · 2023-01-20

**Decision:**

Reject

**Justification For Why Not Higher Score:**

The main drawback of this paper is its novelty. The gradient flow viewpoint itself is not so much novel. The construction of energy functional would be novel, but it do not give a quite new algorithm. Indeed, the resultant algorithm is rather a combination of existing approaches. Hence, this paper is not sufficiently strong for ICLR publication.

**Justification For Why Not Lower Score:**

N/A

**Metareview: Summary, Strengths And Weaknesses:**

This  paper gives a new characterization of GNNs as a gradient flow to minimize a energy functional that is a combination of training loss, Laplacian smoothness penalty and deviation from the initialization. Based on this viewpoint, the authors proposed two methods called GRAFF and GRAFF_NL and these methods are compared with other methods in numerical experiments.

Strength: The paper is well written. The gradient flow characterization is quite intuitive and instructive to understand what the GNNs are doing in the internal layers.
Weakness: On the other hand, there are some weakness in the paper.
First, the gradient flow viewpoint itself is not new. The ODE characterization has been suggested by several existing work as cited in this paper. The main novelty of this paper is to consider a bit different energy functional with attractive and repulsion power (and deviation from the initialization). However, given recent methods utilizing negative weight edge or tailored to heterophilic data, this kind of proposal does not seem entirely new. A more convincing argument to highlights the novelty would be required. (Actually, the numerical experiments do not show a clear performance improvement over existing methods, though this would not be a main issue.)
Another concern is that this paper mainly discusses the linear ODEs, and there is very limited discussion on the nonlinear activation. This is a challenging issue in the literature. However, Theorem 5.1 for nonlinear activation only asserts that the energy does not increase so much and it does not give any convergence of a properly defined energy.